# Molecular basis for the activation of Aurora A and Plk1 kinases during mitotic entry

Anaïs Pillan [1,2], Philippine Ormancey[1,2], Celia Ben Choug [3], Stephen Orlicky[4,5,6], Nicolas Tavernier [1,2], Lucie Van Hove [1,2], Batool Ossareh-Nazari [1,2], Nicolas Joly [1,2], Frank Sicheri [4,5,6], Thierry Lorca [3] & Lionel Pintard [1,2 ✉]

## Abstract

The evolutionarily conserved, intrinsically disordered protein Bora is critical for initiating the activation of mitotic kinases. Once phosphorylated at Ser112 by Cyclin A-Cdk1 kinase, phospho-Bora activates unphosphorylated Aurora A kinase (AURKA), directing it towards Polo-like kinase 1 (Plk1), thus promoting Cyclin B-Cdk1 activation and mitotic entry. Here, by combining structural modeling and in vitro assays, we provide evidence that Bora wraps around the N-terminal lobe of AURKA to position its phospho-Ser112 near AURKA's T-loop, mimicking T-loop phosphorylation. Additionally, Bora transiently interacts with the αC helix of the Plk1 kinase domain through a conserved motif, guiding AURKA activity towards the Plk1 T-loop, which is otherwise impervious to phosphorylation by AURKA. We highlight the importance of this motif for Bora function in vitro and during mitotic entry in *Xenopus laevis* egg extracts. Our results reveal critical molecular details of mitotic kinase activation, which could lead to the development of pathway-specific inhibitors.

**Keywords** Cell Division; Mitotic Kinases; Plk1; Aurora A; Bora
**Subject Categories** Cell Cycle; Post-translational Modifications & Proteolysis; Structural Biology

## Introduction

During mitotic entry, cells profoundly reorganize their architecture and physiology, which is essential for chromosome segregation (Champion et al, 2017). Evolutionarily conserved serine/threonine kinases, including Aurora A (AURKA) (Llamazares et al, 1991; Glover et al, 1995), Polo-like kinase 1 (Plk1) (Sunkel and Glover, 1988), and Cyclin A/B-Cdk1, orchestrate this reorganization by phosphorylating numerous substrates (Nigg, 2001; Dephoure et al, 2008; Lindqvist et al, 2009; Zitouni et al, 2014; Joukov and De Nicolo, 2018). Plk1 promotes mitotic entry (Seki et al, 2008b; Aspinall et al, 2015; Gheghiani et al, 2017; Gobran et al, 2025), centrosome separation and maturation (Lane and Nigg, 1996), the dissociation of cohesin from chromosomes (Sumara et al, 2002), chromosome condensation (Abe et al, 2011), nuclear envelope breakdown (Linder et al, 2017; Martino et al, 2017), spindle assembly (Sumara et al, 2004), chromosome alignment, spindle checkpoint signaling (Elowe et al, 2007), and cytokinesis (Burkard et al, 2007; Petronczki et al, 2007) (For reviews (Archambault and Glover, 2009; Zitouni et al, 2014; Combes et al, 2017; Pintard and Archambault, 2018; Saurin, 2018)), while AURKA regulates mitotic entry (Hirota et al, 2003; Seki et al, 2008b), centrosome separation, maturation, and bipolar spindle assembly (Hannak et al, 2001; Berdnik and Knoblich, 2002; Giet et al, 2002) (For reviews (Archambault and Glover, 2009; Nikonova et al, 2013; Zitouni et al, 2014; Combes et al, 2017; Pintard and Archambault, 2018; Saurin, 2018; Magnaghi-Jaulin et al, 2019)). Overexpression and misregulation of these kinases have been linked to oncogenic transformations and cancers with poor prognosis. However, despite their fundamental importance for cell division, the molecular mechanism by which these kinases are activated during mitotic entry is incompletely understood and is the focus of our study.

Like most eukaryotic kinases, mitotic kinases have a similar structure consisting of an N- and a C-terminal lobe connected by a flexible hinge region (Bayliss et al, 2012; Endicott et al, 2012). Located between the two lobes is the activation segment (a.k.a: T-loop), which provides regulatory control in response to phosphorylation by stabilizing the kinase in an active conformation that supports substrate-binding and catalysis (Johnson and Lewis, 2001; Huse and Kuriyan, 2002; Beenstock et al, 2016; Welburn and Jeyaprakash, 2018).

Phosphorylation of the T-loop is catalyzed either by autophosphorylation or by a distinct upstream kinase. For Plk1, the T-loop is phosphorylated by AURKA on the Thr210 residue (human Plk1) (Jang et al, 2002; Kelm et al, 2002; Macurek et al, 2008; Seki et al, 2008b). AURKA, by contrast, can autophosphorylate its own T-loop (Thr288 on human AURKA) (Littlepage et al, 2002; Bayliss

[1]Université Paris Cité, CNRS, Institut Jacques Monod, Paris F-75013, France. [2]Programme Equipe Labellisée Ligue Contre le Cancer, Paris, France. [3]Centre de Recherche de Biologie Cellulaire de Montpellier, UMR 5237, Université de Montpellier, CNRS, 1919 Route de Mende, 34293, Montpellier Cedex 5, France. [4]The Lunenfeld-Tanenbaum Research Institute, Mount Sinai Hospital, Toronto, ON M5G 1X5, Canada. [5]Department of Molecular Genetics, University of Toronto, Toronto, ON M5S 1A8, Canada. [6]Department of Biochemistry, University of Toronto, Toronto, ON M5S 1A8, Canada. ✉E-mail: lionel.pintard@ijm.fr

et al, 2003; Eyers et al, 2003; Zorba et al, 2014), but the site is rapidly dephosphorylated by counteracting phosphatases during the G2 phase of the cell cycle (Walter et al, 2000; Katayama et al, 2001; Kang et al, 2014). As dephosphorylation maintains AURKA in an inactive state, this raises the crucial question of how AURKA overcomes the repressive effect of phosphatases to drive mitotic commitment.

AURKA activation is controlled by auto-phosphorylation and dephosphorylation of the T-loop by multiple phosphatases. However, full activation also requires binding to one of several allosteric modulators, including Tpx2 (Targeting protein for Xklp2), Cep192 (Centrosomal protein 192), and TACC3 (Transforming Acidic Coiled-Coil Containing Protein 3). Each of these allosteric modulators also localizes AURKA to specific cellular structures during mitosis, thereby providing an additional level of regulatory control (Joukov and De Nicolo, 2018; Levinson, 2018; Tavernier et al, 2021a).

One of the best-characterized AURKA activators is the microtubule-binding protein Tpx2, which recruits and activates AURKA at microtubules for mitotic spindle assembly (Kufer et al, 2002; Eyers et al, 2003; Tsai et al, 2003). Tpx2 binds the N-lobe of AURKA via its first 43 amino acids, using two separate linear sequences, the M1 (aa 7–21) and M2 (30–43) motifs (Bayliss et al, 2003) (Fig. 1A,B, right panels). By binding to AURKA, Tpx2 induces conformational changes leading to a fully active conformation of the kinase (Fig. 1A, right panel) (Bayliss et al, 2003; Dodson and Bayliss, 2012; Zorba et al, 2014; Burgess and Bayliss, 2015; Cyphers et al, 2017).

The protein Bora (Aurora Borealis) (Hutterer et al, 2006) is another allosteric modulator critically required for AURKA activation during mitotic entry (Fig. 1A, left panel). In particular, Bora stimulates AURKA phosphorylation of the T-loop of Plk1 (Fig. 1A) (Macurek et al, 2008; Seki et al, 2008b; Tavernier et al, 2015; Thomas et al, 2016; Tavernier et al, 2021a, 2021b). However, to do so, Bora must first be phosphorylated by the S-G2 Cyclin A-Cdk1 kinase (Thomas et al, 2016; Vigneron et al, 2018; Silva Cascales et al, 2021) (Fig. 1A). Cyclin A-Cdk1 selectively phosphorylates Bora at 27 serine/threonine sites followed by proline (S/T-P sites), but among these, only the phosphorylated serine 112 is critically required for AURKA activation. Indeed, a phospho-null mutation of this site prevents timely mitotic entry in Xenopus egg extracts and human cells (Vigneron et al, 2018; Tavernier et al, 2021b). Bora phosphorylated at S112 preferentially binds and activates AURKA lacking phosphorylation on its T-loop (as mimicked by a Thr288Val mutation [AURKA$^{T288V}$]) (Tavernier et al, 2021b). This observation explains how AURKA, maintained in its inactive dephosphorylated state by phosphatases in the G2 phase of the cell cycle, can be activated during mitotic entry (Tavernier et al, 2021b). Our working model is that phospho-Ser112 on Bora physically substitutes in trans for the phosphorylated T-loop of AURKA and thereby activates AURKA lacking phosphorylation on its T-loop (Tavernier et al, 2021b) (Fig. 1A, left panel). In turn, AURKA phosphorylates the T-loop of Plk1, thereby activating its catalytic function. The structural basis for this unique mechanism of kinase regulation remains to be determined.

Here, we sought to dissect how phospho-Bora (pBora) binds and activates unphosphorylated AURKA and how this activated form of AURKA can target Plk1. By combining AlphaFold structural modeling and extensive structure–function analysis, we provide

evidence that phospho-Bora wraps around the N-terminal lobe of AURKA to position its essential phosphoserine 112 toward the activation loop of AURKA. More importantly, we show that the T-loop of Plk1 is impervious to AURKA phosphorylation in the absence of phospho-Bora. However, by interacting with the αC helix of the Plk1 kinase domain through an evolutionarily conserved motif, pBora focuses AURKA activity towards the Plk1 T-loop. We demonstrate the importance of this conserved motif in Bora for mitotic entry in Xenopus egg extracts. Phospho-Bora thus selectively activates and localizes AURKA close to Plk1 to trigger its activation and mitotic entry. Our results describe the molecular basis of mitotic kinase activation, which provides insights into how cells meet the challenge of spatially and temporally limiting Plk1 activation during mitosis.

## Results

### Inserting the phosphorylated [PpSP] M3 motif of Bora within Tpx2 is sufficient to potentiate non-phosphorylated AURKA kinase activity

Tpx2 binds AURKA using two separate linear sequences, the M1 (aa 7–21) and the M2 (aa 30–43) motifs (Fig. 1B, right panel) (Bayliss et al, 2003). Several aromatic residues in these motifs anchor Tpx2 to hydrophobic pockets in the N-terminal lobe of AURKA (Bayliss et al, 2003; Eyers and Maller, 2004; McIntyre et al, 2017). In particular, the two aromatic residues $W_{34}$ and $F_{35}$ in the helical M2 motif are required for Tpx2 binding to the αC helix of AURKA (Bayliss et al, 2003; McIntyre et al, 2017).

Sequence comparison with Tpx2 identified two potential short AURKA-binding motifs in Bora, the M1 [amino acids 22-35] and M2 [100-109] motifs containing several aromatic residues ($F_{25}$, $Y_{31}$ in the M1 motif and $F_{103}$, $F_{104}$ in the M2 motif) (Fig. 1B, left panel) (Tavernier et al, 2021b). Notably, the presence of two aromatic residues within the helical M2 motif is highly conserved between Tpx2 ($W_{34}$, $F_{35}$) and Bora ($F_{103}$, $F_{104}$) (Fig. 1B). The M2 motif of Bora is located immediately N-terminal to the essential pSer112 (M3 motif) (Fig. 1B, left panel). Assuming that the M2 motif of Bora engages AURKA similarly to the M2 motif of Tpx2 (Fig. 1A, right panel) (Bayliss et al, 2003), we hypothesized that the phosphate of the M3 motif of Bora would be positioned towards the active site of AURKA, allowing it to functionally replace the missing phosphate on the T-loop "in trans", which is typically supplied by autophosphorylation (Littlepage et al, 2002). This would stabilize the active site of the kinase, thereby potentiating kinase activity (Tavernier et al, 2021a, 2021b).

To test this hypothesis, we fused the M3 phospho-motif of Bora to the first 41 amino acids of Tpx2 to generate a phosphorylated pTpx2-Bora chimera (Fig. 1B, phosphopeptide MK51). We then tested whether the phosphate of the M3 motif increased Tpx2 affinity for unphosphorylated AURKA (i.e., AURKA$^{T288V}$, mimicking unphosphorylated AURKA). Using a competitive displacement binding assay with fluorescein-labeled Tpx2 (Fig. 1C) (Tavernier et al, 2021b), we observed that the pTpx2-Bora chimera displaced the interaction between FITC-Tpx2 and AURKA$^{T288V}$ more efficiently (~threefold) than Tpx2, as measured by fluorescence polarization. This indicated that the pTpx2-Bora chimera bound more tightly than Tpx2 to AURKA$^{T288V}$ (Fig. 1C). By contrast, a

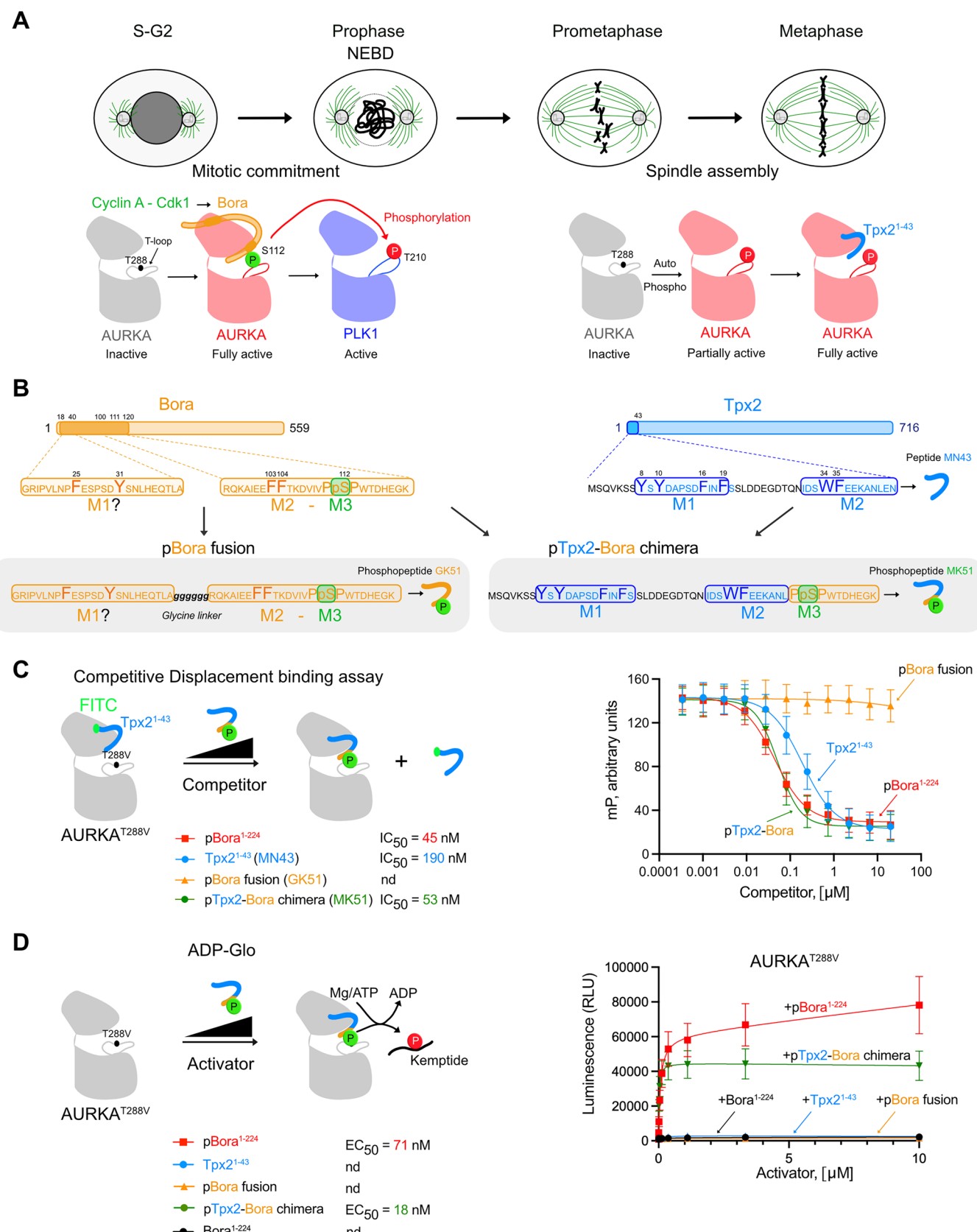

**Figure 1.   A phosphorylated Tpx2-Bora chimera potentiates the kinase activity of non-phosphorylated AURKA.**

(A) Schematic of AURKA activation mechanisms during mitotic entry (left panel) and spindle assembly (right panel) by its allosteric activators phospho-Bora and Tpx2, respectively. In the G2 phase of the cell cycle, AURKA is maintained in a dephosphorylated state on the T-loop (T288) by counteracting phosphatases. During mitotic entry, Cyclin A-Cdk1 phosphorylates Bora, which then binds and activates AURKA. Phospho-Bora preferentially binds unphosphorylated AURKA. Activated by pBora, AURKA phosphorylates the Plk1 T-loop (T210) to activate the kinase and trigger mitotic entry. Then, during mitosis, Tpx2, through its first 43 amino acids, recruits and activates phosphorylated AURKA (pT288) at the microtubules to promote mitotic spindle assembly. (B) Domain architecture of *Homo sapiens* Bora (orange, left panel) and Tpx2 (blue, right panel) with the minimal fragments required for AURKA activation highlighted in dark orange (Bora$^{18-120}$) and dark blue (Tpx2$^{1-43}$). Important sequence elements identified in these regions are indicated, including the aromatic anchors in the M1 and M2 motifs of Bora and Tpx2, as well as the phosphorylatable PSP motif (M3) (green) of Bora, located just downstream of the M2 motif. Based on these sequences, we engineered a phosphorylated Tpx2-Bora fusion (phosphopeptide MK51, right panel) that reconstitutes the regulatory properties of both proteins. We also engineered a minimal Bora construct (phosphopeptide GK51, left panel) by directly fusing the putative M1 motif to the M2 and M3 motifs via a Glycine linker. (C) Competitive binding assay where fluorescein-labeled Tpx2$^{1-43}$ polypeptide, in complex with AURKA$^{T288V}$, is displaced by increasing amounts of competitor (cold Tpx2$^{1-43}$, pBora$^{1-224}$, pBora fusion, and pTxp2-Bora chimera) and monitored through fluorescence polarization signals. The displayed data points and the half-maximal inhibitory concentration (IC$_{50}$) value represent the average fluorescence polarization for each reaction condition, with standard deviations of the mean as error bars ($N = 3$ independent experiments, each performed with $n = 3$ independent experimental samples). ND not determined. (D) Activation of AURKA$^{T288V}$ ATPase activity by different activators as assessed using the ADP Glo assay with Kemptide substrate. Displayed data points and EC$_{50}$ values represent the average luminescence for each reaction condition with standard deviations of the mean as error bars ($N = 3$ independent experiments, each performed with $n = 3$ independent experiment samples). RLU relative light unit. Source data are available online for this figure.

phosphopeptide directly fusing the putative M1, M2, and M3 motifs of Bora (Fig. 1B, phosphopeptide GK51) failed to displace the interaction between FITC-Tpx2 and AURKA$^{T288V}$ (Fig. 1C). This observation suggests that while Tpx2 and Bora share some sequence similarities and can compete for AURKA binding (Tavernier et al, 2021b), their binding mode to AURKA likely differs to some extent.

Next, we tested the potentiating effect of the pTpx2-Bora chimera on AURKA$^{T288V}$ kinase activity toward the model Kemptide substrate by monitoring ATP consumption using the ADP-Glo assay (Zegzouti et al, 2009). This assay quantifies the amount of ADP produced during a kinase reaction (Fig. 1D). AURKA$^{T288V}$ in the presence of non-phosphorylated Bora displayed negligible activity, consistent with our previous observations (Tavernier et al, 2021b). However, the addition of the phosphorylated Tpx2-Bora chimera potentiated the activity of AURKA$^{T288V}$ multifold (Fig. 1D). In fact, the EC$_{50}$ value for potentiation by pTpx2-Bora was similar to phosphorylated pBora (18 nM vs. 71 nM, respectively). In contrast, the phosphopeptide, obtained by directly fusing the putative M1, M2, and M3 motifs of Bora, failed to activate AURKA$^{T288V}$, consistent with the binding experiments (Fig. 1D).

These observations support our hypothesis that the M2 motif of Bora positions the essential phospho-motif M3 in the active site of AURKA, enabling the phosphate on Bora S112 to replace the phosphate on T288 functionally.

### The pTpx2-Bora chimera potentiates AURKA$^{T288V}$ kinase activity but lacks the determinants needed to direct AURKA$^{T288V}$ activity toward the Plk1 kinase domain

As the pTpx2-Bora chimera potentiates AURKA$^{T288V}$ activity toward the artificial kemptide substrate, we next tested whether it also activates AURKA$^{T288V}$ toward Ser 10 of Histone H3 (Crosio et al, 2002) and the Plk1 kinase domain (kinase-dead (K82R) but phosphorylable on the T210 (hereafter Plk1$^{Kdom}$) (Macurek et al, 2008; Seki et al, 2008b; Tavernier et al, 2021b). The pTpx2-Bora chimera, similarly to pBora, readily potentiated AURKA$^{T288V}$ activity toward Histone H3 (Fig. 2A), in line with the results obtained with the Kemptide substrate (Fig. 1D). However, in marked contrast to pBora, the chimeric pTpx2-Bora construct

failed to stimulate AURKA$^{T288V}$ activity toward the Plk1 kinase domain (Fig. EV1A, compare pT210 phosphorylation between lanes 1–6 and 7–14). Hypothesizing that the Plk1 T-loop might not be accessible within the kinase domain, we asked whether the pTpx2-Bora chimera could activate AURKA$^{T288V}$ towards the isolated Plk1 T-loop (aa 190–225) fused to GST (hereafter GST$^{T-loop}$). At first glance, the pTpx2-Bora chimera appeared to activate AURKA$^{T288V}$ toward the isolated Plk1 T-loop similarly to pBora (Fig. EV1B). However, to further compare the efficiency of pBora and the pTpx2-Bora chimera in promoting Plk1 T-loop phosphorylation by AURKA$^{T288V}$, whether the T-loop is embedded in the Plk1 kinase domain or fused to GST, we ran the 30- and 60-min time points of the kinase assays side by side on the same SDS-PAGE. As shown in Fig. 2B, pBora and the pTpx2-Bora chimera activated AURKA$^{T288V}$ toward the GST$^{T-loop}$ but with poor efficacy compared to the effect observed with pBora on AURKA$^{T288V}$ when the Plk1 kinase domain was used as a substrate (Fig. 2B, compare lanes 2, 3 with lanes 8, 9, and 11, 12).

Taken together, these results suggest that pBora contains unique determinants, which are absent in the shorter pTpx2-Bora chimera, that allow unphosphorylated AURKA to phosphorylate the Plk1 T-loop, which is intrinsically a poor substrate for AURKA (Fig. 2C). These findings align with our previous observation that AURKA cannot phosphorylate Plk1 in the absence of pBora (Tavernier et al, 2021b).

### AlphaFold 3 structural modeling predicts that Bora$^{18-120}$ wraps around the N-terminal lobe of AURKA to position the phospho-Ser112 motif in the active site of AURKA

Next, we investigated how pBora activates unphosphorylated AURKA towards the Plk1 kinase domain. Compared to Tpx2, the minimal Bora fragment that activates unphosphorylated AURKA is considerably larger, spanning residues 18–120, with a conserved region between the M1 and M2 motifs unique to Bora, hereafter named the Bora-Specific Motif (BSM) (Fig. 3A) (Tavernier et al, 2021b). Understanding the binding mode of pBora to AURKA requires the structural characterization of the pBora$^{18-120}$•AURKA complex. However, despite several attempts, we did not obtain a crystal structure of the minimal phosphorylated Bora$^{18-120}$ fragment bound to AURKA$^{T288V}$, possibly due to the intrinsically disordered

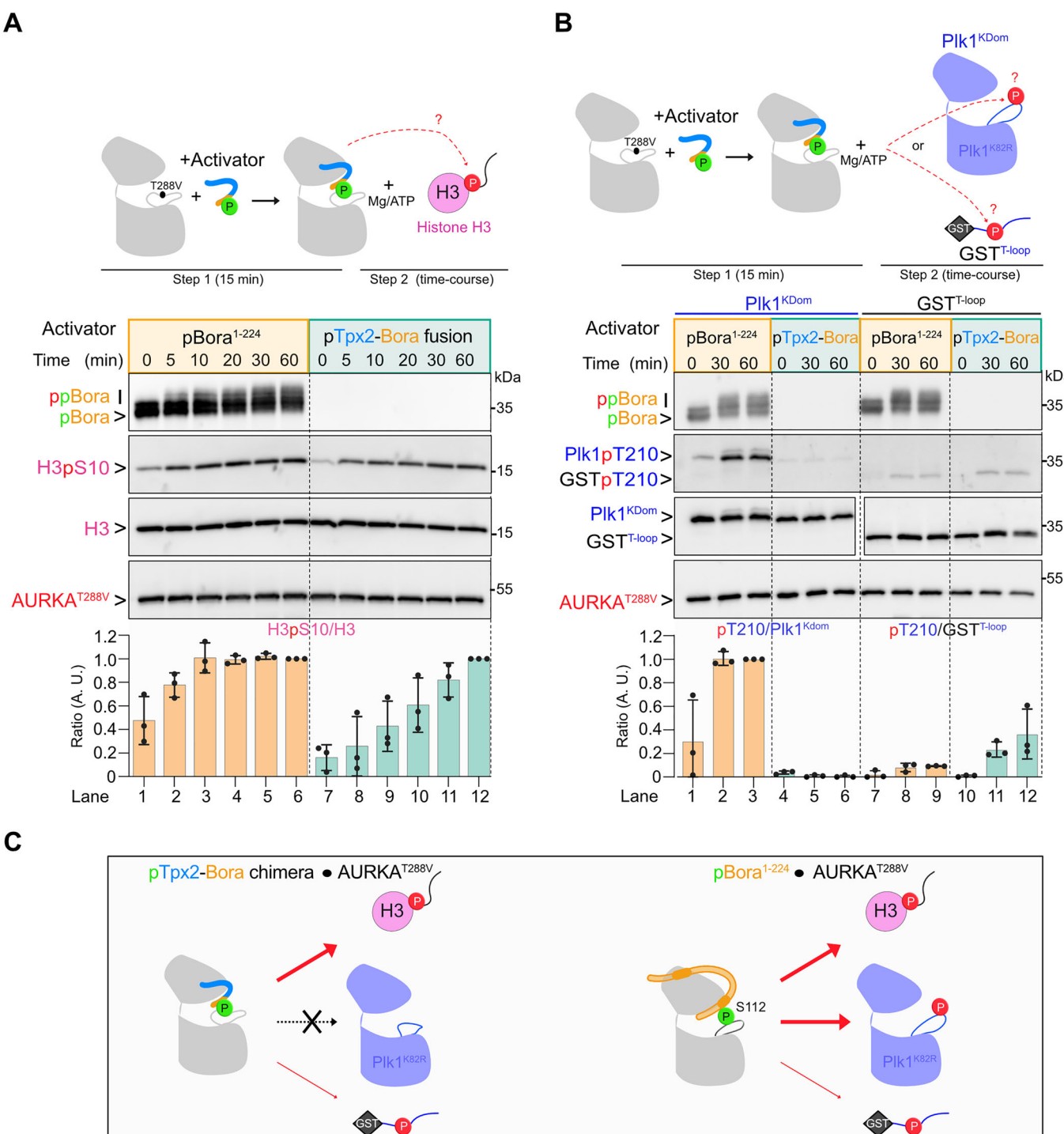

nature of Bora (Tavernier et al, 2021b). Therefore, we utilized the AlphaFold3 (AF3) algorithm to predict the interaction mode between phospho-Bora[18–120] and AURKA, comparing it to other allosteric activators in complex with AURKA, including Tpx2 (PDB: 1OL5 [https://doi.org/10.2210/pdb1OL5/pdb]) (Bayliss et al, 2003; McIntyre et al, 2017), Cep192 (PDB: 8PR7 [https://doi.org/10.2210/pdb8PR7/pdb]) (Holder et al, 2024), (PBD: 8GUW [https://doi.org/10.2210/pdb8GUW/pdb] (Park et al, 2023) and TACC3 [PDB: 5ODT https://doi.org/10.2210/

pdb5ODT/pdb]) (Burgess et al, 2018), for which 3D structures are available.

The crystal structure of the AURKA•Tpx2[1–43] complex showed that Tpx2[1–43] binds AURKA using two separate motifs, the M1 (residues 7–21) and the M2 motifs (residues 30–43), which are connected by a flexible linker. The M1 motif is extended and uses hydrophobic anchors (Y8, Y10, F16, F19) to bind two hydrophobic pockets on the N-lobe of AURKA called the Y and F pockets, respectively (Fig. 3B) (Bayliss et al, 2003; Eyers and Maller, 2004;

◄ **Figure 2.  The pTpx2-Bora chimera lacks determinants necessary to direct AURKA^T288V kinase activity toward the Plk1 kinase domain.**

(**A**) Schematic of the two-step in vitro reconstitution of Histone H3 phosphorylation on S10 by AURKA^T288V and pBora^1-224 or the pTpx2-Bora chimera. In step 1, Bora^1-224 phosphorylated by ERK (noted pBora) or the pTpx2-Bora chimera is incubated with AURKA^T288V for 15 min. In step 2, the reaction mix from step 1 is incubated in the presence of Mg/ATP with Histone H3 for 5, 10, 20, 30, and 60 min. Samples were then analyzed by Western blot. Blots were probed with antibodies to Bora, phospho-S10 Histone H3, or pan Histone H3, and AURKA (from top to bottom). Note that during step 2, pBora^1-224 itself is phosphorylated by activated AURKA^T288V during the reaction, which is manifested by a mobility shift in SDS-PAGE, as reported previously (Tavernier et al, 2021b). ppBora thus denotes Bora phosphorylated by ERK during step 1 and by activated AURKA during step 2 in this and other Figures. The graph presents the normalized quantification of pS10 Histone H3 signal over Histone H3 from $n = 3$ independent experiments. Error bars display the standard deviation. (**B**) Side-by-side comparison of the two-step in vitro reconstitution of T-loop phosphorylation on T210 of Plk1 (pT210) by AURKA^T288V and pBora^1-224 or the pTpx2-Bora chimera when the Plk1 kinase domain or only the isolated T-loop is used as a substrate. In step 1, pBora^1-224 or the pTpx2-Bora chimera is incubated with AURKA^T288V for 15 min. In step 2, the reaction mix in step 1 is incubated in the presence of Mg/ATP with Plk1 kinase domain catalytically dead mutant (Plk1^K82R) or the isolated Plk1 T-loop for 30 and 60 min. Samples were then analyzed by Western blot. Blots were probed with antibodies to Bora, phospho-T210 Plk1, pan Plk1, or GST, and AURKA (from top to bottom). The graph presents the normalized quantification of pT210 Plk1 signal over either Plk1^KDom or GST-Plk1^T-loop from $n = 3$ independent experiments. Error bars display the standard deviation. (**C**) Schematics summarizing the main results. The pTpx2-Bora chimera and pBora^1-224 can similarly activate AURKA^T288V toward Histone H3 (red arrows). However, in sharp contrast to pBora^1-224, the pTpx2-Bora chimera is unable to activate AURKA^T288V toward the Plk1 T-loop embedded in the kinase domain (crossed arrow). Both the pTpx2-Bora chimera and pBora^1-224 can stimulate AURKA^T288V activity towards the isolated T-loop, but with poor efficacy (thin red arrows). Source data are available online for this figure.

McIntyre et al, 2017). Of note, Cep192, which promotes AURKA autophosphorylation at centrosomes (Joukov et al, 2010), also engages the F-pocket of AURKA via two aromatic anchors (Y487 and F490, equivalent to F16 and F19 in Tpx2) (Holder et al, 2024).

In contrast to the extended M1 motif, the Tpx2 M2 motif (residues 30–43) adopts an alpha-helical conformation that binds between the N- and C-terminal lobes of AURKA, adjacent to the αC helix and to the activation segment via aromatic residues, W34 and F35, which engage the W pocket of AURKA (Bayliss et al, 2003; McIntyre et al, 2017) (Fig. 3B). Tpx2 preferentially binds phosphorylated AURKA and induces conformational changes within the catalytic infrastructure and the T-loop, such that the phosphorylated T288 residue contacts several arginine residues, thereby stabilizing the activation loop (Fig. 3C) (Bayliss et al, 2003; Zorba et al, 2014; McIntyre et al, 2017).

In five out of five generated models, AF3 predicts a different mode of interaction between Bora^18-120 and AURKA (Figs. 3D, EV2A, EV3A, and EV4). In contrast to Tpx2^1-43, which binds along the same face of the kinase N-terminal lobe (Fig. 3B), Bora^18-120 wraps circuitously around the N-terminal lobe of AURKA (Figs. 3D, EV2A, EV3A, and EV4). The putative M1 motif of Bora binds the same face of AURKA as observed for the M1 motif of Tpx2. However, the M1 region in Bora is much larger than initially anticipated based on structural alignments (Tavernier et al, 2021b) (extending from residues L22 to F45 rather than 22 to 35) and is oriented in the opposite direction compared to that observed for Tpx2^1-43 (Fig. 3B–D). In Tpx2, the beginning of the M1 motif (Y8, Y10) binds the Y pocket, and the end of the motif (F16, F19) binds the F pocket of AURKA (Figs. 3B and EV3A). However, in Bora, it is the opposite: the beginning of the M1 motif (F25) binds the F pocket, while the end of the motif (V44, F45) binds the Y pocket (Fig. 3D). These two reasons, and possibly others, likely explain why the Bora fusion is not efficient at activating AURKA^T288V (Fig. 1D).

Following the Bora M1 motif, the evolutionarily conserved Bora-specific motif located between T52 and I76 (Fig. 3A) binds at the back of AURKA N-terminal lobe (Figs. 3D–F and EV3A). In particular, the conserved I60 and I71 residues are predicted to dock on two pockets at the back of the AURKA N-terminal domain (Figs. 3F and EV3A). Noteworthy, the other allosteric AURKA activators TACC3 and Cep192 also bind these two pockets of AURKA (Burgess et al, 2018; Park et al, 2023; Holder et al, 2024)

(Fig. EV2B,C). While I60 and I71 residues anchor Bora to AURKA, other highly conserved aromatic or hydrophobic residues, such as F56, W58, and I66, are conspicuously solvent-exposed in this conserved region (Figs. 3F, red arrows and EV3A).

After Bora^18-120 circles the N-terminal lobe of the kinase, the α-helical M2 motif is similarly positioned to that observed in Tpx2^1-43. This segment contains conserved aromatic residues F103 and F104, equivalent to W34 and F35 in Tpx2, and they dock in the hydrophobic W pocket of AURKA while packing against the αC helix (Fig. 3D). Next, the Bora M2 motif positions the pS112 M3 motif in the AURKA active site, allowing it to substitute in trans for the phosphate typically provided by autophosphorylation. The phospho-site is predicted to interact with the "arginine pocket" of AURKA, composed of R255, R180, and R286 (Fig. 3E), which has been previously shown to bind the phosphate on the T288 residue of the T-loop, particularly when phospho-AURKA is bound to Tpx2^1-43 (Fig. 3C) (Bayliss et al, 2003). The model also predicts that AURKA bound to pBora exhibits characteristics of an active kinase with the "DFG motif in" and the R-spine assembled (Fig. EV2D).

Overall, this binding model is consistent with previous observations that Bora and Tpx2 compete for AURKA binding (Chan et al, 2008; Tavernier et al, 2021b), that the "Arginine pocket" is required for high-affinity binding of pBora to AURKA^T288V (Tavernier et al, 2021b), and our present results showing that providing a phosphosite to Tpx2 is sufficient to increase its affinity for unphosphorylated AURKA and to potentiate AURKA kinase activity (Fig. 1C,D).

## A streamlined method for high-throughput screening of the critical determinants of phospho-Bora and AURKA-dependent Plk1 T-loop phosphorylation in *E. coli*

To test our AlphaFold model, we generated Bora variants mutated on evolutionarily conserved residues and residues predicted to contact AURKA (Fig. 3).

To assess Bora's function, we previously used a two-step kinase assay with purified proteins in which, in the first step, we treated Bora with active Cyclin A-Cdk1 or the Mitogen-Activated Protein (MAP)-kinase ERK, to phosphorylate the essential M3 motif pS112 in Bora. In a second step, we then added AURKA^T288V and a kinase-dead mutant of the Plk1 kinase domain (K82R), which can still be phosphorylated on its T-loop as substrate. Western blotting of the

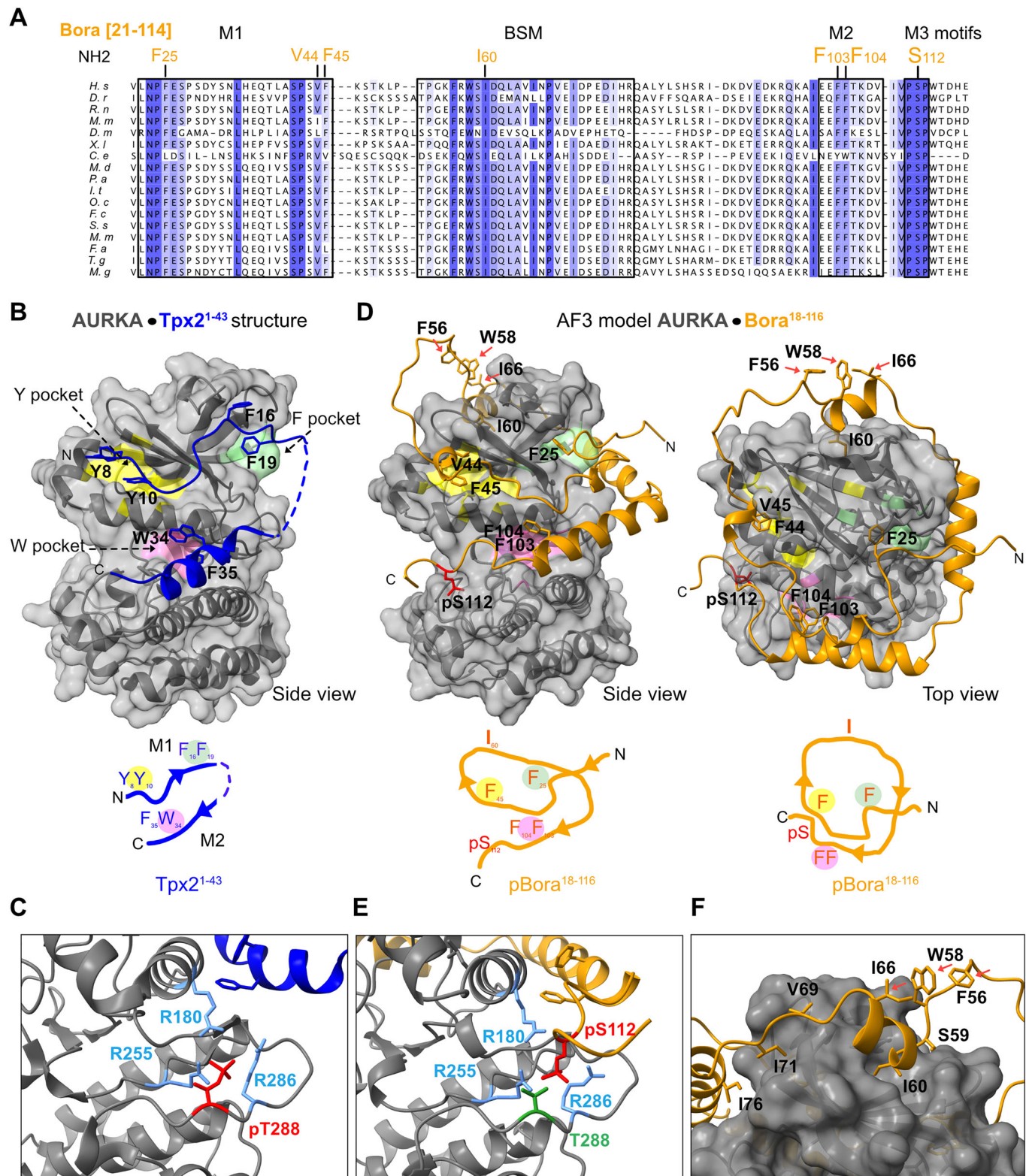

**Figure (A)** Bora [21–114] with M1, BSM, M2, M3 motifs and sequence alignment (H. s, D. r, R. n, M. m, D. m, X. l, C. e, M. d, P. a, I. t, O. c, F. c, S. s, M. m, F. a, T. g, M. g).

**(B)** AURKA•Tpx2^1-43 structure — Side view; Tpx2^1-43.

**(C)** R180, R255, R286, pT288.

**(D)** AF3 model AURKA•Bora^18-116 — Side view and Top view; pBora^18-116.

**(E)** R180, R255, pS112, R286, T288.

**(F)** V69, I66, W58, F56, S59, I71, I60, I76.

reaction allowed the monitoring of Plk1 T210 phosphorylation as a direct readout of AURKA^T288V activity stimulated by pBora (Tavernier et al, 2015; Thomas et al, 2016; Tavernier et al, 2021b).

However, testing multiple Bora variants by this approach would require systematic purification and pre-phosphorylation for each.

To circumvent this bottleneck and to rapidly screen the activity of the phosphorylated Bora variants, we developed a streamlined method that we named MITOKINAC, for MITOtic KINases Activation in E. coli (Fig. 4A). This method reconstitutes pBora-dependent activation of non-phosphorylated AURKA

**Figure 3.  pBora¹⁸⁻¹²⁰ binding to AURKA predicted by AlphaFold compared to the structure of the Tpx2¹⁻⁴³ •AURKA complex.**

(A) Multiple protein sequence alignments of the minimally active Bora fragment from different species, including *Homo sapiens (H. s)* [accession number: Q6PGQ7], *Danio Rerio (D. r)* [Q5U3U6], *Rattus norvegicus (R. n)* [Q5M864], *Mus musculus (M. m)* [Q8BS90], *Drosophila melanogaster (D. m)* [Q9VVR2], *Xenopus laevis (X. l)* [Q6DJL7], *Caenorhabditis elegans (C. e)* [H2FLK2], *Monodelphis domestica (M. d)* [F6XNR5], *Pongo abelii (P. a)* [H2NK15], *Ictidomys tridecemlinea (I. t)* [I3MI81], *Oryctolagus cuniculus (O. c)* [G1TPL0], *Felis catus (F. c)* [M3WCM8], *Sus scrofa (S. s)* [I3LG24], *Macaca mulatta (M. m)* [F7HH76], *Ficedula albicollis (F. a)* [U3JE63], *Taeniopygia guttata (T. g)* [H0ZQH5], *Meleagris gallopavo (M. g)* [G1NQ02]. Sequences were aligned using ClustalW and visualized with Jalview. The darker blue color indicates higher similarity and identity. The M1, M2, M3, and Bora-specific motifs (BSM) are surrounded by a rectangle. (B) Structure of Tpx2¹⁻⁴³ (blue) bound to AURKA (gray) (Bayliss et al, 2003). The critical aromatic residues of Tpx2¹⁻⁴³ engaging the Y pocket (yellow), the F pocket (green), and the W pocket (magenta) are indicated. The schematic at the bottom shows the aromatic residues in the M1 and M2 motifs and the relative orientation of these two motifs when bound to AURKA. The dashed line shows the disordered linker region between the two motifs. The arrowheads show the orientation (N-terminal to C-terminal) of the Tpx2 fragment when bound to AURKA. (C) Zoom in view of the activation segment region of AURKA (gray) bound to Tpx2¹⁻⁴³ (blue). The phosphorylated residue of the T-loop (pT288, in green) is coordinated by direct contact with R180, R255, and R286 (light blue). (D) Side and top view of the AlphaFold3 model of phospho-Bora¹⁸⁻¹¹⁶ (orange) binding to the AURKA kinase domain (gray). AURKA is oriented as in (B) for direct comparison. Aromatic and hydrophobic residues of Bora binding the Y (yellow), F (green), and W (magenta) pockets of AURKA are indicated, as well as the solvent-exposed aromatic and hydrophobic residues of Bora at the back of AURKA (red arrows). (E) Zoom in view of the activation segment region of AURKA (gray) bound to pBora (orange) in an AlphaFold3 model. The helical M2 motif, anchored via F103 and F104 residues binding to the W pocket, projects the essential phosphosite S112 (in red) into the arginine pockets composed of R180, R255, and R286 (light blue). (F) Rear view of the AlphaFold3 model of phospho-Bora¹⁸⁻¹¹⁶ (orange) binding to the AURKA kinase domain (gray), showing the hydrophobic residues I60, I71, and I76 binding AURKA pockets while F56, W58, and I66 residues are solvent exposed (red arrows).

(AURKA^T288V) and then Plk1 T-loop phosphorylation directly in *E. coli*, without the need to purify the proteins, as summarized in Fig. 4B. Briefly, we co-transformed *E. coli* BL21 with three plasmids expressing (i) the active ERK kinase (or ERK kinase-dead K54R as control), which replaces Cyclin A-Cdk1 as a Proline-directed kinase to phosphorylate the M3 motif of Bora, (ii) Bora¹⁻²²⁴ WT or S112A as control, and AURKA^T288V expressed from the same plasmid, and (iii) the Plk1 kinase domain as a substrate (kinase-dead K82R but phosphorylable on the T-loop (T210)) (Fig. 4A). After protein induction, Plk1 T-loop phosphorylation was monitored by Western blot directly on total bacterial extracts using a specific antibody directed against the phospho-Threonine 210 of Plk1 (anti-pT210).

In this assay, wild-type pBora stimulated AURKA^T288V activity toward the Plk1 T-loop (Fig. 4C, lane 3). Importantly, Plk1 T-loop phosphorylation was strictly dependent on the presence of Bora phosphorylated on the M3 motif pS112 by active ERK (Fig. 4C, compare lane 3 with lanes 1, 2, and 4). Indeed, no Plk1 T-loop phosphorylation was detected in the absence of Bora (Fig. 4C, lane 1), or when ERK was replaced with a kinase-dead variant (Fig. 4C, lane 2), or when Bora was mutated on the critical S112 residue (Fig. 4C, lane 4).

The system's versatility enables rapid screening for Bora, AURKA, and Plk1 variants. For instance, we substituted the Plk1 kinase domain with the isolated Plk1 T-loop fused to GST as an alternative AURKA substrate and observed that pBora enhances AURKA^T288V activation towards this target, albeit with relatively poor efficacy since the GST-T-loop is significantly overexpressed in *E. coli*, compared to the Plk1 kinase domain (Fig. 4D). Again, this reaction was strictly dependent on the presence of Bora phosphorylated on the M3 motif pS112 by active ERK (Fig. 4D, compare lane 3 with lanes 2 and 4).

To corroborate these observations and further exploit MITO-KINAC, we used Genetic Code Expansion (Rogerson et al, 2015; Qin and Liu, 2022; Zhu et al, 2022) to insert a phosphoserine directly at position 112 of Bora during translation in *E. coli*, thereby bypassing the need for priming phosphorylation by ERK (Tavernier et al, 2021b) (Fig. 4E). As shown in Fig. 4F, directly inserting a phosphoserine specifically at position 112 of Bora¹⁻²²⁴ during translation promotes AURKA^T288V activation towards the Plk1 T-loop. In line with previous results, no activation was observed

using Bora¹⁻²²⁴ non-phosphorylated, demonstrating that phosphorylation of this S112 position is necessary and sufficient for Bora's function.

Overall, this method is ideally suited for rapidly screening critical determinants required for the activation of unphosphorylated AURKA by phospho-Bora towards the Plk1 kinase domain. More generally, this method can be implemented to study any kinase activation cascade or kinase-substrate relationship (Roumbo et al, 2025).

## The extended Bora M1 motif binds the hydrophobic Y and F pockets of the N-terminal lobe of AURKA via two aromatic plugs

We next used MITOKINAC to screen 39 mutant variants (affecting 35 unique amino acids) within the Bora¹⁸⁻¹²⁰ fragment for their ability to promote AURKA^T288V phosphorylation of the Plk1 kinase domain. Although we expressed the first 224 amino acids of Bora in *E. coli*, we mutagenized only the region between amino acids 18 and 120 because it is the only part of Bora predicted by AF3 to bind AURKA. We systematically substituted hydrophobic residues with acidic residues and charged residues with residues of opposite charge.

Based primarily on sequence alignment between Tpx2 and Bora, we had previously predicted that residues F25 and Y31 in Bora might fulfill a similar function to the aromatic plugs of the Tpx2 M1 motif in binding AURKA. Indeed, the double Bora F25D Y31D mutant was inactive and failed to displace the Tpx2-AURKA interaction (Tavernier et al, 2021b). Here, we screened several additional positions within the extended M1 motif of Bora (amino acids 22-48) (Fig. 5A–C) and found that F25D (Fig. 5B, lane 5) but not Y31D (lane 9) and the double V44D, F45D (lane 20) mutations affected Bora's ability to activate AURKA^T288V towards the Plk1 T-loop. Substitution of F25 by Aspartic acid eliminated Bora's function, whereas the single V44D and F45D substitutions had no detectable effect (Fig. 5B, compare lanes 5, 11, 12, and 20). We only observed reduced Plk1 T-loop phosphorylation by AURKA^T288V when both V44 and F45 were substituted by Aspartic acid. We confirmed these results using purified proteins and in vitro kinase assays (Fig. 5D, compare lanes 2, 4, 6, and 8).

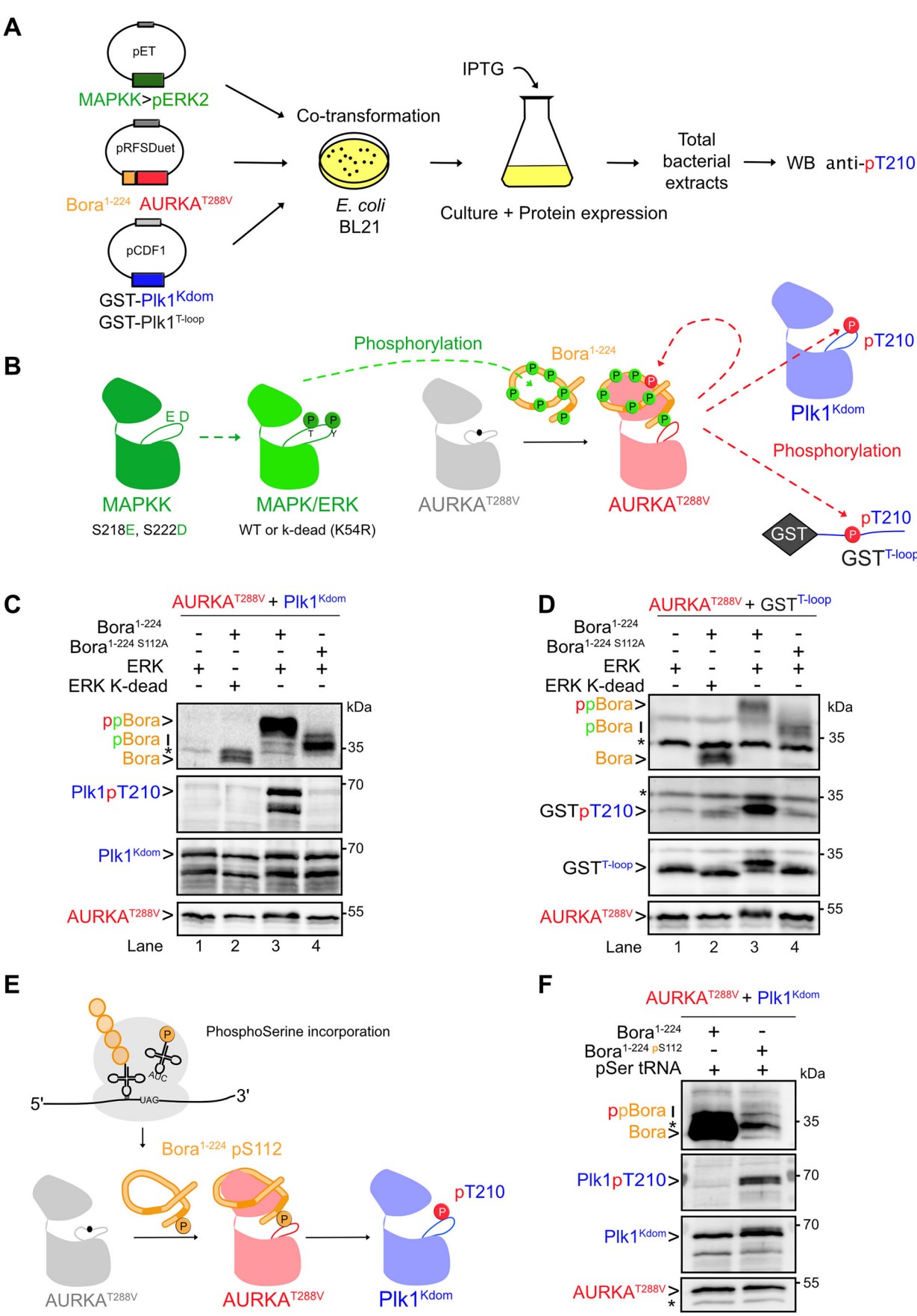

◄ **Figure 4.  Mitotic kinases activation in *E. coli* (MITOKINAC): a streamlined method that reconstitutes phospho-Bora-dependent AURKA^{T288V} activation toward the Plk1 T-loop in *E. coli*.**

(A) Schematic of MITOKINAC: *E. coli* BL21 are transformed with three plasmids containing different replication origins, antibiotic resistance, and expressing (i) the active ERK kinase (MAPKK constitutively activated S118E, S222D, which phosphorylates and activates the MAPK ERK), (ii) Bora^{1-224} and AURKA^{T288V} expressed from the same plasmid, and (iii) the Plk1 kinase domain (kinase-dead K82R but phosphorylable on the T-loop (T210)) or the isolated T-loop fused to GST as substrates. After co-transformation, culture, and protein induction with IPTG, Plk1 T-loop phosphorylation (T210) is monitored by Western blot directly on total bacterial extracts.
(B) Schematic of the mitotic ERK kinase cascade reconstituted in *E. coli*. The activated ERK kinase phosphorylates Bora at multiple sites, including S112 (M3 motif), which binds and activates non-phosphorylated AURKA (AURKA^{T288V}) towards the Plk1 kinase domain. (C, D) Western blot analysis of total bacterial extracts reconstituting pBora^{1-224} and AURKA^{T288V}-dependent T-loop (T210) phosphorylation of the Plk1 kinase domain (Plk1^{Kdom}) (C) or the isolated T-loop fused to GST (GST^{T-loop}) (D). Blots were probed with antibodies to Bora, phospho-T210 Plk1, or pan Plk1 (C) or GST (D), and AURKA (from top to bottom). ERK K-dead: kinase dead. Asterisk denotes a nonspecific band.
(E) Schematic of MITOKINAC using Genetic Code Expansion to produce Bora^{1-224,} uniquely phosphorylated at serine 112, thus bypassing the need for a priming phosphorylation by ERK. The schematic illustrates the translation of Bora^{1-224} (orange) by ribosomes (gray) in *E. coli*, which inserts a phosphoserine (orange) at the amber codon UAG. When produced and phosphorylated at S112, pBora^{1-224} binds to and activates AURKA, which subsequently phosphorylates the Plk1 T-loop (T210).
(F) Western blot analysis of total bacterial extracts reconstituting pBora and AURKA^{T288V}-dependent T-loop (T210) phosphorylation of the Plk1 kinase domain (Plk1^{Kdom}) using Genetic code expansion. Bacteria were transformed with a plasmid expressing wild-type Bora^{1-224} (lane 1) or Bora^{1-224} harboring the amber codon TAG at position S112 (lane 2). Blots were probed with antibodies to Bora, phospho-T210 Plk1, GST, and AURKA (from top to bottom). ERK K-dead: kinase dead. Source data are available online for this figure.

These observations are entirely consistent with the AlphaFold 3 model, which predicts that Bora's extended M1 motif binds AURKA via the two hydrophobic plugs, F25 and V44-F45, which engage the hydrophobic F and Y pockets of AURKA, respectively, in an opposite orientation compared to Tpx2 (Fig. 3B–D), with F25 playing a more significant role. These results also align with recent data showing that a 20-amino acid Bora fragment (aa 18–38) encompassing F25, which is part of an "NPF" motif (aa 23–25), is sufficient for AURKA binding (Keskitalo et al, 2025).

## The M2 motif positions the essential phospho-motif in the active site of AURKA via hydrophobic anchors

Next, we focused on the M2 motif, located upstream of the essential phosphoserine S112 (M3 motif). Based on AF3, the α-helical M2 motif is similarly positioned to the M2 motif of Tpx2^{1-43}. This segment contains the conserved aromatic residues F103 and F104, which are equivalent to W34 and F35 in Tpx2, docking on the hydrophobic W pocket of AURKA (Fig. 3B–D and EV3A). We previously reported that the double Bora F103D F104D mutant phosphorylated on the M3 motif does not bind and activate AURKA^{T288V} (Tavernier et al, 2021b). Here, using MITOKINAC and then in vitro kinase assays, we found that individually mutating F103 and F104 residues is sufficient to abrogate Bora's function, highlighting the critical role of these two residues (Fig. 5E–H). Mutation of the I109 and V110 residues also affected Bora's function (Fig. 5E,F), most likely by preventing the proper positioning of the phosphoserine 112 in the active site of AURKA or, in the case of V110R, by inhibiting S112 phosphorylation by ERK. Accordingly, the Bora V110R variant was not upshifted on SDS-PAGE, as seen with Bora S112A (Fig. 5F), suggesting that ERK did not efficiently phosphorylate this variant.

These results identify the key Bora residues in the M1 and M2 motifs that are critically required for AURKA activation.

## The Bora-specific motif (aa 52–78) dictates AURKA's specificity toward the Plk1 kinase domain

We next screened the highly conserved region (amino acids 50–78) between the M1 and M2-M3 motifs, which is specific to Bora

(Fig. 6A). Using MITOKINAC, we found that mutation of several conserved residues, including F56, W58, S59, I60, I66, P68, V69, I71, I76, and Q79 affected Bora's ability to promote AURKA^{T288V} phosphorylation of the Plk1 T-loop embedded in the kinase domain (Fig. 6A,B). Intriguingly, several of these mutations had no effect when only the isolated Plk1 T-loop fused to GST was used as a substrate (Fig. EV1C). Indeed, in contrast to Bora F25D, which is defective for promoting AURKA^{T288V} phosphorylation of both the Plk1 kinase domain and isolated Plk1 T-loop substrates, the Bora variants, F56D, W58D, S59R, I60D, Q62R, and I66D, retained the ability to promote AURKA^{T288V} phosphorylation of the isolated Plk1 T-loop in *E. coli* (Figs. EV1C and EV3B).

Based on AF3 structural modeling, this region is predicted to bind AURKA at the back of the molecule (Fig. 3D–F). The serine 59, which is part of an AURKA consensus site (RWS) (Deretic et al, 2019) and previously shown to be phosphorylated by AURKA itself (Lössl et al, 2016), is the first residue of a predicted α-helix, also known as the N-cap residue. Substitution of this serine 59 by an Arginine abrogated Bora's ability to promote AURKA^{T288V} phosphorylation of the Plk1 kinase domain. However, substituting this serine with a Glutamic acid as the helix's N-cap residue had no discernible effect (Fig. 6B).

Next to serine 59, Isoleucine 60 at the beginning of the Bora-specific motif (Fig. 6A) and Isoleucine 71 at the end anchor the motif into two pockets of AURKA that TACC3 and CEP192 also use to bind AURKA (Fig. EV2B,C) (Burgess et al, 2018; Park et al, 2023; Holder et al, 2024). However, other conserved residues essential for Bora's function, including F56, W58, and I66, are solvent-exposed and are not predicted to bind AURKA (Figs. 3D, EV2A, and EV3A). We reasoned that some of these residues, absent from the smaller pTpx2-Bora chimera (Fig. 1B), might confer AURKA specificity to the Plk1 kinase domain.

To test this hypothesis, we purified these Bora variants and then compared their ability to activate AURKA^{T288V} toward the Plk1 kinase domain, the isolated Plk1 T-loop, and Histone H3 using in vitro kinase assays. Consistent with the results obtained using MITOKINAC, all these Bora variants failed to promote AURKA^{T288V} phosphorylation of the Plk1 kinase domain (Fig. 6C,D), but retained their ability to promote AURKA^{T288V} phosphorylation of the isolated Plk1 T-loop (Fig. EV1C) or Histone H3 (Fig. 6E,F).

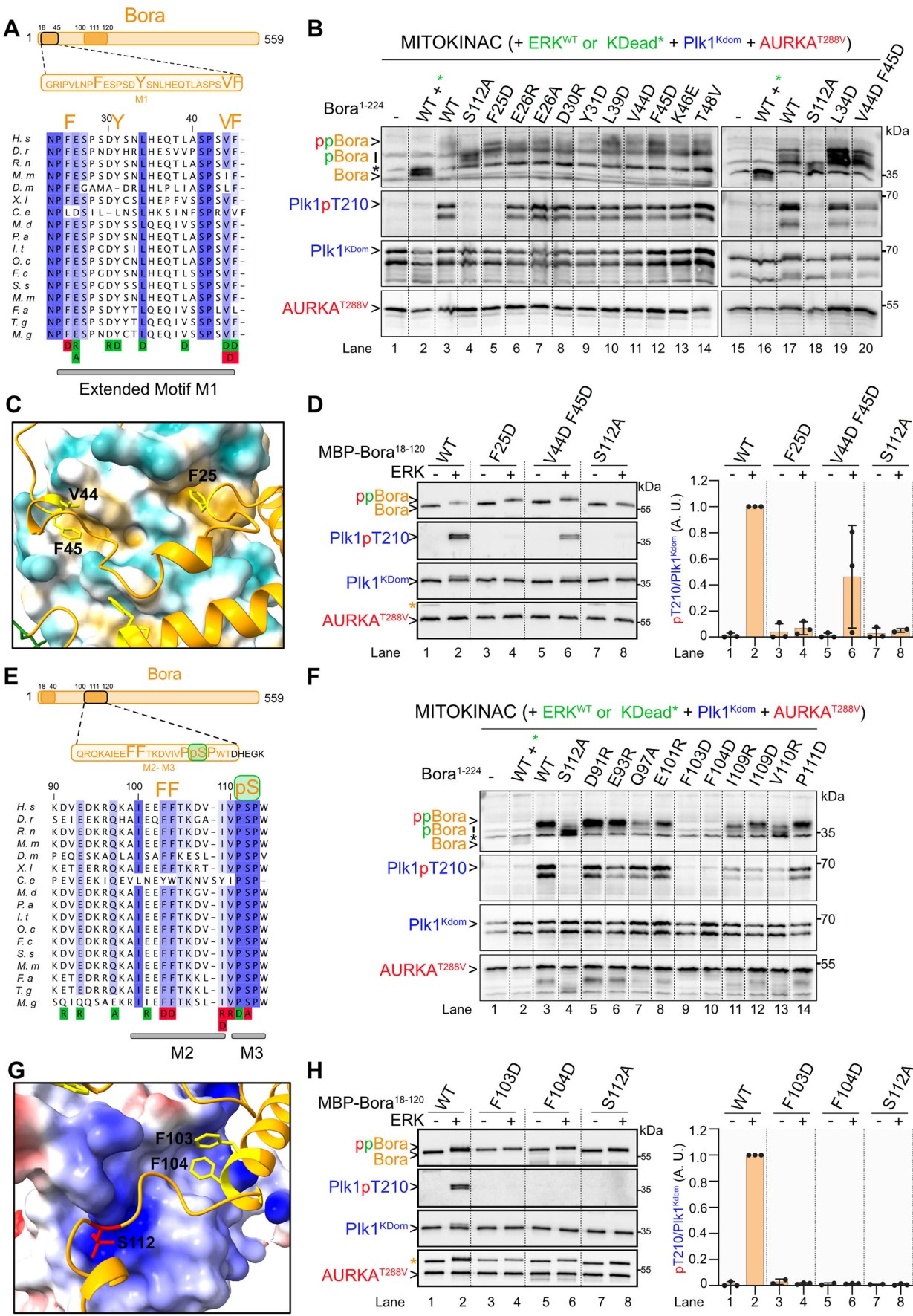

◄ **Figure 5.  Bora¹⁸⁻¹²⁰ binds AURKA through hydrophobic anchors to position the essential phosphosite (pS112) in the AURKA active site.**

(A) Multiple protein sequence alignments of the M1 motif of Bora. Identical residues are in dark blue. The mutated residues and the nature of the substitutions are indicated at the bottom of the alignment. The impact of the mutation on Bora's function is indicated by the rectangle's color around the amino acids. Green rectangles mean no effect, while red rectangles indicate a loss of function. (B) Western blot analysis of bacterial extracts reconstituting the mitotic kinase cascade in *E. coli* BL21, resulting in T-loop (T210) phosphorylation of the Plk1 kinase domain (Plk1$^{Kdom}$) using wild-type and Bora mutants of the M1 motif. Blots were probed with antibodies to Bora, phospho-T210 Plk1, or pan Plk1, and AURKA (from top to bottom). ERK K-dead: kinase dead. (C) View of the extended M1 motif of Bora bound to AURKA predicted by AlphaFold3. The M1 motif of Bora is anchored at one end by F25 binding to the F pocket of and V44, and F45 binding to the Y pocket of AURKA, respectively. (D) Western blot analysis of 2-step kinase reactions carried out with MBP-Bora¹⁸⁻¹²⁰ wild-type or mutant phosphorylated ($+$) or not ($-$) by the ERK kinase (step 1) in the presence of Plk1$^{KDom}$ (substrate) and AURKA$^{T288V}$ (step 2). Blots were probed with antibodies to Bora, phosphoT210 Plk1, or pan Plk1, and AURKA, as indicated (from top to bottom). The graph presents the normalized quantification of pT210 Plk1 signal over Plk1$^{KDom}$ from $n = 3$ independent experiments. Error bars display the standard deviation. (E) Multiple protein sequence alignments of the M2 and M3 motifs of Bora. Identical residues are shown in dark blue. The mutated residues and the nature of the substitutions are indicated at the bottom of the alignment. The impact of the mutation on Bora's function is indicated by the rectangle's color around the amino acids. Green rectangles indicate no effect, while red rectangles indicate a loss of function. (F) Western blot analysis of bacterial extracts reconstituting the mitotic kinase cascade in *E. coli* BL21, resulting in T-loop (T210) phosphorylation of the Plk1 kinase domain (Plk1$^{Kdom}$) using wild-type and Bora mutants of the M2 and M3 motifs. Blots were probed with antibodies to Bora, phospho-T210 Plk1, or pan Plk1, and AURKA (from top to bottom). ERK K-dead: kinase dead. Note that Bora F103D and F104D mutants are expressed in *E. coli* (H) but not detected by our anti-Bora antibody. (G) View of the M2 and M3 motifs of Bora bound to AURKA predicted by AF3. AURKA is shown as a surface colored according to electrostatic potential, with blue indicating positive and red indicating negative. Bora is shown as an orange ribbon. Bora residues F103 and F104 bind to the W pocket of AURKA, while pS112 engages an electropositive pocket. (H) Western blot analysis of 2-step kinase reactions carried out with MBP-Bora¹⁸⁻¹²⁰ wild-type or mutant phosphorylated ($+$) or not ($-$) by the ERK kinase (step 1) in the presence of Plk1$^{KDom}$ (substrate) and AURKA$^{T288V}$ (step 2). Blots were probed with antibodies to MBP, phosphoT210 Plk1, or pan Plk1, and AURKA, as indicated (from top to bottom). ERK K-dead kinase dead. As membranes were sequentially revealed with different antibodies, the orange asterisk indicates the signal revealed by the anti-Bora antibody. The graph presents the normalized quantification of pT210 Plk1 signal over Plk1$^{KDom}$ from $n = 3$ independent experiments. Error bars display the standard deviation. Source data are available online for this figure.

At face value, these observations suggest that this evolutionarily conserved region of Bora is not essential for AURKA$^{T288V}$ kinase activation per se but is required to direct AURKA specificity towards the Plk1 kinase domain.

## Bora docking to the Plk1 kinase domain promotes T-loop phosphorylation by AURKA

Our observations thus suggest that pBora not only activates non-phosphorylated AURKA by bypassing the need for T-loop autophosphorylation, but it also allows AURKA to selectively phosphorylate the T-loop of Plk1 in the context of an intact kinase domain, possibly via the conserved and possibly solvent-exposed residues of the Bora-specific motif. To investigate this possibility further, we interrogated AlphaFold3 for potential interactions between Bora and the Plk1 kinase domain by modeling the AURKA•Plk1 kinase domain•Bora¹⁸⁻¹²⁰ trimeric complex.

Although Bora phosphorylated on its Polo-docking site (S252) by Cyclin-Cdk robustly binds the Plk1 PBD (Polo-box domain) (Chan et al, 2008; Seki et al, 2008a), we never detected a stable interaction between Bora and the Plk1 kinase domain by pull-down experiments or NMR studies (Tavernier et al, 2021b). However, AF3 predicts an interaction between the Bora-specific motif and the αC helix of the Plk1 kinase domain (Fig. 7A,B). The total interaction area is 554 Å². It involves contacts between several residues of the Bora-specific motif, including F56, W58, I66 (Fig. 7C), and R48, R95, R106A, E121A of the Plk1 kinase domain (Fig. 7C).

To assess the significance of the predicted interaction between pBora and the Plk1 kinase domain, we purified mutants of the Plk1 kinase domain and tested them as substrates of pBora•AURKA$^{T288V}$ using in vitro kinase assays. Notably, mutations in the Plk1 residues R48, R95, R106, and E121 reduced the pBora and AURKA-dependent phosphorylation of the Plk1 T-loop (Fig. 7D,E).

Together, these results strongly suggest that transient interaction between Bora and the Plk1 kinase domain is required for the ability of AURKA to efficiently phosphorylate the Plk1 T-loop.

## The Bora-specific motif is critically required for mitotic entry in Xenopus egg extracts

Having determined that the Bora-specific motif is essential for Bora's function in vitro, we next investigated the relevance of this determinant in a physiological context. To this end, we tested Bora's function during mitotic entry in Xenopus egg extracts, whereby interphase egg extracts are forced to enter into mitosis using the hyperactive Gwl$^{K72M}$ kinase (Vigneron et al, 2009, 2018). Indeed, adding a recombinant hyperactive human Gwl kinase (Gwl$^{K72M}$) to interphase extracts partially inhibits the phosphatase PP2A-B55, stabilizing substrate phosphorylation by basal Cyclin-Cdk activity, which forces entry into mitosis (Vigneron et al, 2009). However, prior immunodepletion of Bora from these extracts prevents mitotic entry (Vigneron et al, 2018; Tavernier et al, 2021b). Under these conditions, Plx1 (Xenopus Plk1) activation by AURKA and the dephosphorylation of the inhibitory phosphorylation on Cdk (specifically on Tyr15) are inhibited. This mitotic entry defect can be fully rescued by adding wild-type recombinant Bora (Vigneron et al, 2018; Tavernier et al, 2021b) (Fig. 8A). We therefore tested whether the minimal Bora fragment 18–120 that supports Plk1 phosphorylation by AURKA in vitro could promote mitotic entry in interphase extracts depleted of Bora. We again use the MAPK kinase ERK to pre-phosphorylate Bora on S/T-P sites, as we did in our in vitro reactions, to circumvent inefficient phosphorylation of Bora fragments by Cyclin A-Cdk1 in Xenopus egg extracts. As shown in Fig. 8B, the wild-type Bora¹⁸⁻¹²⁰ fragment, pre-phosphorylated by ERK rescued mitotic entry as revealed by the accumulation of mitotic markers, including the phosphorylation of the Gwl kinase, the phosphorylation of Plx1 on the T-loop (T201), and the dephosphorylation of the inhibitory site on the Tyrosine 15 of Cdk1. By contrast, none of the Bora variants tested rescued immunodepletion of Bora. Taken together, these results indicate that the Bora determinants required for Plk1 binding are essential for mitotic entry in Xenopus egg extracts.

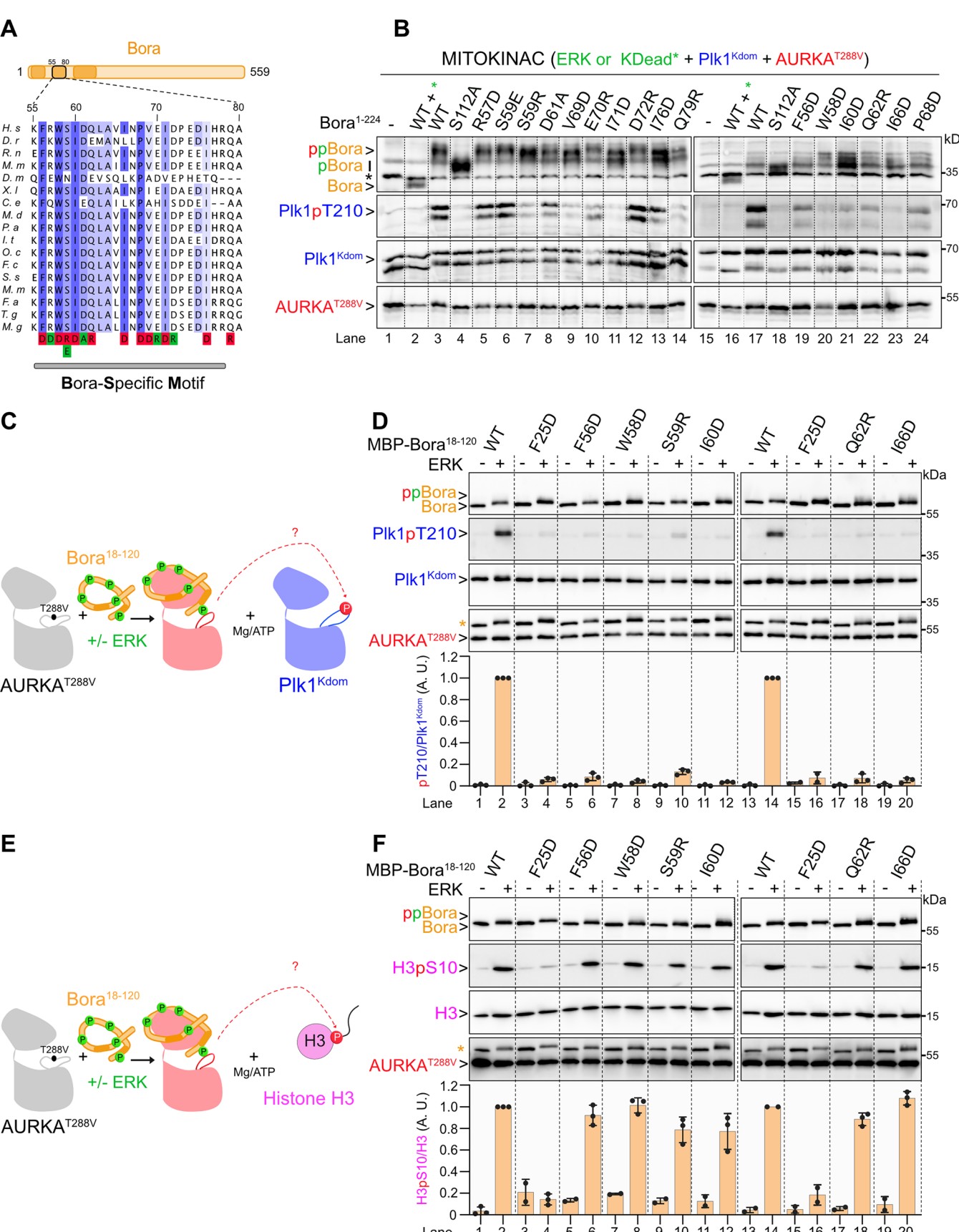

**Figure 6. pBora selectively activates AURKA toward the Plk1 T-loop via a novel evolutionarily conserved motif predicted to bind the Plk1 kinase domain.**

(A) Multiple protein sequence alignments of the Bora region located between the M1 and M2/M3 motifs. Identical residues are shown in dark blue. The residues mutated, and the nature of the substitutions are indicated at the bottom of the alignment. The impact of the mutations on Bora's function is indicated by the rectangle's color around the amino acids. Green rectangles indicate no effect, while red rectangles indicate a loss of function. (B) Western blot analysis of bacterial extracts reconstituting the mitotic kinase cascade in *E. coli* BL21, resulting in T-loop (T210) phosphorylation of the Plk1 kinase domain (Plk1$^{Kdom}$) using wild-type and Bora mutants. Blots were probed with antibodies to Bora, phospho-T210 Plk1, or pan Plk1, and AURKA (from top to bottom). ERK K-dead: kinase dead. (C) Schematic of the two-step in vitro reconstitution of T-loop phosphorylation on T210 of the Plk1 kinase domain (pT210) by AURKA$^{T288V}$ and pBora$^{18-120}$. In step 1, pBora$^{18-120}$ is incubated 1 h at 30 °C with the ERK kinase and Mg/ATP. In step 2, the reaction mix in step 1 is incubated in the presence of Mg/ATP with AURKA$^{T288V}$ and Plk1$^{KDom}$ for 30 min at 30 °C. (D) Western blot analysis of 2-step kinase reactions carried out with MBP-Bora$^{18-120}$ wild-type or mutant phosphorylated (+) or not (−) by the ERK kinase (step 1) in the presence of Plk1$^{KDom}$ (substrate) and AURKA$^{T288V}$ (step 2). Blots were probed with antibodies to Bora, phosphoT210 Plk1, or pan Plk1, and AURKA, as indicated (from top to bottom). As membranes were sequentially revealed with different antibodies, the orange asterisk indicates the signal revealed by the anti-Bora antibody. The graph presents the normalized quantification of pT210 Plk1 signal over Plk1$^{KDom}$ from $n = 3$ independent experiments. (E) Schematic of the two-step in vitro reconstitution of Histone 3 phosphorylation on serine 10 (pS10) by AURKA$^{T288V}$ and pBora$^{18-120}$. In step 1, pBora$^{18-120}$ is incubated 1 h at 30 °C with the ERK kinase and Mg/ATP. In step 2, the reaction mix in step 1 is incubated in the presence of Mg/ATP with AURKA$^{T288V}$ and Histone H3 for 30 min at 30 °C. (F) Western blot analysis of 2-step kinase reactions carried out with MBP-Bora$^{18-120}$ wild-type or mutant phosphorylated (+) or not (−) by the ERK kinase (step 1) in the presence of Histone H3 (substrate) and AURKA$^{T288V}$ (step 2). Blots were probed with antibodies to Bora, phosphoSer10 Histone H3, or pan Histone H3, and AURKA, as indicated (from top to bottom). The graph presents the normalized quantification of pS10 Histone H3 signal over Histone H3 from $n = 3$ independent experiments. Error bars display the standard deviation. Source data are available online for this figure.

## Discussion

By integrating AlphaFold structure modeling with extensive structure–function analysis, we discovered that the intrinsically disordered protein Bora acts as a unique and selective activator of Plk1 T-loop phosphorylation by AURKA. We show that once phosphorylated, Bora not only activates unphosphorylated AURKA by delivering a critical phosphosite near the T-loop in trans to functionally replace the phosphate normally provided by T-loop autophosphorylation, but it also selectively directs AURKA's kinase activity towards the Plk1 T-loop by interacting with the αC helix of the Plk1 kinase domain. pBora thus directs AURKA activity specifically to the Plk1 kinase domain, which is otherwise a poor substrate for AURKA. We have identified the specificity determinants in a conserved region of Bora that tethers AURKA to the Plk1 kinase domain, positioning AURKA's active site close to its substrate. Furthermore, we confirmed the importance of these determinants for Bora's function during mitotic entry in Xenopus egg extracts. Our findings reveal important details about the molecular mechanisms that drive mitotic kinase activation during mitosis, potentially explaining how cells manage the challenge of limiting Plk1 kinase activity in space and time.

### AF3 predicts that the minimal Bora$^{18-120}$ fragment wraps around the N-terminal lobe of AURKA to position the activating phosphate in the T-loop of AURKA

While it is well established that Bora physically binds to and activates AURKA (Hutterer et al, 2006; Bruinsma et al, 2017; Tavernier et al, 2021b), the mode of interaction remains unclear. Based on protein sequence alignments, we previously reported that Bora$^{18-120}$ shares some AURKA-binding elements with Tpx2$^{1-43}$, particularly aromatic residues in the M1 and M2 motifs. Accordingly, Bora$^{18-120}$ and Tpx2$^{1-43}$ compete for AURKA binding, and these aromatic residues are crucial for this competition (Tavernier et al, 2021b). The AlphaFold 3 model of the pBora$^{18-120}$-AURKA complex, thoroughly tested through our structure–function analysis, confirms and extends these observations by providing key additional insights. First, the model predicts that the M1 motif of Bora$^{18-120}$ (aa 22–45) is longer than expected and binds to the AURKA N-terminal lobe in an opposite orientation compared to Tpx2$^{1-43}$, with F25 binding to the

F-pocket and V44 and F45 binding to the Y pocket of AURKA. This explains why the Bora M1-M2-M3 fusion failed to activate AURKA$^{T288V}$ in ADP Glo assay (Fig. 1C,D), as it contains only an incomplete version of the M1 motif (aa 22–40), directly linked to the M2 motif, and thus the binding elements cannot engage AURKA correctly. Second, AF3 predicts that Bora$^{18-120}$ wraps around the N-terminal lobe of the kinase domain. On the other side of AURKA from the M1 and M2 binding motifs, pBora$^{18-120}$ binds to two AURKA pockets through the hydrophobic residues I60 and I71, which are also critical for Bora's function. Notably, previous structural studies have shown that these same two pockets are utilized by Cep192 (Park et al, 2023; Holder et al, 2024) and TACC3 (Burgess et al, 2018) to bind AURKA, suggesting that all these allosteric activators will compete for AURKA binding. This would help ensure their sequential (and not concomitant) action during mitotic entry and progression. After Bora$^{18-120}$ wraps around the N-terminal lobe of AURKA, the M2 motif is similarly positioned to that observed for Tpx2, and we demonstrated that F103 and F104 residues in this motif are critically required for Bora's function. These residues anchor the M2 motif to the W pocket of AURKA and position the phospho-M3 motif in the active site of AURKA. AF3 predicts that the phospho-motif M3 binds to the pocket composed of three arginine residues (R180, R255, and R286), a site normally occupied by the phosphorylated (pT288) T-loop (Bayliss et al, 2003). Thus, the M3 motif influences AURKA catalytic function (as observed for the phosphorylation of the T-loop) by stabilizing a productive conformation of the activation segment, a function normally supported by AURKA T-loop phosphorylation. Our finding that fusing the phospho-M3 motif of Bora to Tpx2 is sufficient to potentiate AURKA$^{T288V}$ catalytic activity further supports this hypothesis.

### The Bora-AURKA complex is a unique Plk1 activator during mitotic entry and progression

AURKA is characterized as an "incomplete" kinase because it relies heavily on its sequential interaction with various regulators to achieve maximal activity during mitotic entry and progression. These regulators localize to different places where they recruit and activate different forms of AURKA (Joukov and De Nicolo, 2018; Levinson, 2018; Tavernier et al, 2021b). In the cytoplasm, before nuclear envelope breakdown, Bora, phosphorylated by Cyclin

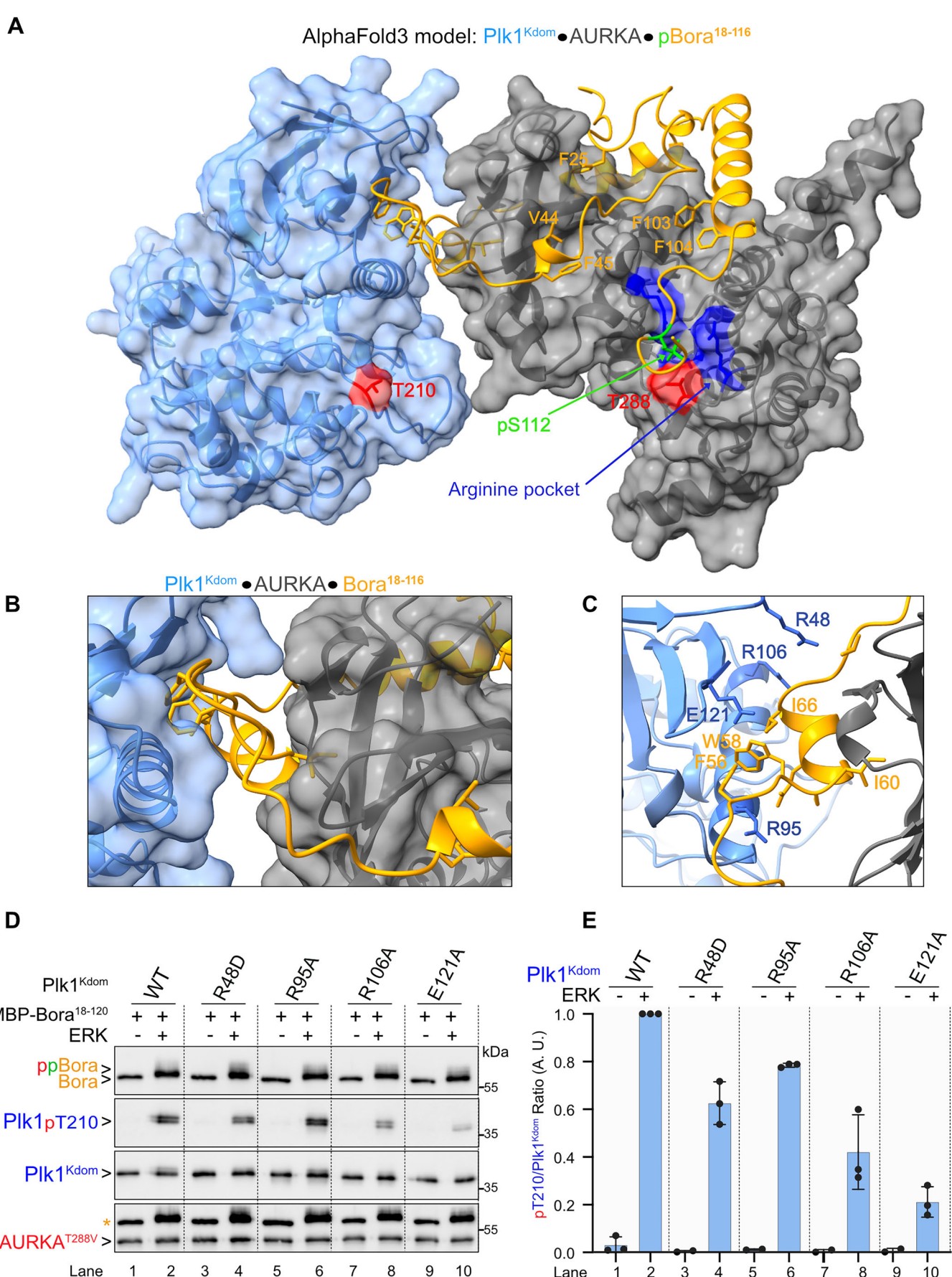

**A** AlphaFold3 model: Plk1^Kdom●AURKA●pBora^18-116

**B** Plk1^Kdom●AURKA●Bora^18-116

**C**

**D**

**E**

◄ **Figure 7. pBora engages the αC helix of Plk1 to deliver AURKA activity to the Plk1 T-loop.**

(A) AlphaFold 3 model of the trimeric complex composed of pBora¹⁸⁻¹²⁰ (orange), AURKA (gray), and the Plk1 kinase domain (blue). The residue T210 of the Plk1 T-loop, T288 of the AURKA T-loop are highlighted in red, and the phosphorylated residue S112 of Bora is highlighted in green. (B, C) Zoom in view of the AURKA/Bora/Plk1 interface, highlighting contacting residues. Color scheme as in (A). (D) Western blot analysis of 2-step kinase reactions carried out with MBP-Bora¹⁸⁻¹²⁰ wild-type or mutant phosphorylated (+) or not (−) by the ERK kinase (step 1) in the presence of the Plk1 kinase domain (substrate) and AURKA^T288V (step 2). Blots were probed with antibodies to Bora, phospho-T210 of Plk1, or pan Plk1, and AURKA, as indicated (from top to bottom). (E) The graph presents the normalized quantification of pT210 Plk1 signal over Plk1 from n = 3 independent experiments. Source data are available online for this figure.

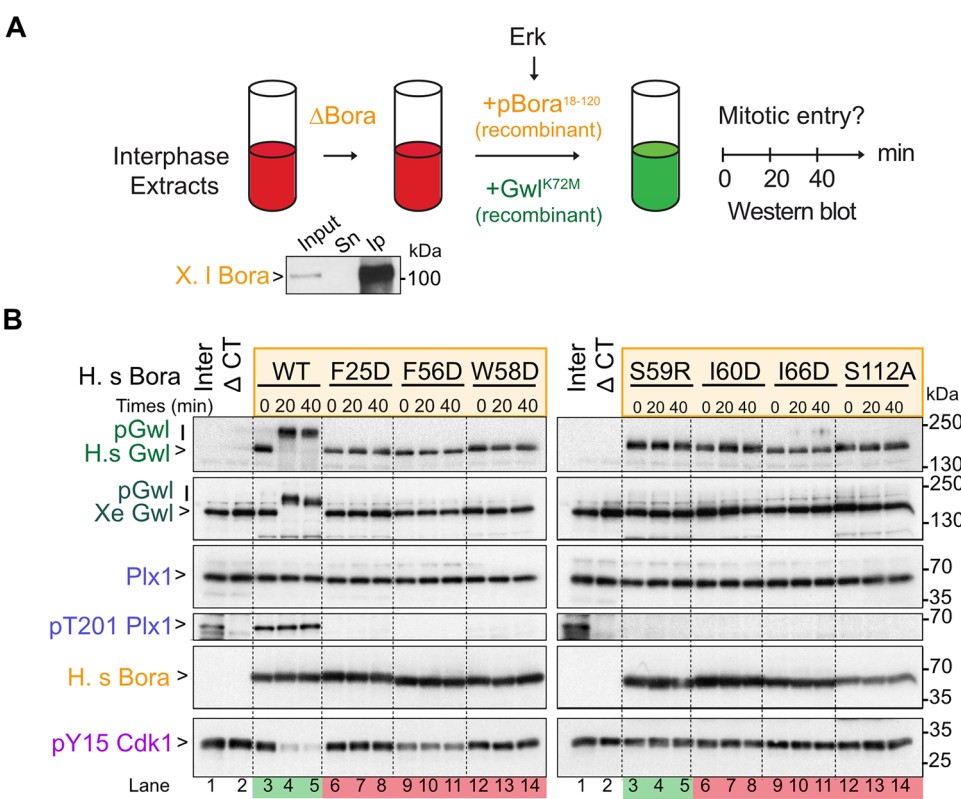

**Figure 8. Disrupting the Bora-Plk1 interface prevents mitotic entry in Xenopus egg extracts.**

(A) Schematic for the structure–function analysis of Bora during mitotic entry in Xenopus egg extracts. Interphase extracts (red) were depleted with Bora antibodies, and 30 min later supplemented with Bora¹⁸⁻¹²⁰ fragments (pre-phosphorylated by the ERK kinase) and with the recombinant human Gwl^K72M hyperactive kinase to force mitotic entry. Samples were collected at different time points and analyzed by Western blot (0, 20, 40 min) using mitotic markers to determine whether the extracts entered mitosis (green). The Western blot shows endogenous Bora levels before (Input) and after (Sn: supernatant) immunodepletion (Ip) from Xenopus egg extracts. (B) Interphase extracts were immunodepleted using anti-Bora antibodies and 30 min later supplemented with recombinant hyperactive human Gwl kinase (Gwl^K72M) and Bora¹⁸⁻¹²⁰ wild-type or variants pre-phosphorylated by ERK. A fraction of the extracts was collected at the indicated time-points (0, 20, 40 min) and analyzed by Western blot using specific antibodies to monitor the levels of Hu and Xe Gwl, Plx1, and Hu Bora as well as phosphorylation of Plx1 on T201 (pT201 Plx1), and Cdk1 on Tyr15 (pY15 Cdk1). Source data are available online for this figure.

A-Cdk1, preferentially binds to and activates unphosphorylated AURKA, which is kept dephosphorylated by the PP1 and PP2A phosphatases (Walter et al, 2000; Katayama et al, 2001; Kang et al, 2014). Here, we show that beyond activating AURKA, Bora specifically recruits its substrate Plk1. By binding both AURKA and Plk1, pBora focuses AURKA's activity on its substrate and likely facilitates the optimal positioning of the phosphoacceptor residue of Plk1, close to the active site of AURKA. Without Bora, AURKA cannot phosphorylate the Plk1 T-loop (Tavernier et al, 2021b). The interaction between Bora and the Plk1 kinase domain is essential for the efficient phosphorylation of the Plk1 T-loop by AURKA. Consistently, the pTpx2-Bora chimera, which lacks the

binding elements (Bora-Specific Motif) to the Plk1 kinase domain, activated unphosphorylated AURKA toward kemptide and Histone H3 robustly but failed to activate AURKA toward the Plk1 kinase domain. Similarly, pBora was inefficient at activating AURKA toward the isolated Plk1 T-loop, highlighting the importance of the transient interaction between this evolutionarily conserved region of Bora and Plk1. Our results show that the Plk1 T-loop is a poor substrate for AURKA. However, by concentrating active AURKA on the Plk1 kinase domain, pBora facilitates the phosphorylation of this impervious T210 site by AURKA.

Extending these observations, Esposito-Verza et al, in the accompanying manuscript, unequivocally demonstrate that the

sequence surrounding the phosphorylated T210 residue of the Plk1 T-loop is a sub-optimal consensus phosphorylation site for AURKA-mediated phosphorylation ([R-K-K-pT] instead of [R-R-X-pT/S]). Accordingly, a single K to R substitution in the −2 position is sufficient to convert the Plk1 T-loop into an optimal AURKA substrate (Esposito-Verza et al, 2026).

Likewise, in an independent but complementary study, Miles et al also reported that phospho-Bora bridges activated AURKA to PLK1 recognition, via the Bora-specific motif involving the F56 and W58 residues (Miles et al, 2026). Notably, S59 phosphorylation in this motif by AURKA potentiates Bora's function. These observations are consistent with our results obtained using MITOKINAC, showing that Bora S59R is partially defective in activating AURKA, whereas the phosphomimetic Bora S59E variant retains activity. This phosphorylation event, by stabilizing the ternary complex, potentiates Plk1 phosphorylation by AURKA. Also consistent with our present and previous observations, they show using Nuclear Magnetic Resonance (NMR) that Bora S112 phosphorylation locally increases Bora binding to the T-loop of AURKA and that pBora competes with Tpx2 and Cep192 for AURKA binding, validating the Alphafold model (Miles et al, 2026).

All these observations, together with our data, indicate that the AURKA•Bora complex is unique in phosphorylating the Plk1 T-loop during mitotic entry and progression.

### Limiting Plk1 activation during mitosis

What are the functional implications of these observations? Plk1 is crucial for mitotic entry and progression. Plk1 regulates mitotic entry, centrosome assembly and maturation, nuclear envelope breakdown, chromosome condensation, kinetochore function, and cytokinesis. Its activity must, therefore, be tightly controlled in space and time. Our results indicate that Plk1 T-loop phosphorylation by AURKA is inefficient and largely relies on phospho-Bora.

This mechanism, together with the extensive Plk1- and ubiquitin-mediated degradation of phospho-Bora (Chan et al, 2008; Seki et al, 2008a), may prevent uncontrolled Plk1 activation in the absence of signaling from Cyclin A/B-Cdk1. Both mechanisms may work in concert to maintain low levels of phosphorylated and, thus, active Plk1 during mitosis. Deregulation of these mechanisms, such as an accumulation of phosphorylated Bora, may lead to hyperactivation of the AURKA-Plk1 axis with detrimental consequences. Accordingly, Bora overexpression has been linked to oncogenic transformation and cancers with poor prognosis (Cheng et al, 2020; Parrilla et al, 2020; Barber et al, 2025). This work may, thus, pave the way for developing inhibitors of this pathway with therapeutic potential.

### Limitations of the study

Although we extensively tested the structural models predicted by Alphafold 3, the position of the residues we have mutated has not yet been observed experimentally using high-resolution structural biology methods. In addition, as mentioned above, outside the minimal active Bora[18–120] fragment, other regions potentially contributing to the assembly of the trimeric AURKA•pBora•Plk1 complex in vivo were not modeled here. In particular, the interaction between the Plk1 PBD domain and Bora phosphorylated at the Polo-docking site (S252). Finally, the stoichiometry of the interaction between the partners is undetermined. We modeled it 1:1:1, but a more complex assembly might form in vivo.

## Methods

### Molecular biology and DNA manipulation

All DNA constructs were generated by Gateway (Invitrogen) or restriction enzymes and verified by DNA sequencing.

### Reagents and Tools Table

| Reagent/resource | Reference or source | Identifier or catalog number |
| --- | --- | --- |
| **Experimental models** | | |
| *Xenopus laevis* | Centre de Ressources Biologiques Xenopes-Rennes | http://www.celphedia.eu/en/centers/crb |
| **Recombinant DNA** | | |
| pET | 6xHis-ERK2-MEK1_R4F | Addgene#39212 (Tavernier et al, 2021b) |
| pETM30-2 | 6xHis-GST-TEV-AURKA | pNT256 (Tavernier et al, 2021b) |
| pETM30-2 | 6xHis-GST-TEV-AURKA[T288V] | pNT257 (Tavernier et al, 2021b) |
| pDONR201 | TEV-Bora[1-224] | pLP1848 (Tavernier et al, 2021b) |
| pDEST17 | 6xHis-TEV-Bora[1-224] | pLP1850 (Tavernier et al, 2021b) |
| pGEX-2T | GST-TEV-Plk1[1-330] [K82R] | pLP2032 (Tavernier et al, 2021b) |
| pCDFDuet1 | GST-TEV-Plk1[1-330] [K82R] | pLP2228 |
| pDNOR201 | TEV-Bora[18-120] | pLP2072 |
| pCDFDuet1 | GST-Plk1 T-loop (aa 190–225) | pLP2546 |
| pRSFDuet-1 | Bora[1-224]-TEV-6xHis/Strep-TEV-AURKA[T288V] | pLP2200 |
| pRSFDuet-1 | Strep-TEV-AURKA[T288V] | pLP2204 |
| pRSFDuet-1 | Bora[1-224][S112A]-TEV-6xHis/Strep-TEV-AURKA[T288V] | pLP2508 |
| pET | 6xHis-ERK2 [K54R]-MEK1_R4F | pLP2548 |
| pKW2 | SepRS(2)/pSertRNA(B4)/EF-Sep | Addgene #173897 |
| pRSFDuet-1 | Bora[1-224]-TEV-(6xHis)- Strep-TEV-AURKA[T288V] [TAA Stop] | pLP3007 |
| pET28b | Bora[1-224]-TEV-6xHis [S112-TAG Amber codon] | pLP1988 |

| Reagent/resource | Reference or source | Identifier or catalog number |
|---|---|---|
| pRSFDuet-1 | Bora[1-224] [S112-TAG Amber codon]TEV (6xHis)- Strep-TEV-AURKA[T288V] [TAA Stop] | pLP2855 |
| pRSFDuet-1 | Bora[1-224] [F25D]-TEV-6xHis/Strep-TEV-AURKA[T288V] | pLP2844 |
| pRSFDuet-1 | Bora[1-224] [E26R]-TEV-6xHis/Strep-TEV-AURKA[T288V] | pLP2845 |
| pRSFDuet-1 | Bora[1-224] [E26A]-TEV-6xHis/Strep-TEV-AURKA[T288V] | pLP2846 |
| pRSFDuet-1 | Bora[1-224] [D30R]-TEV-6xHis/Strep-TEV-AURKA[T288V] | pLP2847 |
| pRSFDuet-1 | Bora[1-224] [Y31D]-TEV-6xHis/Strep-TEV-AURKA[T288V] | pLP2848 |
| pRSFDuet-1 | Bora[1-224] [L39D]-TEV-6xHis/Strep- TEV-AURKA[T288V] | pLP2849 |
| pRSFDuet-1 | Bora[1-224] [V44D]-TEV-6xHis/Strep- TEV-AURKA[T288V] | pLP2850 |
| pRSFDuet-1 | Bora[1-224] [F45D]-TEV-6xHis/Strep- TEV-AURKA[T288V] | pLP2530 |
| pRSFDuet-1 | Bora[1-224] [V44D-F45D]-TEV-6xHis/Strep- TEV-AURKA[T288V] | pLP2851 |
| pRSFDuet-1 | Bora[1-224] [K46E]-TEV-6xHis/Strep- TEV-AURKA[T288V] | pLP2852 |
| pRSFDuet-1 | Bora[1-224] [T48V]-TEV-6xHis/Strep- TEV-AURKA[T288V] | pLP2853 |
| pRSFDuet-1 | Bora[1-224] [L34D]-TEV-6xHis/Strep- TEV-AURKA[T288V] | pLP2529 |
| pRSFDuet-1 | Bora[1-224] [R57D]-TEV-6xHis/Strep- TEV-AURKA[T288V] | pLP2859 |
| pRSFDuet-1 | Bora[1-224] [S59E]-TEV-6xHis/Strep- TEV-AURKA[T288V] | pLP2860 |
| pRSFDuet-1 | Bora[1-224] [S59R]-TEV-6xHis/Strep- TEV-AURKA[T288V] | pLP2861 |
| pRSFDuet-1 | Bora[1-224] [D61A]-TEV-6xHis/Strep- TEV-AURKA[T288V] | pLP2862 |
| pRSFDuet-1 | Bora[1-224] [V69D]-TEV-6xHis/Strep- TEV-AURKA[T288V] | pLP2863 |
| pRSFDuet-1 | Bora[1-224] [E70R]-TEV-6xHis/Strep- TEV-AURKA[T288V] | pLP2864 |
| pRSFDuet-1 | Bora[1-224] [I71D]-TEV-6xHis/Strep- TEV-AURKA[T288V] | pLP2865 |
| pRSFDuet-1 | Bora[1-224] [D72R]-TEV-6xHis/Strep- TEV-AURKA[T288V] | pLP2866 |
| pRSFDuet-1 | Bora[1-224] [I76D]-TEV-6xHis/Strep- TEV-AURKA[T288V] | pLP2867 |
| pRSFDuet-1 | Bora[1-224] [Q79R]-TEV-6xHis/Strep- TEV-AURKA[T288V] | pLP2868 |
| pRSFDuet-1 | Bora[1-224] [F56D]-TEV-6xHis/Strep- TEV-AURKA[T288V] | pLP2895 |
| pRSFDuet-1 | Bora[1-224] [W58D]-TEV-6xHis/Strep- TEV-AURKA[T288V] | pLP2896 |
| pRSFDuet-1 | Bora[1-224] [I60D]-TEV-6xHis/Strep- TEV-AURKA[T288V] | pLP2532 |
| pRSFDuet-1 | Bora[1-224] [Q62R]-TEV-6xHis/Strep- TEV-AURKA[T288V] | pLP2908 |
| pRSFDuet-1 | Bora[1-224] [I66D]-TEV-6xHis/Strep- TEV-AURKA[T288V] | pLP2897 |
| pRSFDuet-1 | Bora[1-224] [P68D]-TEV-6xHis/Strep- TEV-AURKA[T288V] | pLP2534 |
| pRSFDuet-1 | Bora[1-224] [D91R]-TEV-6xHis/Strep- TEV-AURKA[T288V] | pLP2536 |
| pRSFDuet-1 | Bora[1-224] [E93R]-TEV-6xHis/Strep- TEV-AURKA[T288V] | pLP2537 |
| pRSFDuet-1 | Bora[1-224] [Q97A]-TEV-6xHis/Strep- TEV-AURKA[T288V] | pLP2538 |
| pRSFDuet-1 | Bora[1-224] [E101R]-TEV-6xHis/Strep- TEV-AURKA[T288V] | pLP2869 |
| pRSFDuet-1 | Bora[1-224] [F103D]-TEV-6xHis/Strep- TEV-AURKA[T288V] | pLP2870 |
| pRSFDuet-1 | Bora[1-224] [F104D]-TEV-6xHis/Strep- TEV-AURKA[T288V] | pLP2871 |
| pRSFDuet-1 | Bora[1-224] [I109R]-TEV-6xHis/Strep- TEV-AURKA[T288V] | pLP2873 |
| pRSFDuet-1 | Bora[1-224] [I109D]-TEV-6xHis/Strep- TEV-AURKA[T288V] | pLP2874 |
| pRSFDuet-1 | Bora[1-224] [V110R]-TEV-6xHis/Strep- TEV-AURKA[T288V] | pLP2875 |
| pRSFDuet-1 | Bora[1-224] [P111D]-TEV-6xHis/Strep- TEV-AURKA[T288V] | pLP2539 |
| pMAL-C2 | MBP-TEV-Bora[18-120] | pLP2784 |
| pMAL-C2 | MBP-TEV-Bora[18-120] [F25D] | pLP2915 |
| pMAL-C2 | MBP-TEV-Bora[18-120] [V44D-F45D] | pLP2987 |
| pMAL-C2 | MBP-TEV-Bora[18-120] [S112A] | pLP2928 |
| pMAL-C2 | MBP-TEV-Bora[18-120] [F103D] | pLP2988 |
| pMAL-C2 | MBP-TEV-Bora[18-120] [F104D] | pLP2995 |
| pMAL-C2 | MBP-TEV-Bora[18-120] [F56D] | pLP2916 |
| pMAL-C2 | MBP-TEV-Bora[18-120] [W58D] | pLP2917 |
| pMAL-C2 | MBP-TEV-Bora[18-120] [S59R] | pLP2918 |
| pMAL-C2 | MBP-TEV-Bora[18-120] [I60D] | pLP2919 |
| pMAL-C2 | MBP-TEV-Bora[18-120] [Q62R] | pLP2920 |
| pMAL-C2 | MBP-TEV-Bora[18-120] [I66D] | pLP2921 |
| pCDFDuet1 | GST-TEV-Plk1[1-330] [K82R, R48D] | pLP2974 |
| pCDFDuet1 | GST-TEV-Plk1[1-330] [K82R, R95A] | pLP2902 |
| pCDFDuet1 | GST-TEV-Plk1[1-330] [K82R, R106A] | pLP2955 |

| Reagent/resource | Reference or source | Identifier or catalog number |
| --- | --- | --- |
| pCDFDuet1 | GST-TEV-Plk1$^{1-330}$ [K82R, E121A] | pLP2905 |
| **Antibodies** | | |
| Mouse anti-Bora | Santa-Cruz | Cat#sc-393741 |
| Rabbit anti-AURKA | Cell Signaling | Cat#91590 |
| Rabbit anti-Phospho-Plk1 (Thr210) | Cell Signaling | Cat#5472 |
| Rabbit anti-human Plk1 | Bethyl | Cat#A300-250A |
| Rabbit anti-GST | Cell Signaling | #2622 |
| Rabbit anti-Phospho-Histone H3 (Ser10) | Cell Signaling | Cat#3377S |
| Rabbit anti-Histone H3 | Abcam | Cat#AB1791 |
| Mouse anti-MBP | NEB biolabs | Cat#E8032S |
| Rabbit anti-human Greatwall | Burgess et al, 2010 | N/A |
| Recombinant GST-Human Greatwall K72M mutant | Vigneron et al, 2011 | N/A |
| Rabbit polyclonal anti-Xenopus Greatwall | Vigneron et al, 2011 | N/A |
| Rabbit anti-Xenopus Plx1 | Vigneron et al, 2018 | N/A |
| Rabbit anti-phospho-Cdc2 (Tyr15) | Cell Signaling | Cat#9111 |
| Peroxidase goat anti-mouse IgG (H + L) | Sigma | Cat#A9917 |
| Peroxidase goat anti-rabbit IgG (H + L) | Sigma | Cat#A0545 |
| Dynabeads protein G | Life technology | Cat#10004D |
| HRP-conjugated anti-Mouse secondary antibodies | Bio-Rad | Cat#1172-1011 |
| Rabbit polyclonal phospho-Cdc2 (Tyr-15) | CELL SIGNALING | Cat#9111 |
| HRP-conjugated anti-rabbit secondary antibodies | CELL SIGNALING | Cat#7074 |
| **Oligonucleotides and other sequence-based reagents** | | |
| Forward and reverse primers to amplify and clone AURKA in pETM30 NcoI/XhoI (pNT256): CGCCATGGACCGATCTAAAGAAAAC-F GGCCTCGAGCTAAGACTGTTTGCTAGCTG-R | | oNT37 oNT38 |
| Forward and reverse primers for AURKA$^{T288V}$ (pNT257): GCTCCATCCTCCAGGAGGACCGTCCTCTGTGGCACCCTGGACTACC-F GGTAGTCCAGGGTGCCACAGAGGACGGTCCTCCTGGAGGATGGAGC-R | | oNT123 oNT124 |
| Forward and reverse primers to clone Bora$^{1-224}$ Gateway (pLP1848-pLP1850): GGGGACAAGTTTGTACAAAAAAGCAGGCTTCATGGGAGATGTCAAGGAA TCAAAG-F GGGGACCACTTTGTACAAGAAAGCTGGGTCCTACTCTAGTGATGTTTGAACACC-R | | oLP989 oLP990 |
| Forward and reverse primers for Plk1$^{1-330}$ mutagenesis K82R (pLP2032): GAGGTGTTCGCGGGCCGGATTGTGCCTAAGTCTCTGCT-F AGCAGAGACTTAGGCACAATCCGGCCCGCGAACACCTC-R | | oNT118 oNT119 |
| Forward and reverse primers for GST-TEV-Plk1$^{1-330}$ K82R insertion into pCDFDuet-1 (pLP2228) TATACAATTGGATGTCCCCTATACTAGGTTATTGG-F CAGAGGTACCTTAGCTGGGAGCAATCGAAAAC-R | | oLP2195 oLP2196 |
| Forward and reverse primers to clone TEV-Bora$^{18-120}$ (pLP2072) Gateway GGGGACAAGTTTGTACAAAAAAGCAGGCTTCCCAACGACCGAAAACCTGTATTTTCAG GGCGCCATGGATCCGGAATTCAGGATCCCTGTTTTAAATCCTTTTG-F GGGGACCACTTTGTACAAGAAAGCTGGGTCTTATTTCCCTTCATGATCAGTCCAAG-R | | oLP1994 oLP1997 |
| Forward and reverse primers for GST-Plk1 T-loop 190–225 (pLP2546) ATGGATCCGGAATTCAAAATAGGGGATTTTGGAC-F CTCGAGGGTACCTTACTTGCTCAGCACCTCGGGAG-R | | oLP2804 oLP2805 |
| Forward and reverse primers to amplify and clone Bora$^{1-224}$-TEV-6xHis in pRSF-Duet (pLP2200) GGGCCATGGGAGATGTCAAGGAATCAAAG-F CCGGAATTCGGTCAGTGGTGGTGGTGGTGGTGGAATCCTTGGAAGTATAGATTTTC ACCCTCTAGTGATGTTTGAACACCAC-R | | oLP1917oLP2116 |
| Forward and reverse primers to amplify and clone Strep-TEV-AURKA$^{T288V}$ in pRSF-Duet pLP2200) TATACATATGGCTAGCTGGAGCCACCCGCAGTTCGAAAAAGGATCTGGTGGTGGT GGTGGTGAAAACCTGTACTTCCAGGGAATGGACCGATCTAAAGAAAACTGC-F CAGACTCGAGCTAAGACTGTTTGCTAGCTG-R | | oLP2132 oLP2133 |
| Forward and reverse primers to amplify and clone Strep-TEV-AURKA$^{T288V}$ in pLP2204 TATACATATGGCTAGCTGGAGCCACCCGCAGTTCGAAAAAGGATCTGGTGGTGGT GGTGGTGAAAACCTGTACTTCCAGGGAATGGACCGATCTAAAGAAAACTGC-F CAGACTCGAGCTAAGACTGTTTGCTAGCTG-R | | oLP2132 oLP2133 |
| Forward and reverse primers for SDM Bora$^{1-224}$[S112A]-TEV-6xHis in pLP2200. GATGTCATCGTACCCGCTCCTTGGACTGATCATGAAGG-F CCTTCATGATCAGTCCAAGGAGCGGGTACGATGACATC-R | | oLP978 oLP979 |
| Forward and reverse primers for SDM pLP2401 6xHis-ERK2 [K54R]-MEK1_R4F (pLP2548). CAAAGTTCGAGTTGCTATCCGGAAAATCAGTCCTTTTGAGC-F GCTCAAAAGGACTGATTTTCCGGATAGCAACTCGAACTTTG-R | | oLP2806 oLP2807 |
| Forward and reverse primers for SDM Bora$^{1-224}$ [S112-TAG Amber codon] in pET28b (pLP1988). CACTAAAGATGTCATCGTACCCTAGCCTTGGACTGATCATGAAGGG-F CCCTTCATGATCAGTCCAAGGCTAGGGTACGATGACATCTTTAGTG-R | | oLP1880 oLP1881 |

| Reagent/resource | Reference or source | Identifier or catalog number |
|---|---|---|
| Forward and reverse primers to amplify and clone Bora[1-224] with TEV-(6xHis) in pRSF-Duet pLP3007.<br>GGGCCATGGGAGATGTCAAGGAATCAAAG-F<br>CCGGAATTCGGTCAGTGGTGGTGGTGGTGGTGGAATCCTTGGAAGTATAGATTTTCAC<br>CCTCTAGTGATGTTTGAACACCAC-R | | oLP1917oLP2116 |
| Forward and reverse primers to amplify, clone and mutate the stop codon of Strep-TEV-AURKA[T288V] [TAA Stop] in pRSF-Duet pLP3007<br>TATACATATGGCTAGCTGGAGCCACCCGCAGTTCGAAAAAGGATCTGGTGGTGGT<br>GGTGGTGAAAACCTGTACTTCCAGGGAATGGACCGATCTAAAGAAAACTGC-F<br>CCAGACTCGAGTTAAGACTGTTTGCTAGCTGATTC-R | | oLP2132<br>oLP3546 |
| Forward and reverse primers for Bora[1-224] [S112-TAG Amber codon] TEV (6xHis) in pRSF-Duet pLP2855<br>GGGCCATGGGAGATGTCAAGGAATCAAAG-F<br>CCGGAATTCGGTCAGTGGTGGTGGTGGTGGTGGAATCCTTGGAAGTATAGATTTTCACCC<br>TCTAGTGATGTTTGAACACCAC-R | | oLP1917<br>oLP2116 |
| Forward and reverse primers for amplify, clone and mutated the stop codon of Strep-TEV-AURKA[T288V] [TAA Stop] in pRSF-duet pLP2855<br>TATACATATGGCTAGCTGGAGCCACCCGCAGTTCGAAAAAGGATCTGGTGGTGGTGGT<br>GGTGAAAACCTGTACTTCCAGGGAATGGACCGATCTAAAGAAAACTGC-F<br>CCAGACTCGAGTTAAGACTGTTTGCTAGCTGATTC-R | | oLP2132<br>oLP3546 |
| Forward and reverse primers for Bora[1-224] [F25D]-TEV-6xHis in pLP2200 (pLP2844)<br>CCCTGTTTTAAATCCTGATGAAAGTCCTAGTGAT-F<br>CACTAGGACTTTCATCAGGATTTAAAACAGGGAT-R | | oLP3488<br>oLP3489 |
| Forward and reverse primers for Bora[1-224] [E26R]-TEV-6xHis in pLP2200 (pLP2845)<br>CCCTGTTTTAAATCCTTTTCGTAGTCCTAGTGATTATTCT-F<br>GAATAATCACTAGGACTACGAAAAGGATTTAAAACAGGGA-R | | oLP3490<br>oLP3491 |
| Forward and reverse primers for Bora[1-224] [E26A]-TEV-6xHis in pLP2200 (pLP2846)<br>CCCTGTTTTAAATCCTTTTGCGAGTCCTAGTGATTATTCTA-F<br>GAATAATCACTAGGACTCGCAAAAGGATTTAAAACAGGGA-R | | oLP3492<br>oLP3493 |
| Forward and reverse primers for Bora[1-224] [D30R]-TEV-6xHis in pLP2200 (pLP2847)<br>CCTTTTGAAAGTCCTAGTCGTTATTCTAATCTCCATGA-F<br>CATGGAGATTAGAATAACGACTAGGACTTTCAAAAGGAT-R | | oLP3494<br>oLP3495 |
| Forward and reverse primers for Bora[1-224] [Y31D]-TEV-6xHis in pLP2200 (pLP2848)<br>GAAAGTCCTAGTGATGATTCTAATCTCCATGAACAA-F<br>GAGTTTGTTCATGGAGATTAGAATCATCACTAGGACTTTCA-R | | oLP3496<br>oLP3497 |
| Forward and reverse primers for Bora[1-224] [L39D]-TEV-6xHis in pLP2200 (pLP2849)<br>GAGTTTGTTCATGGAGATTAGAATCATCACTAGGACTTTCA-F<br>GATTTAAAAACAGAAGGACTGGCGTCAGTTTGTTCATGGAGATT-R | | oLP3498<br>oLP3499 |
| Forward and reverse primers for Bora[1-224] [V44D]-TEV-6xHis in pLP2200 (pLP2850)<br>CTCTCGCCAGTCCTTCTGATTTTAAATCAACAAAATTACCA-F<br>GGTAATTTTGTTGATTTAAATCAGAAGGACTGGCGAGAGT-R | | oLP3500<br>oLP3501 |
| Forward and reverse primers for Bora[1-224] [F45D]-TEV-6xHis in pLP2200 (pLP2530)<br>GAACAAACTCTCGCCAGTCCTTCTGTTGATAAATCAACAAAATTACCAACTCCAGG-F<br>CCTGGAGTTGGTAATTTTGTTGATTTATCAACAGAAGGACTGGCGAGAGTTTGTTC-R | | oLP2771<br>oLP2772 |
| Forward and reverse primers for Bora[1-224] [V44D-F45D]-TEV-6xHis in pLP2200 (pLP2851)<br>CAAACTCTCGCCAGTCCTTCTGATGATAAATCAACAAAATTACCAACTCCA-F<br>GGAGTTGGTAATTTTGTTGATTTATCATCAGAAGGACTGGCGAGAGTTTGT-R | | oLP3502<br>oLP3503 |
| Forward and reverse primers for Bora[1-224] [K46E]-TEV-6xHis in pLP2200 (pLP2852)<br>CCAGTCCTTCTGTTTTTGAATCAACAAAATTACCA-F<br>GGTAATTTTGTTGATTCAAAAACAGAAGGACTGGCGA-R | | oLP3504<br>oLP3505 |
| Forward and reverse primers for Bora[1-224] [T48V]-TEV-6xHis in pLP2200 (pLP2853)<br>CCTTCTGTTTTTAAATCAGTGAAATTACCAACTCCA-F<br>CCCTGGAGTTGGTAATTTCACTGATTTAAAAACAGA-R | | oLP3506<br>oLP3507 |
| Forward and reverse primers for Bora[1-224] [L34D]-TEV-6xHis in pLP2200 (pLP2529)<br>GTGATTATTCTAATGACCATGAACAAACTCTCGCC-F<br>GGCGAGAGTTTGTTCATGGTCATTAGAATAATCAC-R | | oLP2769<br>oLP2770 |
| Forward and reverse primers for Bora[1-224] [R57D]-TEV-6xHis in pLP2200 (pLP2859)<br>CCAACTCCAGGGAAATTTGATTGGTCTATTGATCA-F<br>GTTGATCAATAGACCAATCAAATTTCCCTGGAGT-R | | oLP3508<br>oLP3509 |
| Forward and reverse primers for Bora[1-224] [S59E]-TEV-6xHis in pLP2200 (pLP2860)<br>GGGAAATTTAGATGGGAAATTGATCAACTAGCT-F<br>CAGCTAGTTGATCAATTTCCCATCTAAATTTCCCT-R | | oLP3512<br>oLP3513 |
| Forward and reverse primers for Bora[1-224] [S59R]-TEV-6xHis in pLP2200 (pLP2861)<br>GGGAAATTTAGATGGCGTATTGATCAACTAGCT-F<br>CAGCTAGTTGATCAATACGCCATCTAAATTTCCCT-R | | oLP3510<br>oLP3511 |
| Forward and reverse primers for Bora[1-224] [D61A]-TEV-6xHis in pLP2200 (pLP2862)<br>GGGAAATTTAGATGGTCTATTGCGCAACTAGCTGTAATAAATCCT-F<br>GGATTTATTACAGCTAGTTGCGCAATAGACCATCTAAATTTCCCT-R | | oLP3514<br>oLP3515 |
| Forward and reverse primers for Bora[1-224] [V69D]-TEV-6xHis in pLP2200 (pLP2863)<br>GCTGTAATAAATCCTGATGAAATAGACCCAGAAGA-F<br>CTTCTGGGTCTATTTCATCAGGATTTATTACAGCT-R | | oLP3516<br>oLP3517 |
| Forward and reverse primers for Bora[1-224] [E70R]-TEV-6xHis in pLP2200 (pLP2864)<br>GCTGTAATAAATCCTGTACGTATAGACCCAGAAGA -F<br>CTTCTGGGTCTATACGTACAGGATTTATTACA-R | | oLP3518<br>oLP3519 |
| Forward and reverse primers for Bora[1-224] [I71D]-TEV-6xHis in pLP2200 (pLP2865)<br>GCTGTAATAAATCCTGTAGAAGATGACCCAGAAGATATTCA-F<br>CGATGAATATCTTCTGGGTCATCTTCTACAGGATTTATTACA-R | | oLP3520<br>oLP3521 |
| Forward and reverse primers for Bora[1-224] [D72R]-TEV-6xHis in pLP2200 (pLP2866)<br>CCTGTAGAAATACGTCCAGAAGATATTCA-F<br>CGATGAATATCTTCTGGACGTATTTCTACAGGA-R | | oLP3522<br>oLP3523 |

| Reagent/resource | Reference or source | Identifier or catalog number |
|---|---|---|
| Forward and reverse primers for Bora[1-224] [I76D]-TEV-6xHis in pLP2200 (pLP2867)<br>GAAATAGACCCAGAAGATGATCATCGTCAAGCTTTATACT-F<br>GTATAAAGCTTGACGATGATCATCTTCTGGGTCTATTTCT-R | | oLP3526<br>oLP3527 |
| Forward and reverse primers for Bora[1-224] [Q79R]-TEV-6xHis in pLP2200 (pLP2868)<br>CCCAGAAGATATTCATCGTCGTGCTTTATACTTAAGTCA-F<br>GACTTAAGTATAAAGCACGACGATGAATATCTTCTGGGT-R | | oLP3528<br>oLP3529 |
| Forward and reverse primers for Bora[1-224] [F56D]-TEV-6xHis in pLP2200 (pLP2895)<br>CCAACTCCAGGGAAAGATAGATGGTCTATTGATC-F<br>GATCAATAGACCATCTATCTTTCCCTGGAGTTGG-R | | oLP3605<br>oLP3606 |
| Forward and reverse primers for Bora[1-224] [W58D]-TEV-6xHis in pLP2200 (pLP2896)<br>CTCCAGGGAAATTTAGAGACTCTATTGATCAACTA-F<br>TAGTTGATCAATAGAGTCTCTAAATTTCCCTGGAG-R | | oLP3607<br>oLP3608 |
| Forward and reverse primers for Bora[1-224] [I60D]-TEV-6xHis in pLP2200 (pLP2532)<br>GGGAAATTTAGATGGTCTGATGATCAACTAGCTGTAATAAATCC-F<br>GGATTTATTACAGCTAGTTGATCATCAGACCATCTAAATTTCCC-R | | oLP2775<br>oLP2776 |
| Forward and reverse primers for Bora[1-224] [Q62R]-TEV-6xHis in pLP2200 (pLP2908)<br>GATGGTCTATTGATCGCCTAGCTGTAATAAATCCT-F<br>AGGATTTATTACAGCTAGGCGATCAATAGACCATC-R | | oLP3629<br>oLP3630 |
| Forward and reverse primers for Bora[1-224] [I66D]-TEV-6xHis in pLP2200 (pLP2897)<br>GAAATTTAGATGGTCTGATGATCAACTAGCTGTAA-F<br>TTACAGCTAGTTGATCATCAGACCATCTAAATTTC-R | | oLP3609<br>oLP3610 |
| Forward and reverse primers for Bora[1-224] [P68D]-TEV-6xHis in pLP2200 (pLP2534)<br>CAACTAGCTGTAATAAATGATGTAGAAATAGACCCAGAAG-F<br>CTTCTGGGTCTATTTCTACATCATTTATTACAGCTAGTTG-R | | oLP2779<br>oLP2780 |
| Forward and reverse primers for Bora[1-224] [D91R]-TEV-6xHis in pLP2200 (pLP2536)<br>TTCTCGAATAGATAAACGTGTGGAAGACAAAAGAC-F<br>GTCTTTTGTCTTCCACACGTTTATCTATTCGAGAA-R | | oLP2783<br>oLP2784 |
| Forward and reverse primers for Bora[1-224] [E93R]-TEV-6xHis in pLP2200 (pLP2537)<br>CGAATAGATAAAGATGTGCGAGACAAAAGACAAAAAGC-F<br>GCTTTTTGTCTTTTGTCTCGCACATCTTTATCTATTCG-R | | oLP2785<br>oLP2786 |
| Forward and reverse primers for Bora[1-224] [Q97A]-TEV-6xHis in pLP2200 (pLP2538)<br>GTGGAAGACAAAAGAGCAAAAGCCATTGAAGAGTTTTTC-F<br>GAAAAACTCTTCAATGGCTTTTGCTCTTTTGTCTTCCAC-R | | oLP2787<br>oLP2788 |
| Forward and reverse primers for Bora[1-224] [E101R]-TEV-6xHis in pLP2200 (pLP2869)<br>GACAAAAGACAAAAAGCCATTCGTGAGTTTTTCACTAAAGATGTC-F<br>GACATCTTTAGTGAAAAACTCACGAATGGCTTTTTGTCTTTTGTC-R | | oLP3588<br>oLP3589 |
| Forward and reverse primers for Bora[1-224] [F103D]-TEV-6xHis in pLP2200 (pLP2870)<br>GACAAAAAGCCATTGAAGAGGATTTCACTAAAGATGTCATCGT-F<br>CGATGACATCTTTAGTGAAATCCTCTTCAATGGCTTTTTGT-R | | oLP3532<br>oLP3533 |
| Forward and reverse primers for Bora[1-224] [F104D]-TEV-6xHis in pLP2200 (pLP2871)<br>GCCATTGAAGAGTTTGACACTAAAGATGTCATCGT-F<br>GGTACGATGACATCTTTAGTGTCAAACTCTTCAATGGCT-R | | oLP3534<br>oLP3535 |
| Forward and reverse primers for Bora[1-224] [I109R]-TEV-6xHis in pLP2200 (pLP2873)<br>GAGTTTTTCACTAAAGATGTCCGCGTACCCTCTCCTTGGACT-F<br>CAGTCCAAGGAGAGGGTACGCGGACATCTTTAGTGAAAAACT-R | | oLP3538<br>oLP3539 |
| Forward and reverse primers for Bora[1-224] [I109D]-TEV-6xHis in pLP2200 (pLP2874)<br>GAGTTTTTCACTAAAGATGTCGACGTACCCTCTCCTTGGACT-F<br>CAGTCCAAGGAGAGGGTACGTCGACATCTTTAGTGAAAAACT-R | | oLP3540<br>oLP3541 |
| Forward and reverse primers for Bora[1-224] [V110R]-TEV-6xHis in pLP2200 (pLP2875)<br>CACTAAAGATGTCATCCGTCCCTCTCCTTGGACTGA-F<br>CAGTCCAAGGAGAGGGACGGATGACATCTTTAGTGA-R | | oLP3542<br>oLP3543 |
| Forward and reverse primers for Bora[1-224] [P111D]-TEV-6xHis in pLP2200 (pLP2539)<br>GAGTTTTTCACTAAAGATGTCATCGTAGACTCTCCTTGGACTGATCATGAAGG-F<br>CCTTCATGATCAGTCCAAGGAGAGTCTACGATGACATCTTTAGTGAAAAACTC-R | | oLP2789<br>oLP2790 |
| Forward and reverse primers to clone MBP-TEV-Bora[18-120] WT by Gateway<br>with pLP2072 (pLP2784)<br>GGGGACAAGTTTGTACAAAAAAGCAGGCTTCCCAACGACCGAAAACCTGTATTTTCA<br>GGGCGCCATGGATCCGGAATTCAGGATCCCTGTTTTAAATCCTTTTG-F<br>GGGGACCACTTTGTACAAGAAAGCTGGGTCTTATTTCCCTTCATGATCAGTCCAAG-R | | oLP1994<br>oLP1997 |
| Forward and reverse primers for MBP-TEV-Bora[18-120] F25D (pLP2915)<br>CCCTGTTTTAAATCCTGATGAAAGTCCTAGTGAT-F<br>CACTAGGACTTTCATCAGGATTTAAAACAGGGAT-R | | oLP3488<br>oLP3489 |
| Forward and reverse primers for MBP-TEV-Bora[18-120] V44D-F45D<br>(pLP2987)<br>CAAACTCTCGCCAGTCC<br>TTCTGATGATAAATCAACAAAATTACCAACTCCA -F<br>GGAGTTGGTAATTTTGTTGATTTATCATCAGAAGGACTGGCGAGAGTTTGT-R | | oLP3502<br>oLP3503 |
| Forward and reverse primers for MBP-TEV-Bora[18-120] S112A (pLP2928)<br>GATGTCATCGTACCCGCTCCTTGGACTGATCATGAAGG-F<br>CCTTCATGATCAGTCCAAGGAGCGGGTACGATGACATC-R | | oLP978<br>oLP979 |
| Forward and reverse primers for MBP-TEV-Bora[18-120] F103D (pLP2988)<br>GACAAAAAGCCATTGAAGAGGATTTCACTAAAGATGTCATCGT-F<br>CGATGACATCTTTAGTGAAATCCTCTTCAATGGCTTTTTGT-R | | oLP3532<br>oLP3533 |
| Forward and reverse primers for MBP-TEV-Bora[18-120] F104D (pLP2995)<br>GCCATTGAAGAGTTTGACACTAAAGATGTCATCGT-F<br>GGTACGATGACATCTTTAGTGTCAAACTCTTCAATGGCT-R | | oLP3534<br>oLP3535 |
| Forward and reverse primers for MBP-TEV-Bora[18-120] F56D (pLP2916)<br>CCAACTCCAGGGAAAGATAGATGGTCTATTGATC-F<br>GATCAATAGACCATCTATCTTTCCCTGGAGTTGG-R | | oLP3605<br>oLP3606 |

| Reagent/resource | Reference or source | Identifier or catalog number |
|---|---|---|
| Forward and reverse primers for MBP-TEV-Bora[18-120] W58D (pLP2917)<br>CTCCAGGGAAATTTAGAGACTCTATTGATCAACTA-F<br>TAGTTGATCAATAGAGTCTCTAAATTTCCCTGGAG-R | | oLP3607<br>oLP3608 |
| Forward and reverse primers for MBP-TEV-Bora[18-120] S59R (pLP2918)<br>GGGAAATTTAGATGGCGTATTGATCAACTAGCT-F<br>CAGCTAGTTGATCAATACGCCATCTAAATTTCCCT-R | | oLP3510<br>oLP3511 |
| Forward and reverse primers for MBP-TEV-Bora[18-120] I60D (pLP2919)<br>GGGAAATTTAGATGGTCTGATGATCAACTAGCTGTAATAAATCC-F<br>GGATTTATTACAGCTAGTTGATCATCAGACCATCTAAATTTCCC-R | | oLP2775<br>oLP2776 |
| Forward and reverse primers for MBP-TEV-Bora[18-120] Q62R (pLP2920)<br>GATGGTCTATTGATCGCCTAGCTGTAATAAATCCT-F<br>AGGATTTATTACAGCTAGGCGATCAATAGACCATC-R | | oLP3629<br>oLP3630 |
| Forward and reverse primers for MBP-TEV-Bora[18-120] I66D (pLP2921)<br>GAAATTTAGATGGTCTGATGATCAACTAGCTGTAA-F<br>TTACAGCTAGTTGATCATCAGACCATCTAAATTTC-R | | oLP3609<br>oLP3610 |
| Forward and reverse primers for GST-TEV-Plk1[1-330] K82R-R48D (pLP2974)<br>GGTCCTAGTGGACCCAGACAGCCGGCGGCGCTATG-F<br>CATAGCGCCGCCGGCTGTCTGGGTCCACTAGGACC-R | | oLP3651<br>oLP3652 |
| Forward and reverse primers for GST-TEV-Plk1[1-330] K82R-R95A (pLP2902)<br>GCTCAAGCCGCACCAGGCGGAGAAGATGTCCATGG-F<br>CCATGGACATCTTCTCCGCCTGGTGCGGCTTGAGC-R | | oLP3615<br>oLP3616 |
| Forward and reverse primers for GST-TEV-Plk1[1-330] K82R-R106A (pLP2955)<br>GGAAATATCCATTCACGCGAGCCTCGCCCACCAGC-F<br>GCTGGTGGGCGAGGCTCGCGTGAATGGATATTTCC-R | | oLP3644<br>oLP3645 |
| Forward and reverse primers for GST-TEV-Plk1[1-330] K82R-E121A (pLP2905)<br>GGATTCCACGGCTTTTTCGCGGACAACGACTTCGT-F<br>ACGAAGTCGTTGTCCGCGAAAAAGCCGTGGAATCC-R | | oLP3621<br>oLP3622 |
| **Chemicals, enzymes, and other reagents** | | |
| ADP Glo[TM] | Promega | Cat#V6930 |
| Coomassie R250 | Sigma | Cat#B014925G |
| Ponceau Red | Sigma | Cat#A1405 |
| Amersham™ Protran® Western blotting membranes, nitrocellulose | Merk | GE10600002 |
| Protease inhibitor cocktail | Roche | 74506500 |
| Bio-Rad protein assay | Bio-Rad | #5000001 |
| IPTG | Euromedex | Cat#EU0008-B |
| Adenosine TriPhosphate (ATP) | Sigma | Cat#A2383 |
| Glutathione | Sigma | Cat#G4251 |
| PVDF Transfer Membrane | Millipore | Cat#88518 |
| Bovine Serum Albumin | Sigma | A2153 |
| 4–20% Mini-PROTEAN® TGX™ Precast Protein Gels | Bio-Rad | #4561096 |
| GSTrap 4B Columns | Cytiva | 28401747 |
| ECL reagent | Millipore | Cat#WBKLS0500 |
| BP Clonase II Enzyme Mix (Gateway cloning) | Invitrogen | Cat#11789-020 |
| LR Clonase II Enzyme Mix (Gateway cloning) | Invitrogen | Cat#11791-020 |
| Pfu | Promega | Cat#M7741 |
| DpnI | Biolabs | Cat#R0176S |
| **Software** | | |
| Clustal W2 | Larkin et al, 2007 | N/A |
| Jalview | Waterhouse et al, 2009 | N/A |
| ImageJ | Schneider et al, 2012 | N/A |
| PRISM | Graphpad | N/A |
| Affinity Designer 2 | Affinity | N/A |
| ChimeraX | RBVI | Meng et al, 2023 |
| **Other** | | |
| Chemidoc™ Gel Imaging System | Bio-Rad | N/A |

Oligonucleotides used for site-directed mutagenesis were purchased from Eurofins Genomics and are detailed, with the list of plasmids used in this study, in the Reagents and Tools Table.

pET-His6-ERK2-MEK1_R4F_coexpression was a gift from Melanie Cobb (Addgene plasmid #39212; http://n2t.net/addgene:39212; RRID: Addgene_39212). The plasmid pKW2: SepRS(2)/pSertR-NA(B4)/EF-Sep was a gift from Jason W Chin (Addgene plasmid #173897; http://n2t.net/addgene:173897; RRID: Addgene_173897), Machinery plasmid for pSer incorporation, chloramphenicol resistance/pBR322 origin of replication (Rogerson et al, 2015).

## MITOKINAC

*Escherichia coli* BL21 DE3 RIL (Thermo Fischer Scientific) chemically competent bacteria were co-transformed with three plasmids expressing (i) active ERK (pET-His6-ERK2-MEK1_R4F), (ii) Strep-AURKA$^{T288V}$ in operon with Bora$^{1-224}$-6xHis WT or mutants, and (iii) GST-Plk1$^{1-330\ K82R}$ (kinase-dead).

Transformants were selected on LB plates containing Kanamycin (30 µg/ml), Ampicillin (50 µg/ml), and Streptomycin (30 µg/ml) after overnight incubation at 37 °C. Colonies were then inoculated in LB medium containing Kanamycin (30 µg/ml), Ampicillin (50 µg/ml), and Streptomycin (30 µg/ml) until the culture reached $OD_{600} = 1–1.5$. All the cultures were then diluted in fresh media at $OD_{600} = 0.1$, and grown until they reached $OD_{600} = 0.6–0.8$. Next, protein expression was induced using 0.5 mM IPTG for 3 h at 25 °C. After protein induction, the optical density (OD) of the culture was measured, and 500 µL of bacteria were collected by centrifugation for 5 min at $11,000× g$ and resuspended directly in Laemmli Buffer to reach a concentration of 5 mOD/µL. In all, 10 µl (50 miliOD) were then analyzed by SDS-PAGE and Western blot.

## MITOKINAC using genetic code expansion to produce pSer112 Bora$^{1-224}$

For the production of Bora$^{1-224}$-6xHis incorporating phosphoserine at Ser112, BL21 B95 (DE3) ΔA ΔfabR ΔserB bacteria were used (Addgene #197655).

In the first step, chemically competent bacteria harboring the plasmid pKW2-EF-Sep (Addgene #173897), which contains an orthogonal aminoacyl-tRNA synthetase/tRNACUA pair, which directs the efficient incorporation of phosphoserine (pSer) into recombinant proteins, were generated.

These bacteria were then co-transformed with pLP2855 and pLP2228 by heat shock for 45 s at 42 °C, followed by recovery in 1 ml SOC medium for 2 h at 37 °C. Co-transformants were selected on an LB plate containing 12.5 µg/ml kanamycin, 12.5 µg/ml streptomycin, and 7 µg/ml chloramphenicol for 24 h at 37 °C. They were then inoculated for an overnight culture in LB with 25 µg/ml kanamycin, 25 µg/ml streptomycin, and 14 µg/ml chloramphenicol. The culture was diluted to $OD_{600} = 0.1$, and protein expression was induced by IPTG for 3 h at 25 °C and processed as described above.

## Recombinant protein expression

Full-length AURKA$^{T288V}$, Plk1$^{1-330}$ kinase domain, Plk1$^{190-225}$ (T-loop), Bora$^{1-224}$, Bora$^{18-120}$ were expressed in *Escherichia coli* BL21 DE3 (Thermo Fisher Scientific), the fragment 10-360 of human ERK was expressed in *E. coli* BL21 DE3 pLyS RIL (Thermo Fisher Scientific). Transformed bacteria were incubated in LB (ERK, Bora$^{18-120}$) or TB (AURKA$^{T288V}$, Plk1$^{1-330}$, Plk1$^{190-225}$, Bora$^{1-224}$) media at 37 °C under agitation, and the cultures were shifted to 18 °C at OD = 0.8. Protein expression was induced by the addition of IPTG (final concentration 500 µM), and the cultures were incubated overnight at 18 °C under agitation, except for Bora$^{18-120}$. In this case, the culture was shifted to 25 °C and incubated for 3 h at 25 °C in the presence of IPTG.

## Recombinant protein purification

6xHis-ERK$^{10-360}$ was produced essentially as described (Khokhlatchev et al, 1997; Tavernier et al, 2021b). Briefly, bacteria expressing 6xHis-ERK10-360 were pelleted and resuspended in lysis buffer (25 mM HEPES pH 7.5, 300 mM NaCl, 20 mM Imidazole, 1 mM TCEP), supplemented with protease inhibitor cocktail (Roche) and lysed by sonication. Total extract was clarified by centrifugation ($30,000× g$, 30 min). The resulting soluble fraction was loaded on a 5 ml Ni$^{2+}$ HiTrap Chelating HP (Cytiva) and eluted with an imidazole gradient on an Akta FPLC system (Cytiva). The pure fractions were collected and concentrated using an ultrafiltration centrifugal protein concentrator (MerckMilipore). The protein was finally resolved on a Hi-Load 16/600 Superdex 200 pg sizing column (Cytiva) equilibrated in 25 mM HEPES pH 7.5, 200 mM NaCl, 1 mM TCEP. The fractions containing the protein were pooled and concentrated using an ultrafiltration centrifugal protein concentrator. The purified protein was aliquoted, flash-frozen, and stored at −80 °C.

6xHis-TEV-Bora$^{1-224}$ WT were cloned as a TEV cleavable 6-His fusion, which leaves non-native residues (GAMDPEF) at the N-terminus after TEV digestion. Bacteria pellets were resuspended in lysis buffer (25 mM Tris pH 7.5, 150 mM NaCl, 20 mM Imidazole, 1 mM TCEP, 2 mM PMSF) and lysed by sonication at 4 °C. Total extract was clarified by centrifugation ($30,000× g$, 30 min), and the resulting insoluble material was resuspended in 25 mM Tris pH 7.5, 6 M Guanidine HCL, 1 mM TCEP at room temperature, sonicated, and clarified by centrifugation ($30,000× g$, 30 min). The supernatant was passed through a 0.45-µm filter and loaded on a 5 ml Ni$^{2+}$ Hitrap Chelating HP (Cytiva) equilibrated in 25 mM Tris pH 7.5, 6 M Guanidine HCL, 1 mM TCEP. After extensive washes, 6xHis-Bora$^{1-224}$ was eluted by 5 column volumes (CV) of elution buffer (25 mM Tris pH 7.5, 6 M Guanidine HCl, 500 mM Imidazole, 1 mM TCEP). Fractions containing the protein of interest were pooled and dialyzed against the dialysis buffer (25 mM Tris pH 7.5, 150 mM NaCl, 1 mM TCEP) for 4 h to remove the guanidine HCL. 6xHis-Bora$^{1-224}$ precipitated in the absence of Guanidine HCl. The content of the dialysis bag was transferred into a 50 ml conical Falcon tube and centrifuged at $4000× g$ for 10 min. The supernatant was discarded, and the resulting pellet containing Bora was resuspended overnight by agitation in 50 ml of resuspending buffer (25 mM Tris pH 7.5, 150 mM NaCl, 2 mM TCEP). All the subsequent steps were performed at 4 °C. The solubilized was clarified by centrifugation at $4000× g$ for 10 min. The supernatant was concentrated to 5 ml using an ultrafiltration centrifugal protein concentrator (MerckMilipore) and incubated overnight with 2 mg of 6xHis-TEV protease. Cleaved Bora was then loaded on a 5 ml Ni$^{2+}$ HiTrap Chelating HP (Cytiva) to remove 6xHis-TEV and 6xHis cleaved tag. The Flow-through containing Bora was concentrated using an ultrafiltration centrifugal protein concentrator. Bora was finally resolved on a Superdex 200 16/600 GL sizing column (Cytiva) equilibrated in 25 mM Tris pH 7.5, 150 mM NaCl, 1 mM TCEP. The fractions containing Bora were pooled and concentrated using an ultrafiltration centrifugal protein concentrator. The purified protein was aliquoted, flash-frozen, and stored at −80 °C.

MBP-TEV-Bora$^{18-120}$ WT and mutants were cloned as a TEV-cleavable MBP fusion. Bacteria were pelleted and resuspended in

lysis buffer (50 mM Tris pH 7.5, 500 mM NaCl, 1 mM DTT), supplemented with protease inhibitor cocktail (Roche), and lysed by sonication at 4 °C. Total extract was clarified by centrifugation ($30,000\times g$, 30 min). The soluble fraction was loaded on 2 ×1 ml MBP-Trap columns (Cytiva), and after washing, the protein of interest was eluted with 15 mM maltose on an AKTA pure system (Cytiva). The pure fractions were collected and injected on a HiLoad 16/600 Superdex 75 pg sizing column (Cytiva) equilibrated in 50 mM Tris pH 7.5, 500 mM NaCl, 1 mM DTT. The fractions containing Bora were pooled and concentrated using an ultrafiltration centrifugal protein concentrator. Finally, the buffer was exchanged using a G25 column for a storage buffer containing 10 mM HEPES pH 7.8, 100 mM KCl, 1 mM CaCl2, 50 mM Sucrose. The purified protein was aliquoted, flash-frozen, and stored at −80 °C.

6xHis-GST-TEV-AURKA[1-403] T288V was cloned as a TEV cleavable 6xHis-GST fusion, which leaves two non-native residues (GA) at the N-terminus after TEV digestion. Bacteria were pelleted and resuspended in lysis buffer (25 mM HEPES pH 7.5, 500 mM NaCl, 1 mM DTT) supplemented with protease inhibitor cocktail (Roche) and lysed by sonication at 4 °C. Total extract was clarified by centrifugation ($30,000\times g$, 30 min). The soluble fraction was loaded onto a 1× 5ml GST-Trap 4B (Cytiva) column. The TEV protease was added to the 1x5 ml GST-Trap 4B (Cytiva) column with the soluble fraction loaded for 8 h. AURKA cleaved protein was then eluted by gravity with the washing step, and the 6-His-GST-tag was eluted with the elution buffer containing 25 mM Hepes pH 7.5, 500 mM NaCl, 20 mM Glutathione, 1 mM DTT (pH with NaOH). The fractions containing cleaved AURKA were pooled, and the salt concentration was lowered to 50 mM NaCl by dilution, and the protein was loaded on 2 ×1 mL HiTrap SP HP column (Cytiva) and eluted with a salt gradient. The pure fractions were collected and concentrated using an ultrafiltration centrifugal protein concentrator (Merck Millipore). The protein was finally resolved on a HiLoad 16/600 Superdex 200 pg sizing column (Cytiva) equilibrated in 25 mM Tris pH 7.5, 150 mM NaCl, 1 mM DTT. The fractions containing the protein were pooled and concentrated using an ultrafiltration centrifugal protein concentrator. The purified protein was aliquoted, flash-frozen in liquid nitrogen, and stored at −80 °C.

6xHis-GST-TEV-Plk1 Kinase Domain[1-330] K82R Kinase Dead, WT or mutated were cloned as a TEV cleavable 6xHis-GST fusion, which leaves seven non-native residues (GAMDPEF) at the N-terminus after TEV digestion. Bacteria were pelleted and resuspended in lysis buffer (50 mM Tris-HCl pH 7.5, 500 mM NaCl, 1 mM DTT) supplemented with protease inhibitor cocktail (Roche) and lysed by sonication at 4 °C. Total extract was clarified by centrifugation ($30,000\times g$, 30 min). The soluble fraction was loaded onto a 1 ml GST-Trap FF (Cytiva) column. The TEV protease is added into the 1 ml GST-Trap FF (Cytiva) column with the soluble fraction loaded for 8 h. Plk1 cleaved protein was then eluted by gravity with the washing step, and the 6xHis-GST-tag was eluted with the elution buffer containing 100 mM Tris-HCl pH 8.8, 500 mM NaCl, 20 mM Glutathione, 1 mM DTT. The fraction containing Plk1 cleaved are pooled and the salt concentration was lowered to 50 mM NaCl by dilution and the protein was loaded on a 1 mL HiTrap SP HP column (Cytiva) (Binding buffer: 50 mM Tris-HCl pH 7.5, 50 mM NaCl, 1 mM DTT) and eluted with a salt gradient (Elution buffer: 50 mM Tris-HCl pH 7.5, 500 mM NaCl,

1 mM DTT). The pure fractions were collected and concentrated using an ultrafiltration centrifugal protein concentrator (Merck Millipore). The protein was finally resolved on a HiLoad 16/600 Superdex 75 pg sizing column (Cytiva) equilibrated in 50 mM Tris pH 7.5, 150 mM NaCl, 1 mM DTT. The fractions containing the protein were pooled and concentrated using an ultrafiltration centrifugal protein concentrator. The purified protein was aliquoted, flash-frozen in liquid nitrogen, and stored at −80 °C.

GST-TEV-Plk1[190-225] T-loop. Bacteria pellets were resuspended in lysis buffer (50 mM Tris-HCl pH 7.5, 500 mM NaCl, 1 mM DTT) supplemented with protease inhibitor cocktail (Roche) and lysed by sonication at 4 °C. Total extract was clarified by centrifugation ($30,000\times g$, 30 min). The soluble fraction was loaded onto 1 ×5 ml GST-Trap 4B (Cytiva) column, and after washing, proteins were eluted using 20 mM Glutathione (Elution buffer: 100 mM Tris-HCl pH 8.8, 500 mM NaCl, 20 mM Glutathione, 1 mM DTT). The pure fractions were collected and concentrated using an ultrafiltration centrifugal protein concentrator (Merck Millipore). The protein was finally resolved on a HiLoad 16/600 Superdex 200 pg sizing column (Cytiva) equilibrated in 25 mM Tris pH 7.5, 150 mM NaCl, 1 mM DTT. The fractions containing the protein were pooled and concentrated using an ultrafiltration centrifugal protein concentrator. The purified protein was aliquoted, flash-frozen in liquid nitrogen, and stored at −80 °C.

The concentration of all the purified recombinant proteins was determined using Bradford reagent (Thermo Fisher Scientific).

## Biochemistry

### ADP Glo™ assay

The ATPase activity of recombinant AURKA was evaluated using the ADP-Glo assay Kinase kit (Promega). Activators were titrated into 2 nM of AURKA[T288V] in a buffer containing 40 mM Tris pH 7.5, 150 mM NaCl, 10 mM MgCl₂, 1 mM DTT, 0.1 mg/ml BSA, 0.01% Brij 35, and 100 µM Kemptide. The final reaction volume was 20 µL. Reactions were initiated by the addition of 10 µM of ATP and incubated at room temperature for 60 min. Reactions were terminated by transferring 10 µL of the reaction mix to a 384-well white plate (Lumitrec 200, VWR) and adding 10 µL of ADP-Glo™ Reagent (Promega). After a 40-min incubation, 20 µL of kinase detection reagent (Promega) was added and allowed to incubate for an additional 30 min. Luminescence was measured on a Biotek Synergy Neo plate reader (BioTek) using a 1-s integration time. Results were plotted in GraphPad Prism V8.4.2 using a one-site total binding curve fitting analysis.

### Fluorescence polarization binding assay

FITC-Tpx2[1-43], unlabeled Tpx2[1-43], GK51, and MK51 were synthesized by Proteogenix (Schiltigheim, France). AURKA[T288V] and FITC-Tpx2[1-43] were mixed in FP buffer (25 mM HEPES pH 7.5, 50 mM NaCl, 10 mM MgCl₂, 0.1 mg/ml BSA (Sigma), 0.01% Brij, 3 mM ADP (Sigma), 2 mM DTT) in a 384-well flat-bottom black plate (Corning 3573) at a final volume of 25 µL. Each data point was derived from a technical triplicate, and each binding curve was performed in triplicate. For the competitive displacement assay, FITC-Tpx2[1-43] (5 nM) and AURKA protein concentrations (35 nM) were kept constant throughout experiments, and unlabeled competitor (either Bora or Tpx2) was added to final concentrations indicated. Fluorescence polarization was measured on a BioTek

Synergy Neo plate reader (BioTek) using Gen5 v2.05 software with excitation and absorbance at 485/528 nm, respectively. Binding graphs and the derived binding constants were generated using GraphPad Prism v8.1.2 and v8.3 (GraphPad).

### Kinase assays

The assay was performed in two steps. Step 1 consisted of the phosphorylation of 2.5 µM of Bora fragments by Erk (molar ratio 10:1) in 10 µL of kinase buffer (25 mM HEPES pH 7.5, 150 mM NaCl, 20 mM $MgCl_2$, 1 mM DTT, 200 µM ATP) incubated for 1 h at 30 °C. Step 2 consisted of the phosphorylation of 160 nM of $Plk1^{K82R}$ by AURKA (molar ratio 1:1) in the presence of 400 nM of Bora from step 1 in 30 µL of kinase buffer, incubated 30 min at 30 °C.

### SDS-PAGE, Western blot experiments

Samples were electrophoresed on SDS-PAGE (12% acrylamide home-made gel) and electro-transferred to nitrocellulose membrane (0.45 µm NC, 10600002; Amersham Protran). Next, Western blots were probed with primary and appropriate secondary antibodies. Western blots were performed following standard procedures.

Antibodies used in this study are listed below. HRP-conjugated anti-rabbit and anti-mouse antibodies (Sigma-Aldrich) were used at 1:5000, and the signal was detected by chemiluminescence (Millipore) with ChemiDoc MP Imaging System (Bio-Rad).

Quantification of Western blot images was performed using ImageJ (FIJI) and ImageLab software (Bio-Rad), with adjusted values obtained by normalizing the pT210/Plk1, pT210/GST-$Plk1^{T-loop}$, or pS10/H3 signal ratio.

### Interphase Xenopus egg extracts

#### Preparation of Xenopus extracts

Interphase egg extracts were obtained from dejellied unfertilized eggs transferred into MMR solution (1.25 mM HEPES-NaOH pH 7.7, 25 mM NaCl, 0.5 mM KCl, 0.25 mM $MgCl_2$, 0.025 mM NaEGTA,) supplemented with Ca2+ ionophore addition (final concentration 2 µg/ml) and 35 min later recovered and rinsed with XB buffer (10 mM HEPES at pH 7.7, 100 mM KCl, 0.1 mM $MgCl_2$, 50 mM Sucrose, 5 mM NaEGTA,) and subsequently centrifuged twice for 20 min at 10,000× g and the cytoplasmic fractions recovered in aliquots of 70 µL (Lorca et al, 2010). In total, 1 µL of Interphase egg extracts was loaded on a polyacrylamide gel and transferred onto N Immobilon-P membranes. Membranes were blocked either with 5% milk TBST for the detection of hs Gwl, Xl Gwl, Plx1, and hs Bora proteins, or with 2% BSA TBST for the detection of pPlx1 and P-Y-15 CDK1 phospho-proteins. Rabbit peroxidase (HRP)-conjugated secondary antibodies were used. Proteins reacting with the antibodies were detected using ECL reaction (Merck Millipore).

#### Immunodepletion of Xenopus extracts

To deplete Xenopus egg extracts, immunoprecipitations were carried out for 20 min at room temperature using 2 µg of affinity-purified antibodies immobilized on 20 µl protein G-Dynabeads (Dynal) and 20 µl of Xenopus egg extracts. When supernatant was used, beads were removed by magnetic racks, and supernatants were recovered. Two consecutive immunoprecipitations were performed to completely remove endogenous Bora proteins. Bora immunoprecipitation was performed using antibodies raised against GST-full-length Xenopus Bora, whereas Western blot was performed with antibodies against Xenopus-Nter-Bora (Vigneron et al, 2018).

### Protein sequence alignment

Sequences were aligned using ClustalW (Larkin et al, 2007) and visualized with Jalview (Waterhouse et al, 2009).

### AlphaFold3 modeling

Protein sequences were downloaded from Uniprot. We accessed AlphaFold 3 from its virtual server (https://alphafoldserver.com/welcome) to run pairwise predictions with 5 models per prediction. Predicted structures were visualized with ChimeraX-1.8 (Pettersen et al, 2021; Meng et al, 2023).

*AlphaFold3 model: Bora•AURKA•Plk1^KDom complex:*
>$AURKA^{126-403}$ (278 aa)
RQWALEDFEIGRPLGKGKFGNVYLAREKQSKFILALKVLF-KAQLEKAGVEHQLRREVEIQSHLRHPNILRLYGYFHDATRVY-LILEYAPLGTVYRELQKLSKFDEQRTATYITELANALSYCHSKR-VIHRDIKPENLLLGSAGELKIADFGWSVHAPSSRRTTLCGTL-DYLPPEMIEGRMHDEKVDLWSLGVLCYEFLVGKPPFEANTY-QETYKRISRVEFTFPDFVTEGARDLISRLLKHNPSQRPMLREV-LEHPWITANSSKPSNCQNKESASKQS
>$Bora^{18-116}$ (99 aa)
RIPVLNPFESPSDYSNLHEQTLASPSVFKSTKLPTPGKFRW-SIDQLAVINPVEIDPEDIHRQALYLSHSRIDKDVEDKRQ-KAIEEFFTKDVIVPSPWTD
> $Plk1^{KDom\ 1-332}$ (332 aa)
MSAAVTAGKLARAPADPGKAGVPGVAAPGAPAAAPPA-KEIPEVLVDPRSRRRYVRGRFLGKGGFAKCFEISDADTKEV-FAGKIVPKSLLLKPHQREKMSMEISIHRSLAHQHVVGFHGF-FEDNDFVFVVLELCRRRSLLELHKRRKALTEPEARYYLR-QIVLGCQYLHRNRVIHRDLKLGNLFLNEDLEVKIGDFGLATK-VEYDGERKKTLCGTPNYIAPEVLSKKGHSFEVDVWSIGCI-MYTLLVGKPPFETSCLKETYLRIKKNEYSIPKHINPVAA-SLIQKMLQTDPTARPTINELLNDEFFTSGYIPARLPITCL-TIPPRFSIAPSSL
*Atomic structures:*
AURKA•Tpx2
PDB: 1OL5 [https://doi.org/10.2210/pdb1OL5/pdb] (Bayliss et al, 2003; McIntyre et al, 2017)
AURKA•Cep192
PDB: 8PR7 [https://doi.org/10.2210/pdb8PR7/pdb] (Holder et al, 2024),
PBD: 8GUW [https://doi.org/10.2210/pdb8GUW/pdb] (Park et al, 2023)
AURKA•TACC3
PDB: 5ODT [https://doi.org/10.2210/pdb5ODT/pdb] (Burgess et al, 2018).

## Data availability

This study includes no data deposited in external repositories.

The source data of this paper are collected in the following database record: biostudies:S-SCDT-10_1038-S44318-025-00679-8.

## Peer review information

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

## Acknowledgements

We thank P Moussounda for helping with media preparation. R Karess for the critical reading of the manuscript. We are grateful to Marc Plays and the ZEFIX animal facility for antibody production. We thank A Esposito Verza and A Musacchio for discussions and for sharing results before publication. AP is supported by a PhD Fellowship from La "Ligue Nationale Contre le Cancer". CBC is supported by a PhD Fellowship from CBS2 Montpellier University. NJ is supported by a grant from the Fondation ARC pour la Recherche sur le Cancer (ARCPJA2022050005002) and a grant from Université Paris-Cité IDEx (ANR-18-IDEX-0001, "Emergence en Recherche" RS30J23IDX64_KATAREP). Work in the laboratory of TL is supported by grants from "Agence Nationale pour la Recherche" (ANR, France – MITODISSECT, ANR-22-CE13-0022) and by the « fondation ARC », ARC 2023 PJA3. Work in the laboratory of FS is supported by a Canadian Institutes of Health Research grant (PJT-178026) and a Terry Fox Research Institute grant (TFRI-1107-04). Work in the laboratory of LP is supported by grants from "Agence Nationale pour la Recherche" (ANR, France —MITODISSECT, ANR-22-CE13-0022), by the IdEx Université Paris Cité, ANR-18-IDEX-0001, and by the "Ligue Nationale Contre le Cancer" (Equipe labéllisée, France).

## Author contributions

**Anaïs Pillan**: Conceptualization; Formal analysis; Investigation; Visualization; Methodology; Writing—review and editing. **Philippine Ormancey**: Formal analysis. **Celia Ben Choug**: Formal analysis. **Stephen Orlicky**: Formal analysis. **Nicolas Tavernier**: Resources; Formal analysis. **Lucie Van Hove**: Resources; Formal analysis; Validation; Methodology. **Batool Ossareh-Nazari**: Investigation; Methodology. **Nicolas Joly**: Funding acquisition; Investigation; Methodology. **Frank Sicheri**: Resources; Supervision; Investigation; Methodology; Writing—review and editing. **Thierry Lorca**: Conceptualization; Resources; Data curation; Formal analysis; Supervision; Funding acquisition; Validation; Investigation; Methodology. **Lionel Pintard**: Conceptualization; Resources; Data curation; Formal analysis; Supervision; Funding acquisition; Validation; Investigation; Visualization; Methodology; Writing—original draft; Project administration; Writing—review and editing.

Source data underlying figure panels in this paper may have individual authorship assigned. Where available, figure panel/source data authorship is listed in the following database record: biostudies:S-SCDT-10_1038-S44318-025-00679-8.

## Disclosure and competing interests statement

The authors declare no competing interests.

# Expanded View Figures

**Figure EV1.  Comparison of the ability of pBora variants and the pTpx2-Bora chimera to stimulate AURKA activity towards Plk1 substrates.** ▶

(A) Schematic of the two-step in vitro reconstitution of T-loop phosphorylation on T210 of Plk1 (pT210) by AURKA$^{T288V}$ and pBora$^{1-224}$ or the pTpx2-Bora chimera. In step 1, pBora$^{1-224}$ or the pTpx2-Bora chimera is incubated with AURKA$^{T288V}$ for 15 min. In step 2, the reaction mix in step 1 is incubated in the presence of Mg/ATP with Plk1 kinase domain catalytically dead mutant (Plk1$^{K82R}$) for 5, 10, 20, 30, and 60 min. Samples were then analyzed by Western blot. Blots were probed with antibodies to Bora, phospho-T210 Plk1, or pan Plk1, and AURKA (from top to bottom). The graph presents the normalized quantification of pT210 Plk1 signal over Plk1$^{KDom}$ from $n = 3$ independent experiments. (B) Same experiment as in panel A, except that AURKA$^{T288V}$ activated by pBora$^{1-224}$ or the pTpx2-Bora chimera is incubated with the isolated Plk1 T-loop fused to GST as substrate. Blots were probed with antibodies to Bora, phospho-T210 Plk1, GST, and AURKA (from top to bottom). The graph presents the normalized quantification of pT210 Plk1 signal over GST-Plk1$^{T-loop}$ from $n = 3$ independent experiments. (C) Schematic of the two-step in vitro reconstitution of T-loop phosphorylation on T210 of the isolated Plk1 T-loop fused to GST by AURKA$^{T288V}$ and pBora$^{18-120}$. In step 1, pBora$^{18-120}$ is incubated 1 h at 30 °C with the ERK kinase and Mg/ATP. In step 2, the reaction mix in step 1 is incubated in the presence of Mg/ATP with AURKA$^{T288V}$ and the isolated Plk1 T-loop fused to GST for 30 min at 30 °C. Western blot analysis of 2-step kinase reactions carried out with MBP-Bora$^{18-120}$ wild-type or mutant phosphorylated (+) or not (−) by the ERK kinase (step 1) in the presence of the T-loop of Plk1 (aa 190-225) fused to GST (substrate) and AURKA$^{T288V}$ (step 2). Blots were probed with antibodies to Bora, phosphoT210 Plk1, GST, and AURKA, as indicated (from top to bottom). The graph presents the normalized quantification of pT210 Plk1 signal over GST-Plk1$^{T-loop}$ from $n = 3$ independent experiments. Error bars display the standard deviation.

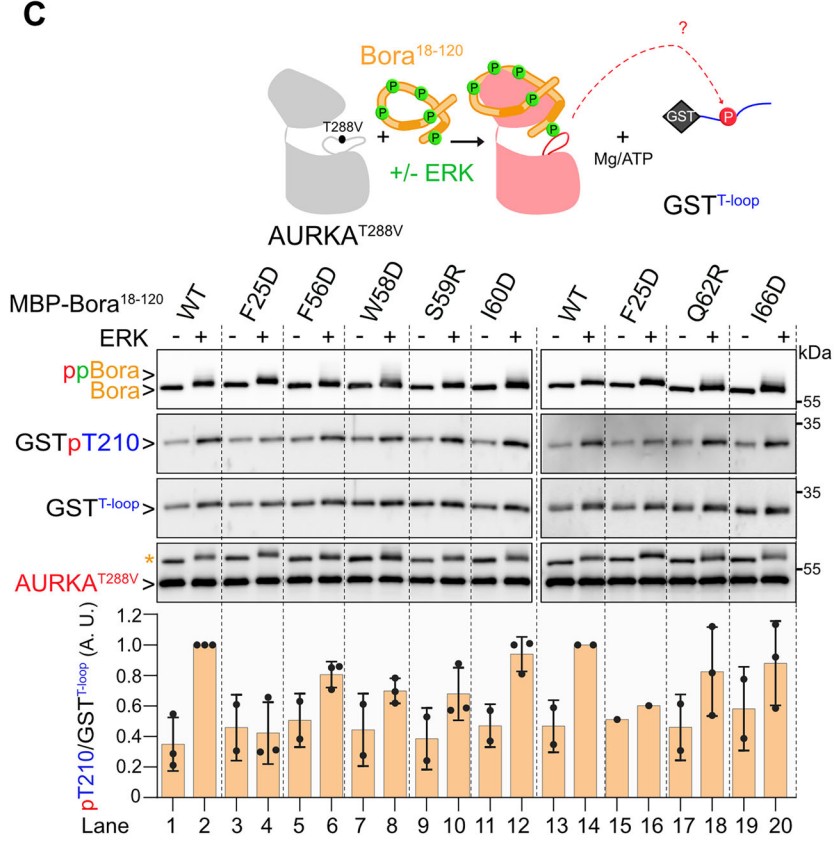

**A**

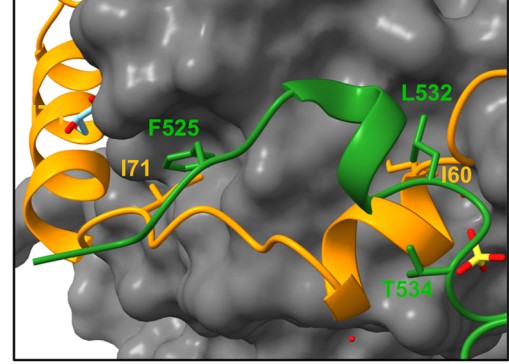

AF3 **Bora** • AURKA

COOH

NH2

ipTM = 0.79, pTM = 0.84

**B**

**Bora** • **TACC3**

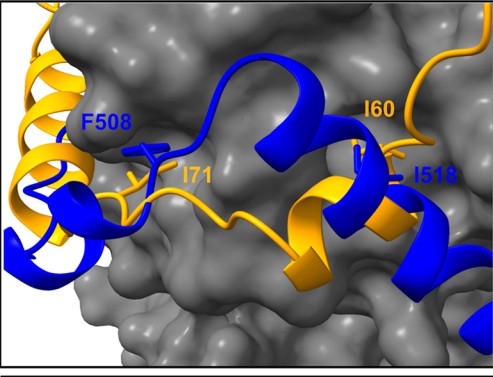

**C**

**Bora** • **Cep192**

**D**

AF3 **Bora** • AURKA

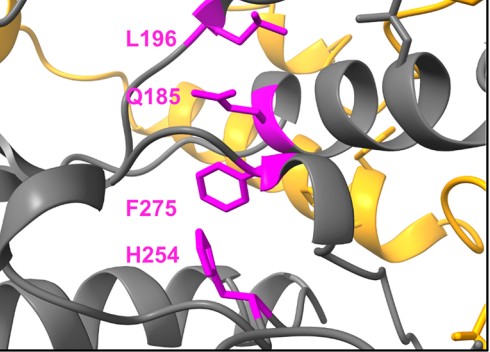

DFG-in **R-spine** assembled

**Figure EV2.   AlphaFold 3 model of pBora bound to AURKA compared to the structure of Cep192 or TACC3 fragments bound to AURKA.**

(A) Superposition of the five pBora▪AURKA complex predictions generated by Alphafold 3. AURKA is in gray and Bora is in orange. The predicted template modeling (pTM) score and the interface predicted template modeling (ipTM) scores are indicated. (B) Superposition of the predicted pBora▪AURKA AlphaFold 3 model with the TACC3▪AURKA structure [PDB: 5ODT] (Burgess et al, 2018). AURKA is shown as gray surface. (C) Superposition of the predicted pBora▪AURKA AlphaFold 3 model with the Cep192▪AURKA structure (PDB: 8PR7) (Holder et al, 2024). AURKA is shown as gray surface. (D) Cartoon representation of the R-Spine assembly in the AURKA▪pBora model predicted by Alphafold3. The R-Spine comprises L196, Q185, F274, and H254 residues (magenta). The « DFG » motif adopts the inward "in" position. When bound to pBora, AURKA exhibits the structural features of an active kinase.

**A**

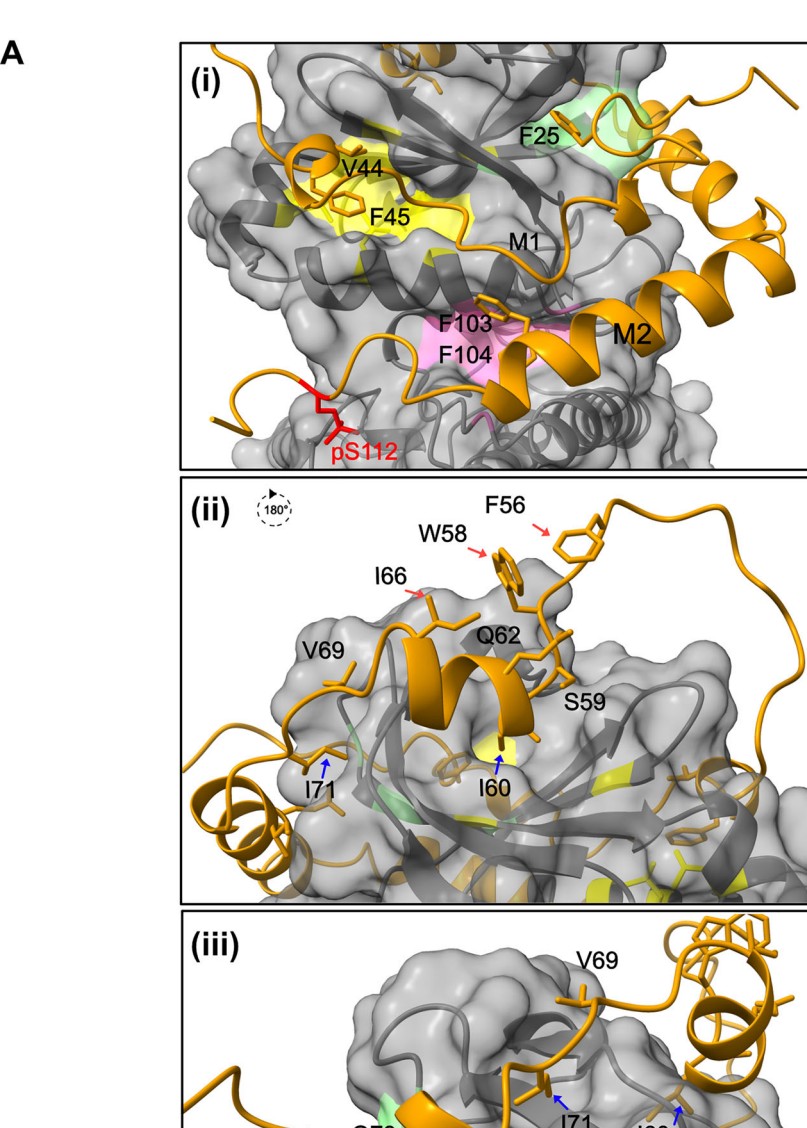

**B**

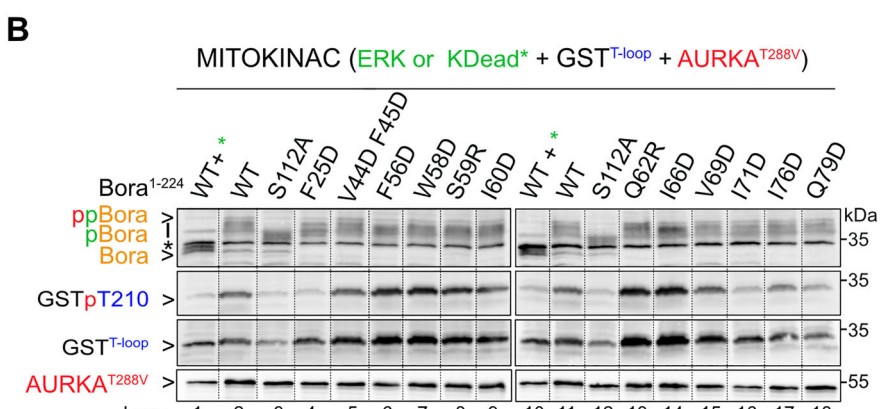

**Figure EV3.  Mutation of the Bora-specific motif does not affect phosphorylation of the isolated Plk1 T-loop by AURKA^T288V.**

(A) (i) View of the M1 and M2 motifs of Bora bound to AURKA predicted by AlphaFold 3. The Bora residues F25, V44, and F45 of the M1 motif binding to the F (green) and Y (yellow) pockets, as well as the residues F103 and F104 of the M2 motif binding to the W pocket (magenta), are shown. (ii) View of the Bora-specific motif bound to AURKA. I60 and I71 (blue arrows) anchor the motif to AURKA, whereas other hydrophobic and aromatic residues, such as F56, W58, and I66 (red arrows), are solvent-exposed. (iii) View of the Bora segment comprising amino acids 60 to 79 bound to AURKA. (B) Western blot analysis of total bacterial extracts reconstituting pBora and AURKA^T288V-dependent T-loop (T210) phosphorylation of the isolated Plk1 T-loop fused to GST (GST^T-loop). Blots were probed with antibodies to Bora, phospho-T210 Plk1, GST, and AURKA (from top to bottom). ERK K-dead kinase dead.

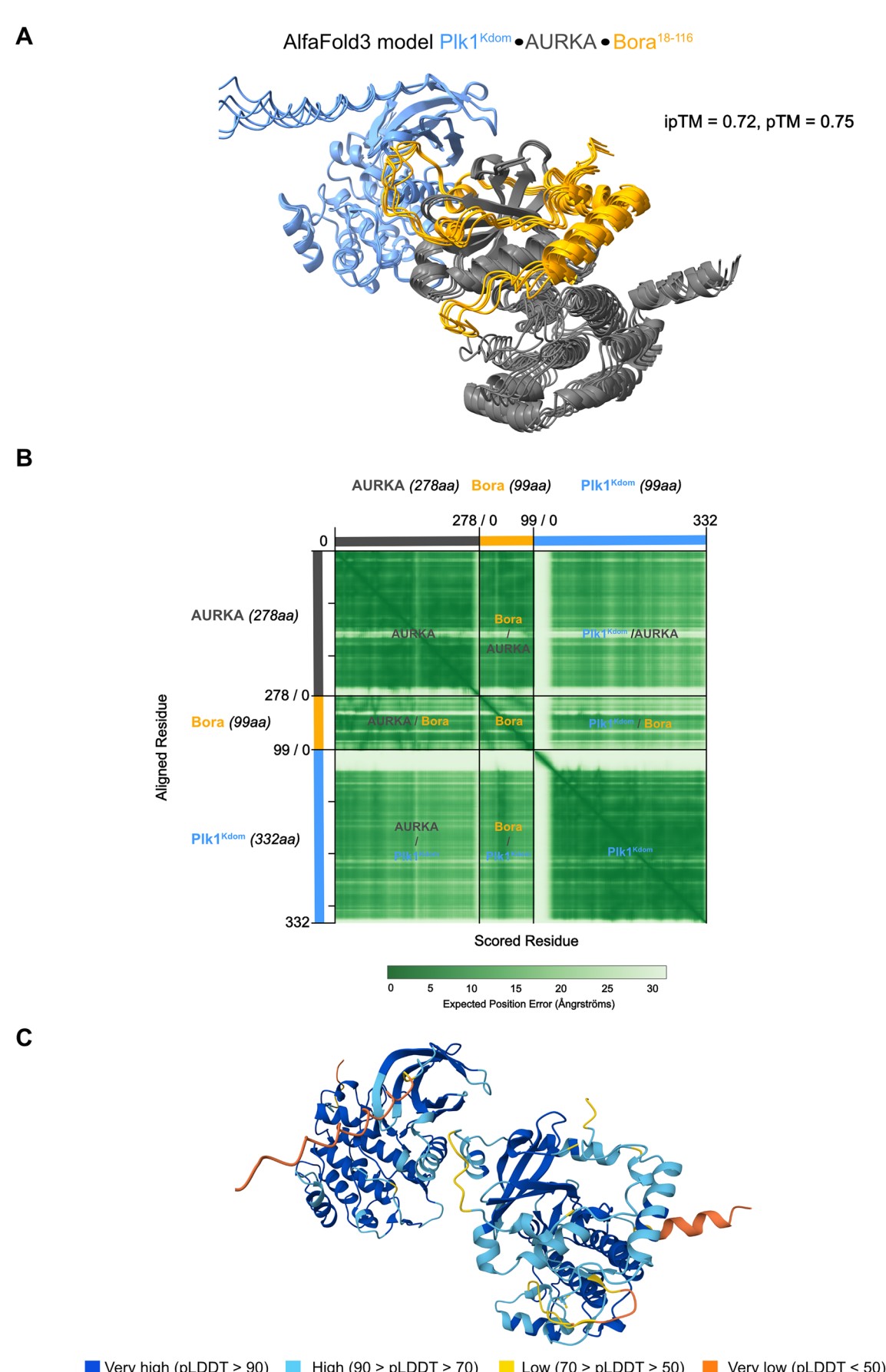

A

AlfaFold3 model Plk1$^{Kdom}$•AURKA•Bora$^{18-116}$

ipTM = 0.72, pTM = 0.75

B

AURKA *(278aa)*   Bora *(99aa)*   Plk1$^{Kdom}$ *(99aa)*

Aligned Residue

AURKA *(278aa)*

Bora *(99aa)*

Plk1$^{Kdom}$ *(332aa)*

Scored Residue

Expected Position Error (Ångrströms)

C

■ Very high (pLDDT > 90)   ■ High (90 > pLDDT > 70)   ■ Low (70 > pLDDT > 50)   ■ Very low (pLDDT < 50)

◀ **Figure EV4. Details on the structural modeling of *H. sapiens* pBora•AURKA•Plk1^KDom complex generated by Alphafold 3.**

(A) Structural alignment of the five pBora▪AURKA▪Plk1^KDom complex predictions generated by Alphafold 3 in a 1:1:1 binding stoichiometry. AURKA is in gray, Plk1 is in blue, and Bora is in orange. The predicted template modeling (pTM) score and the interface predicted template modeling (ipTM) scores are indicated. (B) Predicted Alignment Error (PAE) map calculated by AlphaFold3 showing the confidence of the distances between the residues in contact in the H. sapiens AURKA (126–403) (gray) ▪ Bora (18–116) (orange) ▪ Plk1 (1–332) (blue) complex generated by AlphaFold 3 in a 1:1:1 binding stoichiometry. (C) Per-residue confidence (pLDDT) of the pBora▪AURKA▪Plk1^KDom complex predicted by AlphaFold 3.

