## [Peer Review File · The EMBO Journal]

Molecular basis for the activation of Aurora A and Plk1 kinases during mitotic entry

Anaïs Pillan, Philippine Ormancey, Celia Ben Choug, Stephen Orlicky, Nicolas Tavernier, Lucie Van Hove, Batool Ossareh-Nazari, Nicolas Joly, Frank Sicheri, Thierry Lorca, and Lionel Pintard

Corresponding author: Lionel Pintard (lionel.pintard@ijm.fr)

Review Timeline:	Transferred from Review Commons:	3rd Nov 25
	Editorial Decision:	20th Nov 25
	Revision Received:	16th Dec 25
	Accepted:	17th Dec 25

Editor: Hartmut Vodermaier

Transaction Report:

This manuscript was transferred to The EMBO Journal following peer review at Review Commons.

Review #1**1. Evidence, reproducibility and clarity:****Evidence, reproducibility and clarity (Required)**

Pillan and colleagues reconstitute the activation module that is known to exist between Aurora A and PLK1 (PLK1 T210 being a putative Aurora A substrate) in vitro in the presence of one of the known Aurora A activators, BORA. BORA S112A mutation prevents (timely) mitotic entry in *Xenopus* egg extracts and human cells. Careful modelling, biochemistry and cellular analysis shows that BORA is phosphorylated on Ser112 (this is a CDK site), which mimics activation of Aurora A (normally occurring itself through T-loop phosphorylation), thus generating an active conformer of Aurora A, which is then able to phosphorylate T210 of PLK1 with help from guidance by BORA.

2. Significance:**Significance (Required)**

This is a strong study worthy of publication, and will be of interest to a broad audience of kinase, cell cycle and mechanistic biologists, blending as it does excellent and well controlled work with timely AF3 modelling; a strength is the *E.coli* work, the confirmatory work in *Xenopus*, which agrees with the model of sequential phosphorylation from CDKs to PLK1 via BORA/Aurora A, and together which helps resolve the issue of how Aurora A regulates PLK1 via pBORA.

There are no major limitations, and the work is a significant advance. Highlights include the finding that fusing the phospho-M3 motif of Bora to Tpx2 is sufficient to potentiate AURKAT288V catalytic activity towards PLK1, the use of genetic code expansion to install a pSer on BORA, the exploitation of AF3 to nail the mechanism, development of an in vitro (*E.coli*) model to assess the activation mechanism using multiple proteins and a convincing *Xenopus* assay. Overall, this is an excellent study worthy of publication.

****Minor points:****

1. ERK is used as a proxy for CDK1 (Figure 4E). But is ERK active in the *E.coli* system due to the MEK fusion, ie, can you confirm it is providing the BORA phosphorylation signal given how many other proteins are present? A kinase-dead ERK suggests this is the case (Figure 5), but I found the Figure a bit confusing.
2. An analysis of intramolecular Aurora A and TPX2 regulation was first published in PMID:

14701852, comparing T295A/D/E and highlighting the importance of Y8/Y10(the hydrophobic 'plug'), which is not cited. Noteworthy (but mysterious) was that TPX2 generates Aurora A activity towards TPX2 in T295A (more in D/E), but only towards TPX2, not an exogenous substrate (HH3). This may make sense now!

3. There might be a critical comment about why T288/T295A/V mutants might still be active in biological systems in the presence of appropriate activators, which might circumvent Aurora A T-loop phosphorylation, and indeed subvert the role of phosphatases.

3. How much time do you estimate the authors will need to complete the suggested revisions:

Estimated time to Complete Revisions (Required)

(Decision Recommendation)

Less than 1 month

4. Review Commons values the work of reviewers and encourages them to get credit for their work. Select 'Yes' below to register your reviewing activity at Web of Science Reviewer Recognition Service (formerly Publons); note that the content of your review will not be visible on Web of Science.

No

Review #2

1. Evidence, reproducibility and clarity:

Evidence, reproducibility and clarity (Required)

This study builds on the recent findings from the same group (Tavernier et al. Nat Commun, 2021), that identified the ability of the coactivator Bora phosphorylated by CDK1 to fully activate Aurora-A kinase during the mitotic entry. Here, the authors performed structural modeling using AlphaFold and validated the predictions by in vitro kinase assays and this led to identification of several conserved short motifs in the Bora sequence that determine its function. Motifs M1, BSM, M2 and M3 in the N-terminal region of Bora promote its binding to Aurora-A and phosphorylation of S112 (within M3) promotes Aurora-A activation. Subsequently, they developed a nice assay called MITOKINAC that allowed efficient functional testing of an impressive set of 39 various mutants of Bora and identified several key residues in the above-mentioned motifs. Finally, using similar approaches, they

identified several key residues that mediate transient interaction between the BSM motif of Bora and the catalytic domain of PLK1 and promote its modification by Aurora-A. The functional relevance of the identified structural determinants of Aurora-A:Bora:PLK1 interaction and activation was further validated in *Xenopus* egg extracts where the mutants in the critical residues failed to promote mitotic entry. Overall, the study contains an extensive data set, the methods are described in sufficient detail, and all the presented data is convincing. I have only a minor suggestion for improving the manuscript. Throughout the study, the authors use Aurora-A T288V mutant that cannot be activated by autophosphorylation. I wonder to what extent can pBora boost the activation of the wild-type Aurora-A.

2. Significance:

Significance (Required)

PLK1 is essential for progression through mitosis, and its function depends on modification of its T-loop by Aurora-A and its cofactor Bora. Although these proteins have already been known, the precise molecular organization remained unclear. This comprehensive study explains at molecular detail how the unstructured protein Bora directs the activity of Aurora-A towards PLK1 that is otherwise poor substrate. In addition, the study shows similarities as well as unique features of Bora compared to other Aurora-A cofactors including TPX2. Overall, the study will be of great interest for the broad cell cycle community.

3. How much time do you estimate the authors will need to complete the suggested revisions:

Estimated time to Complete Revisions (Required)

(Decision Recommendation)

Less than 1 month

Yes

Review #3

1. Evidence, reproducibility and clarity:

Evidence, reproducibility and clarity (Required)

****Summary:****

The intrinsically disordered protein Bora has 3 motifs M1/M2/M3 that were long thought to be involved in activating AurKA. The present manuscript uses AlphaFold structural modeling to show how Bora binds to AurKA to activate it. The authors first show that the binding happens in a different manner compared to how another activator, Tpx2 binds to AurKA: Bora wraps around the N-terminal lobe of AurKA to position its phospho-Ser112 near AurKA's T-loop, mimicking and eliminating the need for T-loop phosphorylation.

They then use a high-throughput screening platform (MITOKINAC) based on co-expressing all relevant components in *E. coli*, to test the importance of specific residues. Thereby they are able to systematically map structure and function of M1/M2 residues, revealing a novel Bora-specific motif and how it contributes to function.

They further show that Bora may transiently interact with Plk1 to facilitate Plk1 phosphorylation by AurKA, and finally they test key predictions of their hypothesis in *Xenopus* egg extracts showing how the identified mutants affect Plk1 activation and mitotic entry timing.

The manuscript is overall very clear and of very high quality. The data are presented in large and complex, but at the same time clear and well annotated figures. These data support the conclusions drawn. However, while this is still standard in the field, I would have preferred if Western blots were developed using fluorescent antibodies that in turn would have allowed a much more precise quantification. On quantifications it is very easy to show results from multiple experiments instantly giving an impression of variability. I very much appreciate that this has been done for a few select experiments, but ideally all experiments should be quantified.

****Major comments:****

- could the authors clarify why the pTpx2-Bora chimera and not the pBora-fusion succeed in activation of AurKA and phosphorylation of Plk1? If pTpx2-Bora chimera contains residues of the Bora-specific motif, these should be shown and the specific residues should be

clearly indicated.

- on page 9: "However, the M1 region in Bora is much larger than initially anticipated (extending from residues L22 to F45 rather than 22 to 35) based on structural alignments 59 and is oriented in the opposite direction compared to that observed for Tpx21-43 (Fig. 3B, 3D)." Could the authors clarify, if it's oriented in the opposite direction, and if so, I recommend showing a sequence explaining how the opposite sequence of each would fit in the same region.

****Minor comments:****

- on page 6: "the Bora M2 motif is relatively well conserved between Bora (Fig. 1B, left panel) and Tpx2 (Fig. 1B, right panel) and is located immediately N-terminal to the essential pSer112 (M3 motif) (Fig. 1B)" Could the similarity be highlighted to make it easier to see?
- on Fig. 2B and C are the same as D and thus both could potentially be moved to the supplementary figures.
- the figures are highly complex, and not all details are explained/referred to in the results section.

2. Significance:

Significance (Required)

This is a very important study revealing intricate structural details of Plk1 activation by Aurora A-Bora, critical for mitotic entry. The group identifies structure-function relationships of the Bora-AurKA-Plk1 interactions through structural modeling combined with an advanced screening platform based on co-expression of components in bacteria. The in vitro assays and the systematic methodology are very remarkable in their complexity and detail. Finally, the cellular effects are tested in *Xenopus* egg extracts.

I see this work highly relevant for at least two specific fields: from the structural biology perspective, the study provides a beautiful illustration of the complexity of kinase regulation such as pBora mimicking AurKA T-loop phosphorylation. For the cell cycle community, it provides an explanation for how spatial and temporal specificity in kinase activities may be achieved to orchestrate the highly coordinated reorganization of cellular architecture upon mitotic entry.

The manuscript complements the study by Verza et al. that has been submitted at the same time. Pintard et al. provide more details on the Aurora A-Bora complex, whereas Verza and coworkers focus more on effects on Plk1 activity, which they also extensively

validate in vivo, in the cellular context. Together, these two studies make a very substantial advance in understanding the structural details of Plk1 activation by Aurora A-Bora revealing many unique features. At the same time they provide one of the first models for how mitotic kinase activities are controlled in space and time.

3. How much time do you estimate the authors will need to complete the suggested revisions:

Estimated time to Complete Revisions (Required)

(Decision Recommendation)

Between 1 and 3 months

Yes

Revision Plan

Manuscript number: RC-2025-03133

Corresponding author(s): Lionel, PINTARD

1. General Statements

We would like to thank the Reviewing Editor and the three reviewers who assessed our work. We were particularly pleased to see that all three reviewers considered the study to be both excellent and important for the cell cycle community.

2. Description of the planned revisions

Insert here a point-by-point reply that explains what revisions, additional experimentations and analyses are planned to address the points raised by the referees.

We have addressed all of the reviewers' comments in the revised version of the manuscript by amending the text and figures.

3. Description of the revisions that have already been incorporated in the transferred manuscript

Please insert a point-by-point reply describing the revisions that were already carried out and included in the transferred manuscript. If no revisions have been carried out yet, please leave this section empty.

Below is a point-by-point response to the issues raised by the reviewers, which should help to clarify the rationale behind some of the experiments and their conclusions.

Our response is in blue and italicized. we have also highlighted in blue the parts of the manuscript that we have modified

Reviewer #1 (Evidence, reproducibility and clarity (Required)):

Pillan and colleagues reconstitute the activation module that is known to exist between Aurora A and PLK1 (PLK1 T210 being a putative Aurora A substrate) in vitro in the presence of one of the known Aurora A activators, BORA. BORA S112A mutation prevents (timely) mitotic entry in *Xenopus* egg extracts and human cells. Careful modelling, biochemistry and cellular analysis shows that BORA is phosphorylated on Ser112 (this is a CDK site), which mimics activation of Aurora A (normally occurring itself through T-loop phosphorylation), thus generating an active conformer of Aurora A, which is then able to phosphorylate T210 of PLK1 with help from guidance by BORA.

Reviewer #1 (Significance (Required)):

This is a strong study worthy of publication, and will be of interest to a broad audience of kinase, cell cycle and mechanistic biologists, blending as it does excellent and well controlled work with timely AF3 modelling; a strength is the *E. coli* work, the confirmatory work in *Xenopus*, which agrees with the model of sequential phosphorylation from CDKs to PLK1 via BORA/Aurora A, and together which helps resolve the issue of how Aurora A regulates PLK1 via pBORA.

Revision Plan

There are no major limitations, and the work is a significant advance. Highlights include the finding that fusing the phospho-M3 motif of Bora to Tpx2 is sufficient to potentiate AURKAT288V catalytic activity towards PLK1, the use of genetic code expansion to install a pSer on BORA, the exploitation of AF3 to nail the mechanism, development of an in vitro (E.coli) model to assess the activation mechanism using multiple proteins and a convincing Xenopus assay.

Overall, this is an excellent study worthy of publication

Minor points:

1) ERK is used as a proxy for CDK1 (Figure 4E). But is ERK active in the E.coli system due to the MEK fusion, ie, can you confirm it is providing the BORA phosphorylation signal given how many other proteins are present? A kinase-dead ERK suggests this is the case (Figure 5), but I found the Figure a bit confusing.

We describe MITOKINAC first in Figs. 4B, C, and D. We demonstrate that Plk1 T210 phosphorylation depends on ERK kinase activity, as expression of the kinase-dead version of ERK (using the mutation K54R that abolishes the essential Glu-Lys salt bridge of the kinase) prevents T210 phosphorylation. We included this control in every experiment presented in Figures 4, 5, 6, and 7. To improve the clarity, we modified Fig. 4 to better highlight this control in the schematic that first describes the system. We also cite this control line 336 of the manuscript.

2) An analysis of intramolecular Aurora A and TPX2 regulation was first published in PMID: 14701852, comparing T295A/D/E and highlighting the importance of Y8/Y10 (the hydrophobic 'plug'), which is not cited. Noteworthy (but mysterious) was that TPX2 generates Aurora A activity towards TPX2 in T295A (more in D/E), but only towards TPX2, not an exogenous substrate (HH3). This may make sense now!

We now cite the mentioned reference in the manuscript.

3) There might be a critical comment about why T288/T295A/V mutants might still be active in biological systems in the presence of appropriate activators, which might circumvent Aurora A T-loop phosphorylation, and indeed subvert the role of phosphatases.

We do not investigate the role of the AURKA^{T288V} mutation in vivo or in physiological contexts in this manuscript. However, this version would indeed subvert the in vivo function of the phosphatases.

Reviewer #2 (Evidence, reproducibility and clarity (Required)):

This study builds on the recent findings from the same group (Tavernier et al. Nat Commun, 2021), that identified the ability of the coactivator Bora phosphorylated by CDK1 to fully activate Aurora-A kinase during the mitotic entry. Here, the authors performed structural modeling using AlphaFold and validated the predictions by in vitro kinase assays and this led to identification of several conserved short motifs in the Bora sequence that determine its function. Motifs M1, BSM, M2 and M3 in the N-terminal region of Bora promote its binding to Aurora-A and

Revision Plan

phosphorylation of S112 (within M3) promotes Aurora-A activation. Subsequently, they developed a nice assay called MITOKINAC that allowed efficient functional testing of an impressive set of 39 various mutants of Bora and identified several key residues in the above-mentioned motifs. Finally, using similar approaches, they identified several key residues that mediate transient interaction between the BSM motif of Bora and the catalytic domain of PLK1 and promote its modification by Aurora-A. The functional relevance of the identified structural determinants of Aurora-A:Bora:PLK1 interaction and activation was further validated in *Xenopus* egg extracts where the mutants in the critical residues failed to promote mitotic entry. Overall, the study contains an extensive data set, the methods are described in sufficient detail, and all the presented data is convincing. I have only a minor suggestion for improving the manuscript. Throughout the study, the authors use Aurora-A T288V mutant that cannot be activated by autophosphorylation. I wonder to what extent can pBora boost the activation of the wild-type Aurora-A.

We have previously reported that pBora can also boost the activity of wild-type AURKA, albeit less efficiently than AURKA^{T288V} (Tavernier et al. Nat Com 2021).

Reviewer #2 (Significance (Required)):

PLK1 is essential for progression through mitosis, and its function depends on modification of its T-loop by Aurora-A and its cofactor Bora. Although these proteins have already been known, the precise molecular organization remained unclear. This comprehensive study explains at molecular detail how the unstructured protein Bora directs the activity of Aurora-A towards PLK1 that is otherwise poor substrate. In addition, the study shows similarities as well as unique features of Bora compared to other Aurora-A cofactors including TPX2. Overall, the study will be of great interest for the broad cell cycle community.

Reviewer #3 (Evidence, reproducibility, and clarity (Required)):

Summary: The intrinsically disordered protein Bora has 3 motifs M1/M2/M3 that were long thought to be involved in activating AurKA. The present manuscript uses AlphaFold structural modeling to show how Bora binds to AurKA to activate it. The authors first show that the binding happens in a different manner compared to how another activator, Tpx2 binds to AurKA: Bora wraps around the N-terminal lobe of AurKA to position its phospho-Ser112 near AurKA's T-loop, mimicking and eliminating the need for T-loop phosphorylation.

They then use a high-throughput screening platform (MITOKINAC) based on co-expressing all relevant components in *E. coli*, to test the importance of specific residues. Thereby, they are able to systematically map structure and function of M1/M2 residues, revealing a novel Bora-specific motif and how it contributes to function.

They further show that Bore may transiently interact with Plk1 to facilitate Plk1 phosphorylation by AurKA, and finally they test key predictions of their hypothesis in *Xenopus* egg extracts showing how the identified mutants affect Plk1 activation and mitotic entry timing.

The manuscript is overall very clear and of very high quality. The data are presented in large and complex, but at the same time clear and well annotated figures. These data support the conclusions drawn. However, while this is still standard in the field, I would have preferred if Western blots were developed using fluorescent antibodies that in turn would have allowed a

Revision Plan

much more precise quantification. On quantifications it is very easy to show results from multiple experiments instantly giving an impression of variability. I very much appreciate that this has been done for a few select experiments, but ideally all experiments should be quantified.

We fully agree with this comment. In fact, we initially quantified a few selected in vitro reconstitution experiments when the difference between conditions was not obvious (i. e. Fig. 7D). That said, the experiments have been repeated at least three times, allowing for quantifications of Figure 2 panels A-B-C-D, Figure 5 panels D-H, and Figure 6 panels B-D-F-H, which are now presented in the revised Figures 2A, 2B, 5D, 5H, 6D, 6F, S1A and S1B.

However, please note that we did not quantify the MITOKINAC experiments, as these are better suited to quickly screening mutants than obtaining quantitative results. Quantification is more difficult in co-expression in E. coli because the different components may be expressed at slightly different levels.

Major comments:

- could the authors clarify why the pTpx2-Bora chimera and not the pBora-fusion succeed in activation of AurKA and phosphorylation of Plk1?

The pBora fusion is not functional at activating AURKA^{T288V} for at least two reasons (and possibly others):

- 1- The pBora fusion (peptide GK51, Fig. 1B) contains a truncated M1 motif (22 to 35aa instead of 22 to 45aa), and thus lacks the V₄₅F₄₅ hydrophobic residues binding the Y pocket of AURKA. We originally designed this construct based only on the relatively weak sequence similarities between Bora and Tpx2.*
- 2- In Tpx2, the beginning of the M1 motif (Y₈, Y₁₀) binds the Y pocket, whereas the end of the motif (F₁₆F₁₉) binds the F pocket. However, in Bora, it is the opposite: the beginning of the M1 motif (F₂₅) binds the F pocket of AURKA, while the end of the motif (V₄₄F₄₅) binds the Y pocket.*

If pTpx2-Bora chimera contains residues of the Bora-specific motif, these should be shown and the specific residues should be clearly indicated.

We thank the reviewer for this comment. We apologize for the confusion and need to clarify this important point.

The pTpx2-Bora chimera:

- 1- can activate AURKA T288V towards the kemptide (Fig. 1D) or Histone H3 (Fig. 2A) but not towards the Plk1 kinase domain.*
- 2- lacks the key determinants located between the M1 and M2 motifs (Bora-specific motif) required for binding the Plk1 kinase domain.*

- on page 9: "However, the M1 region in Bora is much larger than initially anticipated (extending from residues L22 to F45 rather than 22 to 35) based on structural alignments 59 and is oriented in the opposite direction compared to that observed for Tpx21-43 (Fig. 3B, 3D)." Could the authors clarify, if it's oriented in the opposite direction, and if so, I recommend showing a sequence explaining how the opposite sequence of each would fit in the same region.

Revision Plan

*We have modified the text and **Fig. 3** to better explain the boundaries of the M1 motif of Bora and the fact that it engages AURKA in an opposite orientation as compared to the Tpx2 M1 motif (lines 279-287).*

Minor comments:

- on page 6: "the Bora M2 motif is relatively well conserved between Bora (Fig. 1B, left panel) and Tpx2 (Fig. 1B, right panel) and is located immediately N-terminal to the essential pSer112 (M3 motif) (Fig. 1B)" Could the similarity be highlighted to make it easier to see?

We have revised this section to provide more background and make it clearer. An alignment showing the sequence conservation between the M1 and M2 motifs of Bora and Tpx2 has already been published (Tavernier et al., 2021, Fig. 5C).

- om Fig. 2B and C are the same as D, and thus both could potentially be moved to the supplementary figures.

We agree with the reviewer and have moved Fig. 2B and C to the supplementary Figure S1. We have also included the quantification of these experiments.

- the figures are highly complex, and not all details are explained/referred to in the results section.

We have checked the text and figures again to ensure that all details are explained in the Results section.

Reviewer #3 (Significance (Required)):

This is a very important study revealing intricate structural details of Plk1 activation by Aurora A-Bora, critical for mitotic entry. The group identifies structure-function relationships of the Bora-AurKA-Plk1 interactions through structural modeling combined with an advanced screening platform based on co-expression of components in bacteria. The in vitro assays and the systematic methodology are very remarkable in their complexity and detail. Finally, the cellular effects are tested in Xenopus egg extracts.

I see this work highly relevant for at least two specific fields: from the structural biology perspective, the study provides a beautiful illustration of the complexity of kinase regulation such as pBora mimicking AurKA T-loop phosphorylation. For the cell cycle community, it provides an explanation for how spatial and temporal specificity in kinase activities may be achieved to orchestrate the highly coordinated reorganization of cellular architecture upon mitotic entry.

The manuscript complements the study by Verza et al. that has been submitted at the same time. Pintard et al. provide more details on the Aurora A-Bora complex, whereas Verza and coworkers focus more on effects on Plk1 activity, which they also extensively validate in vivo, in the cellular context. Together, these two studies make a very substantial advance in understanding the structural details of Plk1 activation by Aurora A-Bora revealing many unique

Revision Plan

features. At the same time they provide one of the first models for how mitotic kinase activities are controlled in space and time.

4. Description of analyses that authors prefer not to carry out

The three reviewers did not request additional experiments.

Dr. Lionel Pintard
CNRS-UMR7592 / Institut Jacques Monod
Team "Cell Cycle & Development"
Buffon B, CNRS UMR7592
15 rue Helene Brion
Paris 75205
France

20th Nov 2025

Re: EMBOJ-2025-122942-T
Molecular basis for the activation of Aurora A and Plk1 kinases during mitotic entry

Dear Lionel,

Thank you for submitting your revised Review Commons manuscript for consideration by The EMBO Journal. Given the interest of the subject and the very positive transferred referee reports, I decided to treat the work like a regular EMBO Journal revision, and returned it directly to the original referee 3, who was fully satisfied with your responses and revision. We shall therefore be happy to accept the study for publication, following adjustment to our specific journal format and incorporation of a few other editorial modifications as follows:

GENERAL:

- Please download and complete our author checklist (link provided below).
- For the editor-written 'Synopsis' accompanying the online version of the article, please provide suggestions for a short 'blurb' text prefacing and summing up the conceptual aspect of the study in two sentences (max. 250 characters), followed by 3-5 one-sentence 'bullet points' with brief factual statements of key results of the paper (ideally use the Highlights listed currently on the first manuscript page, from where they should be removed). Please also upload a synopsis image, which can be used as a "visual title" for the synopsis section of your paper. The image should be in PNG or JPG format, and please make sure that it remains in the modest dimensions of (exactly) 550 pixels wide and 300-600 pixels high.
- You shall also receive a separate message from our Source Data curation team, with instructions on how to prepare and upload relevant image and numerical raw data.

TEXT:

- Please adjust the order as well as the headers of the different manuscript sections: Title page with complete author information, Abstract, Keywords, Introduction, Results, Discussion, Methods, Data Availability, Acknowledgements, Disclosure and Competing Interests Statement, References, Main Figure Legends, Tables, Expanded Figure Legends.
- Please double-check to make sure that each figure panel (e.g. Fig. 5g, Fig. 5H) is called out in the text at least once. If referring to a multi-panel figure as a whole, please simply reference "Fig. 5A-H"
- Please note that Materials and Methods need to be described in the main text using our 'Structured Methods' format (for detail, see <https://www.embopress.org/page/journal/14693178/authorguide#structuredmethods>). The in-text "Methods" section should contain method and protocol descriptions (ideally using a step-by-step protocol format to facilitate adoption of the methodologies across labs), while all key reagents, experimental models, software and relevant equipment - including their sources and relevant identifiers - should be listed in a separately uploaded Reagents and Tools Table, a template for which can be downloaded from the above section of our Author Guidelines.
- As we are switching from a free-text author contribution statement towards a more formal statement based on Contributor Role Taxonomy (CRediT) terms, please remove the present Author Contribution section and instead specify each author's contribution(s) directly in the Author Information page of our submission system during upload of the final manuscript. See <https://casrai.org/credit/> for more information.
- Please rename the Competing Interest section into "Disclosure and Competing Interests Statement", in accordance with our updated Guide to Authors (<https://www.embopress.org/competing-interests>), and remove the duplicate statement included in the Acknowledgement section.
- Please carefully go through the reference list and make sure that each reference is complete with citation year, journal name,

volume/page/locator numbers - some of these informations are currently missing in several entries. Also, for references with more than 10 authors, please make sure to list only the first 10, followed by "et al", in the reference list.

Furthermore, please adjust the format for citation of preprints as specified in our author guidelines:

The citation in the text should be: "(preprint: NAME1 et al, YEAR)"

The citation in the reference list: "NAME1, NAME2, ... (YEAR) ARTICLE TITLE. bioRxiv/medRxiv/ResearchSquare(...) doi: XXX"

For citation of the cosubmitted work, I would suggest to best add a placeholder "in press" citation which could be updated at proof stage.

- Please simplify the "Data Availability" section, which should only refer to deposited datasets generated in the present study, not to source or raw data. Suggested wording: "The [structural coordinates | microarray | mass spectrometry] data from this publication have been deposited to the [name of the database] database [URL] and assigned the identifier [accession | permalink | hashtag]." Should there no data deposition to public repositories linked to the study, this should still be stated as "This study includes no data deposited in external repositories."

- For the "Data and Code Availability" paragraph in the Methods section, please consider adding dedicated Data References/Data Citations, as explained in our Guide to Authors (<https://www.embopress.org/page/journal/14602075/authorguide#referencesformat>)

- Please make sure to include all relevant funding information not only in the manuscript text, but also in our submission system; currently missing is the "grant from Université Paris-Cité IDEX (ANR-18-IDEX-0001, "Emergence en Recherche" RS30J23IDX64_KATAREP)"

DATA:

- Please upload all main Figures and Expanded View figures as individual, image-only files with sufficient resolution/quality for production.

- Please refer to our author guide (www.embopress.org/page/journal/14602075/authorguide#expandedview) regarding "supplementary information". I would suggest to turn the current supplemental figures into Expanded View figures - nomenclature/call-outs "Figure EV1/2/3..." - and their legends should be included at the very end of the manuscript text.

- For "Supplementary Tables S1-2", please either turn them into Expanded View Tables (callout: "Table EV1/2"), uploading them as separate DOCX or XLSX files; or collate them in a dedicated Appendix PDF as "Appendix Tables S1-2"; or -maybe best- consider whether their contents could become incorporated into the required Reagents & Tools table (see above).

- Finally, during routine pre-acceptance checks, our data editors have raised the following queries regarding figures, data, and legends; I would appreciate if you briefly answered to them in the cover letter of your final submission, and made the requested text modifications with changes/additions highlighted via the "Track changes" option, to facilitate our final checking: Please note that the error bars are not defined in the legends of figures 2A, B; 5D, H; 6D, F; 7E, S1A-C

Should you need additional guidance/feedback regarding this final adjustments, please do not hesitate to contact us directly. Thank you again for the opportunity to consider this work for The EMBO Journal, and I look forward to receiving your final version!

With kind regards,

Hartmut

9) To facilitate reproducibility and cross-laboratory adoption of methodologies, please structure the Materials & Methods section as outlined in our guide to authors, including a completed Reagents and Tools Table that can be downloaded from our author guidelines as well (<https://www.embopress.org/page/journal/14602075/authorguide#structuredmethods>).

10) Digital image enhancement is acceptable practice, as long as it accurately represents the original data and conforms to community standards. If a figure has been subjected to significant electronic manipulation, this must be clearly noted in the figure legend and/or the 'Materials and Methods' section. The editors reserve the right to request original versions of figures and the original images that were used to assemble the figure. Finally, we generally encourage uploading of numerical as well as gel/blot image source data; for details see: embopress.org/page/journal/14602075/authorguide#sourcedata

In the interest of ensuring the conceptual advance provided by the work, we recommend submitting a revision within 3 months (18th Feb 2026). Please discuss the revision progress ahead of this time with the editor if you require more time to complete the revisions. Use the link below to submit your revision:

Link Not Available

Referee #1:

We reviewed this manuscript for Review Commons, and evaluated very positively, in accord with the other two reviewers. The revised version now submitted to EMBO Journal addresses all our (relatively minor) criticisms. Importantly, the authors now show quantifications for most Western blots, which I consider a substantial improvement. Therefore, I am happy to recommend the manuscript for publication.

Rev_Com_number: RC-2025-03133

New_manu_number: EMBOJ-2025-122942-T

Corr_author: Pintard

Title: Molecular basis for the activation of Aurora A and Plk1 kinases during mitotic entry

The authors addressed the remaining editorial issues.

Dr. Lionel Pintard
CNRS-UMR7592 / Institut Jacques Monod
Team "Cell Cycle & Development"
Buffon B, CNRS UMR7592
15 rue Helene Brion
Paris 75205
France

17th Dec 2025

Re: EMBOJ-2025-122942R
Molecular basis for the activation of Aurora A and Plk1 kinases during mitotic entry

Dear Lionel,

Thank you for submitting your final revised manuscript for our consideration. I am pleased to inform you that we have now accepted it for publication in The EMBO Journal.

You may qualify for financial assistance for your publication charges - either via a Springer Nature fully open access agreement or an EMBO initiative. Check your eligibility: <https://link.springer.com/journal/44318/how-to-publish-with-us>

With kind regards,

Hartmut

Please note that it is The EMBO Journal policy for the transcript of the editorial process (containing referee reports and your response letters) to be published as an online supplement to each paper. If you should prefer removal of any referee-only figures included in the point-by-point response(s), e.g. because they may still be used for future publication or because they have been reproduced from published work by others, please do let us know immediately via response email.

More information is available here: <https://link.springer.com/partners/embo-press/editorial-policies#Peer%20review>